# Distinct roles of dentate gyrus and medial entorhinal cortex inputs for phase precession and temporal correlations in the hippocampal CA3 area

**Siavash Ahmadi** [1], **Takuya Sasaki** [1,2], **Marta Sabariego** [1], **Christian Leibold** [3], **Stefan Leutgeb** [1,4,5] ✉ & **Jill K. Leutgeb** [1,5] ✉

The hippocampal CA3 subregion is a densely connected recurrent circuit that supports memory by generating and storing sequential neuronal activity patterns that reflect recent experience. While theta phase precession is thought to be critical for generating sequential activity during memory encoding, the circuit mechanisms that support this computation across hippocampal subregions are unknown. By analyzing CA3 network activity in the absence of each of its theta-modulated external excitatory inputs, we show necessary and unique contributions of the dentate gyrus (DG) and the medial entorhinal cortex (MEC) to phase precession. DG inputs are essential for preferential spiking of CA3 cells during late theta phases and for organizing the temporal order of neuronal firing, while MEC inputs sharpen the temporal precision throughout the theta cycle. A computational model that accounts for empirical findings suggests that the unique contribution of DG inputs to theta-related spike timing is supported by targeting precisely timed inhibitory oscillations. Our results thus identify a novel and unique functional role of the DG for sequence coding in the CA3 circuit.

Although it is well established that the hippocampus supports episodic and spatial memory[1,2], the computations that are performed by each of its subregions, particularly by the dentate gyrus (DG), are not well characterized. The sparse activity of the DG granule cells in both the number of active neurons and firing rate has motivated theories that DG facilitates memory formation through the distinct encoding of unique events. This is supported by the observation that spatial firing patterns of DG cells show pattern separation[3–5], which is in turn consistent with the general conceptual framework that the DG mossy fiber projections to CA3 support memory by promoting the generation of new and distinct hippocampal firing patterns[6–10]. However, behavioral

studies have suggested that the role of DG in memory may be broader due to its essential role in support of complex spatial working memory where items need to be held in temporary storage for successful goal-directed behaviors[11,12].

Consistent with the role of DG in working memory in addition to pattern separation, CA3 sharp wave ripples (SWRs) have been shown to occur during working memory, to depend on the DG during working memory, and to predict subsequent correct choices[12]. These findings suggest that the associative circuits created by the dense direct and indirect recurrent pathways in the DG-CA3 network[13,14] support the generation of SWRs (150 – 250 Hz oscillation), which are thought to

[1]Neurobiology Department, School of Biological Sciences, University of California, San Diego, CA, USA. [2]Graduate School of Pharmaceutical Sciences, Tohoku University, Sendai, Japan. [3]Fakultät für Biologie & Bernstein Center Freiburg, Albert-Ludwigs-Universität Freiburg, Freiburg, Germany. [4]Kavli Institute for Brain and Mind, University of California, San Diego, CA, USA. [5]Institute for Advanced Study, Berlin, Germany. ✉e-mail: sleutgeb@ucsd.edu; jleutgeb@ucsd.edu

originate in CA3 with modulation from the CA2 and DG subregions[12,15–17]. SWRs occur during slow-wave sleep and during pauses in ongoing behavior and consist of short bouts of neuronal activity that correspond to a time-compressed replay of sequences from behavioral episodes[18–21]. Decoding of such sequences in CA1 has revealed that they reflect available trajectories within an environment including spatial locations either behind or in front of the animal[22,23]. Stored hippocampal sequences have, in turn, been shown to be important for learning and memory consolidation[24–27] and are also proposed to guide behavior by facilitating decision-making and future route planning[20,21,23,28] (but see ref. 29).

While the relation between SWRs and memory has been predominantly investigated in the hippocampal CA1 region, the finding that DG-dependent SWR activity in CA3 reflects the planning of future action[12] is consistent with a proposed broader role of DG in which its circuits are hypothesized to perform error correction to maintain accurate temporal order during the chaining of CA3 sequences in behavior[30]. However, complex memory tasks are highly dynamic with frequent transitions between brain states associated with predominant frequency ranges, each likely reflecting distinct underlying network mechanisms for memory encoding and retrieval[31]. A complementary brain oscillation that is also strongly associated with sequential neuronal activity is theta (6–10 Hz oscillation). Unlike SWR activity that occurs during non-movement, theta is prominent and critical when memory is encoded during periods of active exploration[32–34].

During theta states, the binding of cells into sequences has been proposed to be supported by theta phase precession[35–37], which is a network mechanism that results in sequences when the spiking of multiple overlapping place cells is organized within each theta cycle. For each place cell, spiking first occurs at a late theta phase upon entry into a place field and at progressively earlier phases of theta at the exit from the field. This mechanism organizes the sequential firing such that place cells that are further ahead of the animal spike later within the theta cycle. The resulting compression of sequences from the behavioral time scale (seconds) to within the time scale of synaptic plasticity (milliseconds) may facilitate the storage of sequences in synaptic matrices[38–40]. At the population level, the precession of individual place cells can lead to spike sequences within each theta cycle[19,37,41,42] that may also reflect past and future paths of the animal[43,44]. Therefore, phase precession is a critical mechanism during theta states for organizing spike sequences during behavior and has been observed in the neural networks of all hippocampal subregions as well as in their theta-modulated input, the medial entorhinal cortex (MEC)[45,46].

Although theta phase precession is observed throughout the hippocampus and entorhinal cortex, most mechanistic models of phase precession have focused on the CA1 region and on recurrent connections in CA3[47–51]. Involvement of the DG in phase precession has initially been suggested in a model that considered the strong synaptic facilitation of mossy fiber synapses onto CA3 cells as a potential source for increasing excitation throughout the extent of the place field[52]. Conversely, the DG has also been noted in network models as a brain region that can complement the direct recurrent CA3 to CA3 connections by a longer recurrent loop that includes dentate mossy cells and dentate granule cells. Accordingly, the connections in this loop have been proposed to include fixed asymmetrical weights that can give rise to sequential and predictive firing patterns[30,53]. Although these computational models raise the possibility that DG may contribute to phase precession with a function that differs from other input pathways, it has not been experimentally tested whether DG inputs are even necessary for phase precession at its direct target cells in CA3, and if necessary, whether the observed effects after selective loss of DG inputs can further constrain computational approaches.

While the effects of DG inputs on theta-related spike timing have not been determined, MEC inputs to CA1 are known to be necessary for

CA1 phase precession and for CA1 cell pairs to maintain their spiking order in theta cycles[54]. However, given that the MEC inputs to CA3 are complemented by a second extrinsic input from strongly phase precessing cells in DG[46], it is possible that MEC inputs are not as strictly required for phase precession in CA3 compared to CA1[12]. Here, we, therefore, compared the contributions of DG inputs and MEC inputs to provide an understanding of the respective contributions of two external excitatory inputs to CA3 theta phase precession. To distinguish the role of the two inputs, we analyzed and compared CA3 network dynamics during theta oscillations from previously published recordings of CA3 cells during working memory tasks with either intact or diminished MEC or DG inputs[12,55]. Based on our finding that DG contributes to prospective coding during SWRs[12], we hypothesized that DG also predominantly controls the emergence of prospective coding during theta oscillations. Our results are consistent with a contribution of DG, but not MEC inputs, to prospective coding and to the organization of temporal relations between CA3 neurons. We devised a phenomenological computational model to synthesize these findings and to make predictions of how the summation of inhibitory and excitatory oscillatory inputs might support phase precession in CA3.

## Results

To test the contribution of DG and MEC inputs to CA3 phase precession, two previously published datasets with recordings of neuronal activity in the rat hippocampal CA3 region were analyzed[12,55]. In these datasets, CA3 cells were recorded in hippocampus-dependent working memory tasks after lesioning either dentate granule neurons or the MEC (Supplementary Figs. S1, S2a, b), and each lesion group was paired with a respective control group (Supplementary Table S1). Because the working memory tasks required the rats to follow chosen trajectories, coverage of space was inevitably non-uniform. We, therefore, reasoned that the method of defining spike trains by first identifying place fields and then identifying passes through fields may not be precise as a result of uneven coverage and directly identified bouts of each cell's increased neuronal activity by selecting spike trains from the temporal firing patterns (see Supplementary Fig. S2c for details on the criteria). We only considered neuronal activity during locomotion (Supplementary Fig. S2d) when there is a reliable occurrence of theta oscillations. Given that we included only spikes within trains and during movement periods, only a proportion of all recorded spikes (34.3% vs. 20.4% DG control and lesion; 30.6% vs. 22.5% MEC control and lesion) were included as qualifying trains and further analyzed (Supplementary Fig. S3). While spike trains were identified solely by timing and velocity criteria, we confirmed that the trains clustered preferentially at one or few spatial locations, as would be expected for CA3 place cells. In addition, we confirmed that there were only minor differences in firing rate and spatial precision measurements when comparing cells between the control group for DG lesions and the control group for MEC lesions, even though the number of spikes per train and the path length during trains differed between these groups (Supplementary Fig. S3a). These analyses confirm that the spatial firing characteristics of the two control datasets are comparable even though they were taken from two different spatial working memory tasks[12,55].

Furthermore, we also examined whether lesioning a large proportion of DG or MEC inputs to CA3, which are each excitatory, had major effects on firing rates and spatial firing characteristics. Compared to controls, DG lesions did not alter firing rate, sparsity, or spatial information, but decreased selectivity and yielded longer path lengths during trains (Supplementary Fig. S3b). On balance, there is not a major overall change in excitation relative to inhibition, but if there are any effects, they are generally in the direction of broader fields. The results are, therefore, not consistent with the notion of a merely reduced excitation, such that only the late (i.e., higher rate)

portion of the spatial field is observed in the lesion condition. Performing the same analyses for MEC lesions compared to controls yielded lower selectivity and spatial information, longer path lengths during trains, and increased sparsity (Supplementary Fig. S3c). Again, the less precise spatial firing is inconsistent with the notion that only a portion of the firing field is retained. However, the minor decrease in firing rate with MEC lesions suggests that loss of excitation may be more prominent with MEC lesions compared to DG lesions.

In addition to examining the changes in the spatial firing patterns and firing rates with the lesions, we also examined whether LFP oscillations were changed by either of the two lesions (Supplementary Fig. S4). We did not observe any major effects of either lesion on the power of delta, theta, and slow gamma oscillations, but the power of fast gamma oscillations was increased with the MEC lesions, and theta phase-fast gamma amplitude co-modulation was markedly reduced with DG lesions. In addition, the lesions resulted in a minor decrease in theta oscillation frequency [DG control vs. lesion: $7.92 \pm 0.21$ Hz and $7.28 \pm 0.56$ Hz, median ± interquartile range (iqr), $n = 7$ and 11 sessions, rank sum $= 103$, $p = 2.5 \times 10^{-4}$; MEC control vs. lesion: $7.32 \pm 0.40$ Hz and $7.07 \pm 0.84$ Hz, $n = 13$ and 20 sessions, z-statistic $= 2.23$, $p = 0.026$, Mann-Whitney (MW) tests], and MEC lesions in a loss of speed modulation of theta frequency (MEC control and lesion: $r = 0.238$, $p = 1.1 \times 10^{-5}$ and $r = 0.038$, $p = 0.35$, linear regression). However, the theta oscillation frequencies after DG and MEC lesions were indistinguishable (DG vs. MEC lesion, z-statistic $= 1.09$, $p = 0.27$, MW test) such that any differences in the precise timing of CA3 cells between the lesion groups cannot be attributed to a difference in LFP patterns. We also examined the symmetry of the wave shape of theta oscillations because notable asymmetry has been reported in CA1[56]. In our CA3 LFP recordings, we did not find major asymmetry of theta waves, and there was no added asymmetry with either DG or MEC lesions (Supplementary Fig. S5). The shape of theta waves is, therefore, not a source of bias for any of the phase measurements.

## DG granule cell input was necessary for the full expression of phase precession in CA3 neurons

Although it has long been known that there is substantial phase precession in DG cells[46], it is not clear whether DG inputs are necessary for phase precession in its direct target cells in CA3. We, therefore, compared phase precession in CA3 cells between DG-lesion and control rats, which were trained to perform a dentate-dependent radial 8-arm maze WM task (Supplementary Table S1)[12]. We began by determining the level of phase precession in each control CA3 cell by plotting the theta phase of all spikes in a cell's qualifying trains against the distance covered during the train. The slope was then calculated for each cell ("slope-by-cell" analysis), and phase precession was evident in the negative circular-linear regression slopes (Fig. 1a). We then performed the corresponding analyses in DG-lesion rats. In rats with DG lesions (Supplementary Fig. S1), substantial loss of mossy fiber innervation was previously confirmed for all recording sites that are included in the analysis, and the extent of DG granule cell loss was previously quantified by scoring the remaining mossy fiber density at CA3 recording sites[12]. Here, we combined CA3 cells from all recordings at sites with complete or partial mossy fiber loss (see "Methods" for a detailed description). In CA3 cells recorded at these sites, phase precession was less pronounced and more variable than in controls (Fig. 1b). The median circular-linear regression slope value in the slope-by-cell analysis was $-149.4°$ for the control CA3 cells and only $-79.2°$ for CA3 cells from the DG-lesion animals. Both medians were significantly negative (Fig. 1c; control: $n = 84$ cells, z-statistic $= -6.19$, signed rank $= 396$, $p = 2.96 \times 10^{-10}$; DG-lesion: $n = 68$ cells, z-statistic $= -2.63$, signed rank $= 742$, $p = 0.0043$, one-sided sign tests) though with a reduced slope of phase precession in the DG-lesion compared to the control group (Fig. 1c; z-statistic $= -3.63$, $p = 2.84 \times 10^{-4}$, MW test). When considering the fraction of CA3 cells with negative compared to positive

slopes, a lower proportion displayed negative slopes in lesion compared to control rats (88.1% in CTRL[DG] vs. 69.1% in LESION[DG]; $\chi^2$ test for proportions, $\chi^2 = 8.34$, $p = 0.0039$). When adding the further condition that the slopes had to be not just negative, but also pass a significance criterion, the proportion of cells with significantly negative slopes also differed (Fig. 1c, shaded bars; 61.9% in CTRL[DG] vs. 38.2% in LESION[DG]; $\chi^2$ test for proportions, $\chi^2 = 8.43$, $p = 0.0037$).

Differences between the control and DG-lesion groups were also apparent from the distribution of slopes obtained from the circular-linear regression analysis of single pass data ("slope-by-train" analysis; Fig. 1d). Here, we used the slopes of individual spike trains and, for statistical comparisons, averaged the slopes of each cell's trains (Fig. 1e). We then compared the cells' averages across groups and found that the median cell-averaged slope was significantly less than zero in control and DG-lesion rats (CTRL[DG]: $n = 84$ cells, z-statistics $= -6.3$, signed rank $= 381$, $p = 1.9 \times 10^{-10}$, LESION[DG]: $n = 68$ cells, z-statistics $= -1.98$, signed rank $= 849$, $p = 0.024$, one-sided sign tests). In addition, the median slope of cells from lesion rats was significantly different from the median slope of control cells (z-statistic $= -4.39$, $p = 1.2 \times 10^{-5}$, MW test). Further, the proportion of CA3 cells with negative mean slopes was higher in cells from control than from lesion rats (Fig. 1e, right; 83.3 % vs. 69.1 %, $\chi^2 = 4.3$, $p = 0.038$, $\chi^2$ test of proportions). Taken together, these analyses demonstrate that CA3 phase precession is diminished when the dentate granule cell input to CA3 neurons is reduced. In particular, the analyses with single-train slopes revealed that the remaining inputs to CA3 after DG lesions yield less reliable single-train phase precession.

## MEC inputs to CA3 were also necessary for the expression of phase precession in CA3 neurons

The MEC is known to be necessary for CA1 phase precession[54]. However, it is not known whether CA3 also requires MEC input to generate phase precession or can generate phase precession with DG connectivity alone. Thus, we next tested whether DG alone can support CA3 phase precession by analyzing recordings of CA3 cells in MEC-lesion rats. The MEC lesions were consistent between rats and included 93.0% of the total volume, with damage approximately matched across cell layers (95.3% of layer II, 92.4% of layer III, and 91.4% of deep layers; Supplementary Fig. S1c)[55]. We extracted qualifying spike trains recorded in CA3 of MEC-lesion animals as described above. The slope-by-cell analysis revealed that the CA3 cells of control rats displayed phase precession (Fig. 2a) and that precession was reduced, but not abolished in MEC-lesion animals (Fig. 2b, c; CTRL[MEC]: $n = 101$ cells, median slope: $-120.4°$; LESION[MEC]: $n = 158$ cells,$- 66.9°$; control vs. lesion: z-statistic $= -2.34$, $p = 0.0193$, MW test; control less than zero: z-statistic $= -6.18$, signed rank $= 750$, $p = 3.16 \times 10^{-10}$; lesion group less than zero: z-statistic $= -3.6$, signed rank $= 4204$, $p = 1.57 \times 10^{-4}$, one-sided sign tests). The proportion of cells with negative slopes was lower in the MEC-lesion rats when all cells with negative slopes (83.2% and 70.3%, CTRL[MEC] vs. LESION[MEC], $\chi^2 = 5.52$, $p = 0.0188$) and when only cells with significantly negative slopes were considered (54.5% in CTRL[MEC] vs. 38.6% in LESION[MEC], $\chi^2 = 6.26$, $p = 0.0124$, $\chi^2$ tests for proportions). As with DG lesions, the slope-by-train analysis showed that reliable negative single-train slopes were seen for trains from control CA3 cells but less for trains from CA3 cells of MEC-lesion rats (Fig. 2d, e; medians less than zero: CTRL[MEC], z-statistic $= -6.9$, signed rank $= 550$, $p = 3.4 \times 10^{-12}$; LESION[MEC], z-statistic $= -2.9$, signed rank $= 4628$, $p = 0.0021$, one-sided sign tests; Median of CTRL[MEC] vs. LESION[MEC]: z-statistic $= -3.9$, $p = 9.3 \times 10^{-5}$, MW test; proportions of negative slopes: 84.2% and 65.8%, CTRL[MEC] vs. LESION[MEC], $\chi^2 = 10.5$, $p = 0.0012$, $\chi^2$ test). These observations support a role for MEC in the generation of robust phase precession in the CA3 of rats. Therefore, the DG-CA3 network alone is incapable of generating phase precession at control levels−for this, both the DG and MEC inputs are necessary.

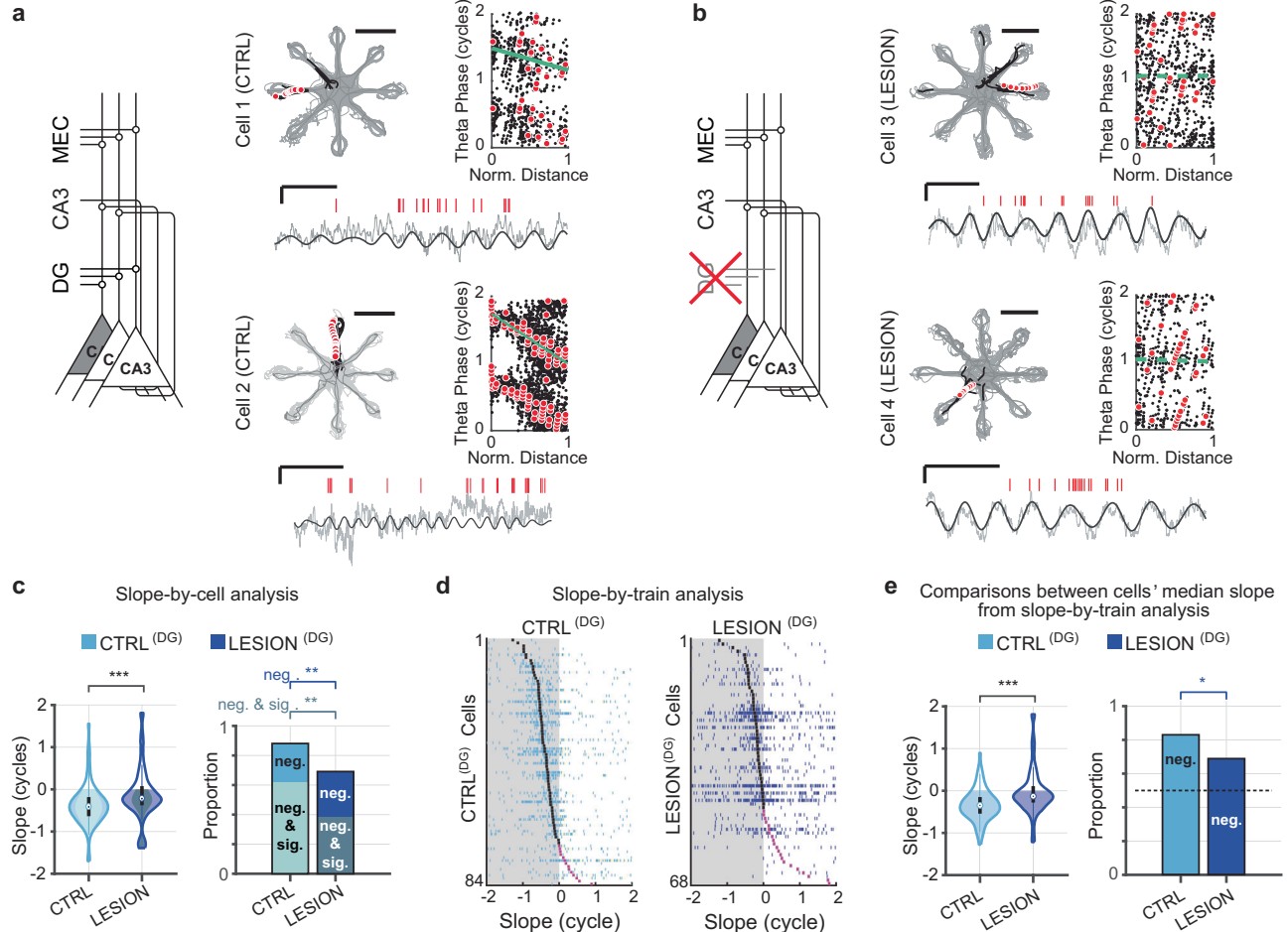

**Fig. 1 | Dentate granule cell input is required for intact phase precession in CA3.** **a** Intact phase precession in control CA3 cells. Left, schematic of the major theta-modulated excitatory inputs to CA3. Right, Two example CA3 cells and their spatial firing patterns (gray lines, path; black dots, spike locations; red dots, spike locations of an example spike train; scale bar, 50 cm), an example spike train and corresponding LFP trace (red ticks, spikes; solid line, 6–10 Hz filtered LFP; gray line, raw LFP; scale bars, 250 ms and 500 μV), and phase-versus-normalized distance plot (black dots, spikes; red dots, spikes of example train, 1 cycle = 360°). Solid green lines in the plot indicate significant phase precession ($p < 0.05$; circular-linear regressions). **b** Data as in (**a**) but for CA3 cells without DG inputs. Dashed green lines indicate the lack of phase precession ($p > 0.05$; circular-linear regressions; LFP scale bars, 250 ms and 500 μV; Path scale bar, 50 cm). **c** Phase precession slopes and proportions of phase precessing cells. One slope per cell was obtained by pooling the spikes of all trains and by fitting a circular-linear regression to this pool (slope-by-cell analysis). The median magnitude of the slopes (violin plots; $n = 84$ control (CTRL) and 68 DG lesion (LESION) CA3 cells, z-statistic = $-3.62$, $p = 2.8 \times 10^{-4}$, MW

test) and the proportion of negative slopes (bar plots; only negative slopes, $\chi^2 = 8.34$, $p = 0.0039$; negative and significant slopes, $\chi^2 = 8.43$, $p = 0.0037$, chi-square test) were reduced by the DG lesion. **d** Slopes for all trains from control (CTRL(DG), left) and DG-lesion (LESION(DG), right) CA3 cells (slope-by-train analysis). Each row depicts the slope values from each of the trains of one cell (blue ticks), and cells are sorted from top to bottom by their trains' median slope (black tick when negative, purple tick when positive). Shaded regions correspond to negative values. **e** Phase precession slopes (violin plots; $n = 84$ control and 68 DG lesion CA3 cells, z-statistic = $-4.39$, $p = 1.2 \times 10^{-5}$, MW test) and proportions of phase precessing cells (bar plots; $\chi^2 = 4.28$, $p = 0.038$, chi-square test) from the slope-by-train analysis. For analysis of proportions, a cell was considered phase precessing if the median slope was negative. Violin plots in panels (**c**, **e**): Outline, distribution; shading, negative slopes (inner shading in **c**, negative and significant slopes); error bars, 1.5 times the interquartile interval above the third and below the first quartile. *$p < 0.05$,**$p < 0.01$, ***$p < 0.001$. Source data are provided as a Source Data file.

## Putative granule cells exhibited a narrow theta phase preference at the onset of spiking

To next ask whether the temporal profile of DG granule cell spiking is precise enough to organize the spiking phase of CA3 neurons, we analyzed neuronal activity from rats in which we were able to record single units from the DG ($n = 5$ cells, see Methods). Putative granule cells showed phase precession (Supplementary Fig. S6a, b), which was accompanied by a strikingly narrow theta phase preference at the onset of spiking. The phase preference then broadened over the course of the spike train. Putative mossy cells and CA3 pyramidal cells, although phase precessing, showed a relatively broad theta phase variability throughout the entire spike train (Supplementary Fig. S6c). These findings are suggestive of a particularly critical role of DG granule cells in providing

temporal information to CA3 pyramidal cells upon entering the place field.

## DG and MEC lesions had qualitatively distinct effects on CA3 phase precession

After confirming that both the DG and MEC were necessary for CA3 phase precession at control levels, we asked whether there were qualitative differences in the phase precession patterns when each of these inputs were diminished. Phase precession can be reduced by either limiting the theta phase range over which spiking occurs or by heightening the variability around a monotonically decreasing precession slope, or both. To determine whether the theta phase range was altered by the lesions, we calculated the onset and offset theta phase of CA3 spike trains. The onset phase of trains – defined by first

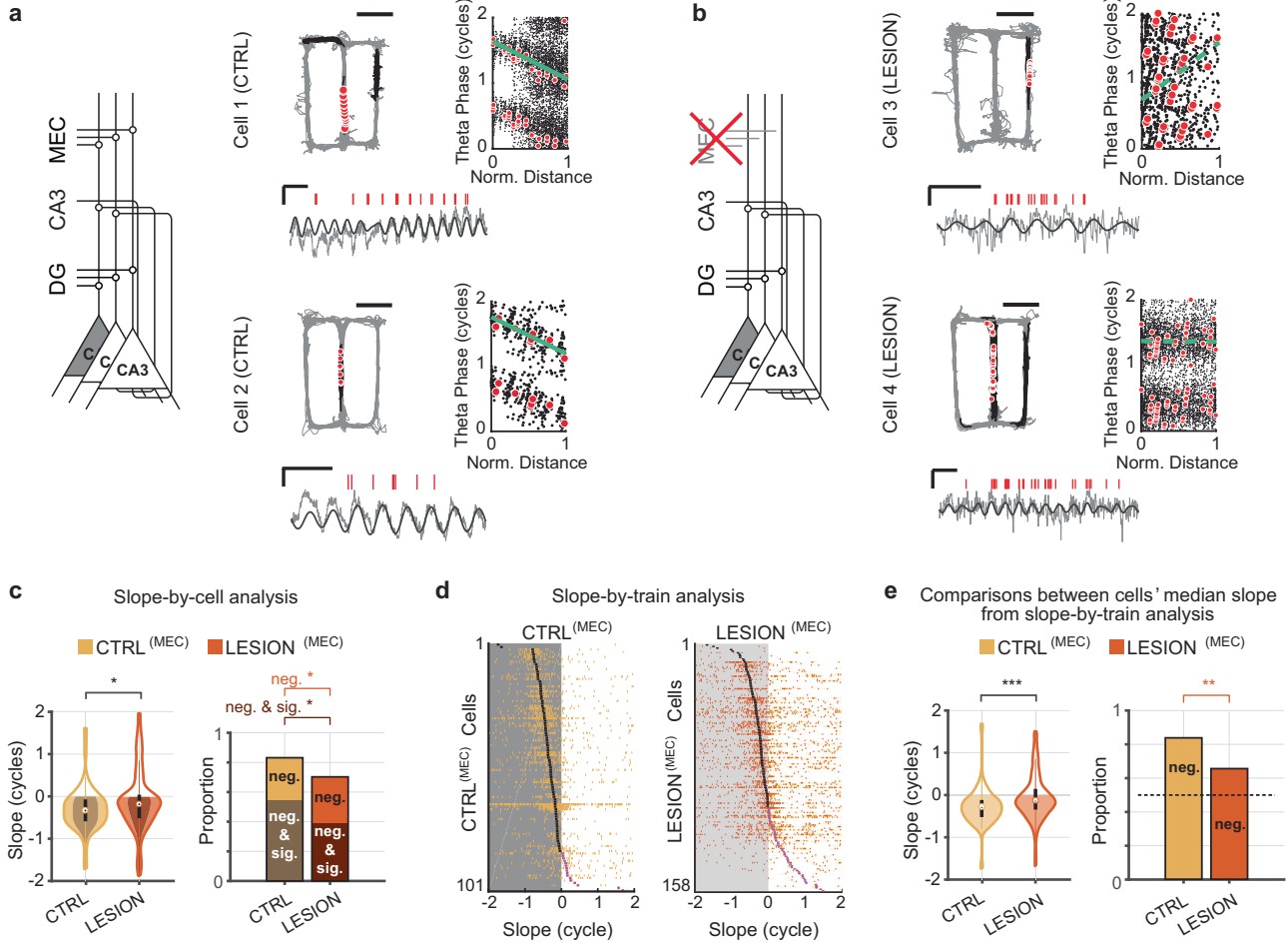

**Fig. 2 | Medial entorhinal cortical input is required for intact phase precession in CA3.** This figure follows the presentation of Fig. 1 but with data from CA3 cells with lesioned MEC inputs and respective controls. **a, b** Firing patterns of example CA3 cells in control and MEC-lesion rats. Solid green lines in the phase-versus-normalized distance plot indicate significant phase precession ($p < 0.05$; circular-linear regressions) and dashed green lines indicate a lack of significant phase precession ($p > 0.05$; circular-linear regressions). Scale bars for LFP: 500 µV and 250 ms, for path: 50 cm. **c** Phase precession slopes (violin plots; $n = 101$ control (CTRL) and 158 MEC lesion (LESION) CA3 cells, z-statistic = −2.34, $p = 0.019$, MW test) and proportions of phase precessing cells (bar plots; controls vs. lesion, only negative slopes, $\chi^2 = 5.52$, $p = 0.019$; negative and significant, $\chi^2 = 6.26$, $p = 0.0124$, chi-square tests) from the slope-by-cell analysis. The magnitude of the slopes and

the proportion of negative slopes were reduced by the MEC lesions. **d** Slopes for all trains from CTRL(MEC) (left) and LESION(MEC) (right) CA3 cells. Each row depicts the slope values from each of the trains of one cell (yellow and orange ticks), and cells are sorted from top to bottom by their trains' median slope (black tick when negative, purple tick when positive). **e** Phase precession slopes (violin plots; $n = 101$ control and 158 MEC lesion CA3 cells, z-statistic = −3.91, $p = 9.3 \times 10^{-5}$, MW test) and proportions of phase precessing cells (bar plots; $\chi^2 = 10.50$, $p = 1.2 \times 10^{-3}$; chi-square test) from the slope-by-train analysis. Violin plots in panels (**c, e**): Outline, distribution; shading, negative slopes (inner shading in c, negative and significant slopes); error bars, 1.5 times the interquartile interval above the third and below the first quartile. * $p < 0.05$, ** $p < 0.01$, *** $p < 0.001$. Source data are provided as a Source Data file.

calculating the circular mean of first-cycle spikes of each train and by then taking the median over all of the cell's trains – no longer consistently occurred at late phases in CA3 cells of DG-lesion rats (Fig. 3a, b). For control cells, there was a clear peak in the distribution of onset phases ($\Phi_{on}$) during the late phase of the theta cycle, past the trough, and accordingly, the circular mean of all onset phases was 228.4°. For cells from DG-lesion rats, onset phases had approximately the same circular mean (226.7°), but strikingly, were broadly distributed over the theta cycle ($n = 84$ and 68 cells for control and lesion, $\chi^2 = 24.4$, $p = 5.0 \times 10^{-6}$, circular MANOVA; phase concentration parameters: CTRL(DG) $\kappa = 1.91$, LESION(DG) $\kappa = 0.42$, U = 25.8, $p = 3.7 \times 10^{-7}$, concentration test). As expected for phase precessing cells, the distributions of offset phases ($\Phi_{off}$) – defined as the circular median phase of the spikes in the last cycle of each train – were earlier in the theta cycle for cells from control and DG-lesion rats (mean offset phases, 79.6°, and 53.1°). In contrast to the onset phases, the offset phases did not show differences in their distributions or concentrations between

cells from DG-lesion rats compared to control cells (Fig. 3b; $\chi^2 = 3.25$, $p = 0.20$, circular MANOVA; concentration: CTRL(DG) $\kappa = 0.80$, LESION(DG) $\kappa = 0.95$, U = 0.32, $p = 0.57$, concentration test). Selective effects on the timing of the onset phases, but not of the offset phases were further confirmed by measuring the dispersion of the phases of the first spikes and of the phases of the last spikes across spike trains of each cell. With DG lesions, the dispersion of the first spikes, but not of the last spikes increased (Fig. 3c; first: z-statistic = −4.11, $p = 3.9 \times 10^{-5}$; last: z-statistic = −1.33, $p = 0.18$, MW tests).

With MEC lesions, effects on the CA3 cells' median onset and offset phases were not observed (Fig. 3d and e; onset phases: CTRL(DG) 246.1°, LESION(DG) 238.7°, $n = 101$ and 158 cells for control and lesion, $\chi^2 = 1.18$, $p = 0.56$, circular MANOVA; onset phase concentration: CTRL(MEC) $\kappa = 1.49$, LESION(MEC) $\kappa = 1.33$, U = 0.44, $p = 0.51$, concentration test; offset phases: CTRL(MEC): 77.0° LESION(MEC): 90.5°, $\chi^2 = 2.18$, $p = 0.34$, circular MANOVA; offset phase concentration: CTRL(MEC) $\kappa = 1.08$, LESION(MEC) $\kappa = 1.16$, U = 0.15, $p = 0.70$, concentration test).

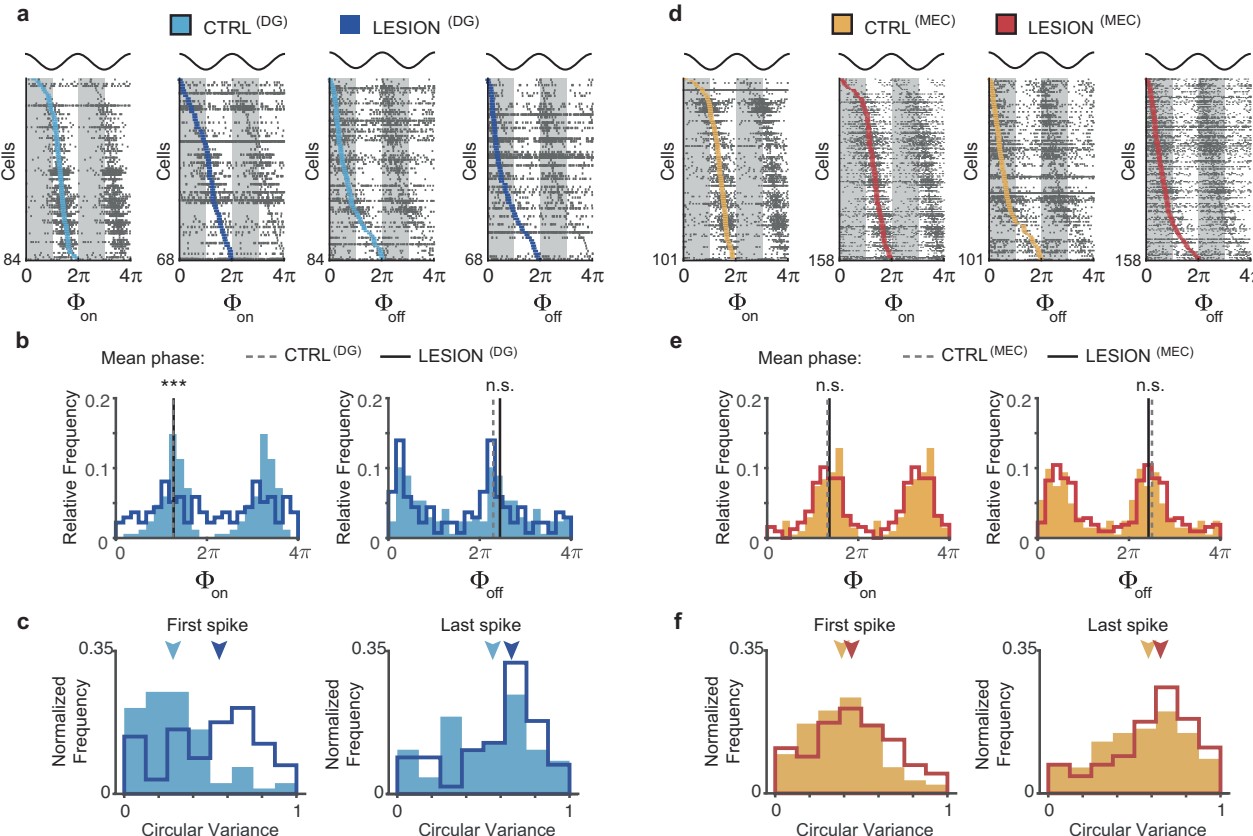

**Fig. 3 | DG lesions, but not MEC lesions substantially broaden the onset phase of CA3 spike trains. a** Onset and offset phases of spike trains from CA3 cells in control (CTRL(DG)) and DG-lesion rats (LESION(DG)). In each of the four raster plots, each row displays the onset phase ($\Phi_{on}$) or offset phase ($\Phi_{off}$) of the trains of a cell (in gray), and the cell's median onset or offset phase (light blue, control; dark blue, DG-lesion). Cells within each panel are sorted from top to bottom by their median onset/offset phase. Data are repeated from $2\pi$ to $4\pi$ for clarity, and two LFP theta cycles are displayed on top for reference. **b** The cells' median onset and offset phases (shown in blue in panel **a**) were compared between control and DG-lesion rats. Data are repeated from $2\pi$ to $4\pi$. Note the greater concentration of onset phases in the ascending (i.e., late) portion of the theta cycle in cells from control compared to DG-lesion rats. The distribution of onset phases differed between CTRL(DG) (light blue bars; $n = 84$ cells) and LESION(DG) groups (dark blue line; $n = 68$ cells, $\chi^2 = 24.4$, $p = 5.0 \times 10^{-6}$, circular MANOVA) with a higher concentration in the control group (phase concentration: CTRL(DG) $\kappa = 1.91$, LESION(DG) $\kappa = 0.42$, $U = 25.8$, $p = 3.7 \times 10^{-7}$, concentration test). Offset phases were not altered by the lesion ($\chi^2 = 3.25$, $p = 0.20$, circular MANOVA; phase concentration: CTRL(DG) $\kappa = 0.80$, LESION(DG) $\kappa = 0.95$, $U = 0.32$, $p = 0.57$, concentration test). **c** The dispersion of the

phase of the first and last spikes was calculated across spike trains of each cell, and the cells' dispersions were compared between control and DG-lesion rats. Dispersions of the cells' onset phase but not of the cells' offset phase increased with DG lesions (onset: z-statistic $= -4.11$, $p = 3.9 \times 10^{-5}$; offset: z-statistic $= -1.33$, $p = 0.18$, MW tests). The arrowheads mark the median circular variance values (control, light blue; lesion, dark blue). **d–f**, As (**a–c**), but for the cells from MEC-lesion rats (LESION(MEC)) and their corresponding controls (CTRL(MEC)). Onset phase values again peaked in the ascending phase of the theta cycle, but their distribution was not altered by the lesion ($n = 101$ CTRL(MEC) and 158 LESION(MEC) cells, $\chi^2 = 1.18$, $p = 0.56$, circular MANOVA; phase concentration: CTRL(MEC) $\kappa = 1.49$, LESION(MEC) $\kappa = 1.33$, $U = 0.44$, $p = 0.51$, concentration test). Similarly, the distribution of offset phases did not differ between cells of MEC-lesion and control rats ($\chi^2 = 2.18$, $p = 0.34$, circular MANOVA; concentration: CTRL(MEC) $\kappa = 1.08$, LESION(MEC) $\kappa = 1.16$, $U = 0.15$, $p = 0.70$, concentration test). Dispersions of onset and offset phases were moderately broadened by MEC lesions (onset: z-statistic $= -2.30$, $p = 0.021$; offset: z-statistic $= -2.16$, $p = 0.031$, MW tests), which is consistent with an overall increase in variability in theta phase preference. n.s., not significant, *** $p < 0.001$. Source data are provided as a Source Data file.

Furthermore, effects on the dispersions of first and last spikes across spike trains were moderate and approximately matched (Fig. 3f; first: z-statistic $= -2.30$, $p = 0.021$; last: z-statistic $= -2.16$, $p = 0.031$, MW tests), which is consistent with an overall increase in variability in theta phase preference. Taken together, the selective effects on the onset, but not offset phase with DG, but not with MEC lesions indicate that it is predominantly the DG input rather than the MEC input to CA3 that is involved in setting the narrow, late-onset theta phase of CA3 spikes. These results were also confirmed when repeating the same analyses by using the cells' place fields rather than the cells' spike trains (Supplementary Fig. S7), which confirms that qualitative differences in the effect of DG and MEC lesions on temporal firing patterns in CA3 cannot be attributed to methodological details.

The finding that DG lesion effects on the onset phase are coupled with a broadening of the phase dispersion can be interpreted as the emergence of "noise" spikes in early theta phases which would increase

the phase variance and shift the mean to earlier phases upon entry to the place field. To further examine this possibility, we analyzed several measurements of spike phase distribution. When measuring spike phase distribution across all theta cycles regardless of the distance traveled by the rat, the CA3 spike phase in control cells was concentrated in the middle of the theta cycle, as expected (Fig. 4a)[57,58]. Spike phase distributions across theta cycles shifted towards earlier phases of the theta cycle with DG lesions while the shift was in the opposite direction with MEC lesions compared to their respective controls (Fig. 4a; mean theta phase: CTRL(DG) $n = 84$ cells, mean phase $= 173.4°$, LESION(DG) $n = 68$ cells, mean phase $= 74.1°$, $\chi^2$ statistic $= 18.4$, $p = 0.0001$; CTRL(MEC) $n = 101$ cells, mean phase $= 157.1°$, LESION(MEC) $n = 158$ cells, mean phase $= 172.3°$, $\chi^2$ statistic $= 0.74$, $p = 0.69$; $\chi^2$ test for proportion of spikes contained in each of the three theta bins). The shift to earlier phases with DG lesions was accompanied by an increase in the proportion of spikes in the first third of the

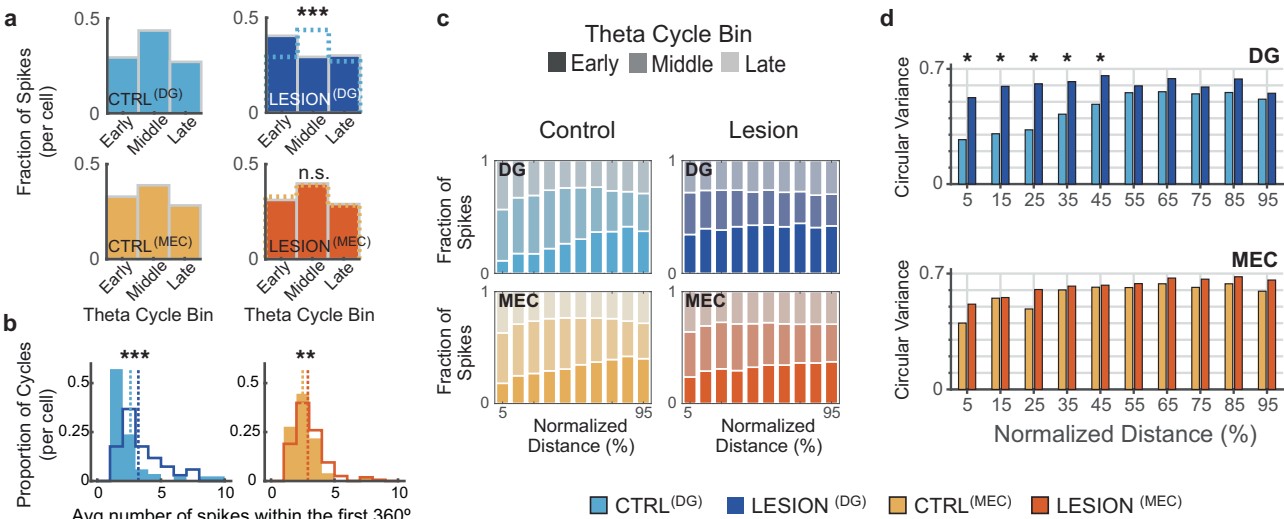

**Fig. 4 | With reduced DG input, peak probability of CA3 spiking shifts to earlier theta phases. a** Left, For control CA3 cells, distribution of spike incidence across early, middle, and late theta phases. Top right, The proportion of CA3 spikes early in the theta cycle was markedly increased with DG lesions (mean theta phase: CTRL(DG) $n = 84$ cells, mean phase = 173.4°, LESION(DG) $n = 68$ cells, mean phase = 74.1°, $\chi^2$ statistic = 18.4, $p = 0.0001$, $\chi^2$ test for proportion of spikes in each of the three thirds). Dotted lines, data from control CA3 cells, as shown to the left. Bottom right, with MEC lesions (LESION(MEC)), there was no change in the phase distribution of CA3 spikes (CTRL(MEC) $n = 101$ cells, mean phase = 157.1°, LESION(MEC) $n = 158$ cells, mean phase = 172.3°, $\chi^2$ statistic = 0.74, $p = 0.69$; $\chi^2$ test for proportion). **b** Average number of spikes over a 360°-cycle from the first spike in the train. In trains from cells of DG-lesion rats, the median number of initial-cycle spikes increased by 23.8% compared to control cells (from 2.61 to 3.23, $n = 84$ CTRL(DG) and 68 LESION(DG) cells, z-statistic = − 4.38, $p = 1.2 \times 10^{-5}$, MW test). In MEC-lesion rats, the median number of initial-cycle spikes increased by 16.5% compared to control trains (from 2.49 to

2.9 spikes, $n = 101$ CTRL(MEC) and 158 LESION(MEC) cells, z-statistic = − 2.74, $p = 0.006$, MW test). **c** Fraction of CA3 spikes at early, middle, and late phases of the theta cycle as a function of normalized (%) distance during the train. At the onset of control trains, a low proportion of spikes is typically observed at early phases, but this proportion increases with DG lesions. **d** Top, DG lesions selectively increased the variability of spike phase in the first half of a spike train (from 5% to 45% of the normalized distance, MW tests; Holm-Bonferroni corrected; see Supplementary Table S2 for statistics). See Fig. 1b, for example spike trains that show this effect. Bottom, MEC lesions did not increase the variability of theta phase along the distance through the field (MW tests; Holm-Bonferroni corrected; Supplementary Table S2). * $p < 0.05$, ** $p < 0.01$, *** $p < 0.001$. For most theta cycle-based analyses, similar results were obtained when repeating the analyses by using the cells' place fields rather than the cells' spike trains (Supplementary Fig. S7ii). Source data are provided as a Source Data file.

theta cycle in DG-lesion rats, which could result either from the addition of spurious spikes compared to controls or solely from redistributing the same number of spikes. To test whether additional spikes were present in early theta cycles in DG-lesion rats, we counted the number of spikes in the first 360° cycle after the first spike and for each neuron, averaged the number of spikes over all spike trains. The number of initial-cycle spikes increased by 23.8% in trains from cells of DG-lesion compared to control rats (from 2.61 to 3.23, $n = 84$ CTRL(DG) and 68 LESION(DG) cells, z-statistic = − 4.38, $p = 1.2 \times 10^{-5}$, MW test). Similarly, the median number of initial-cycle spikes increased by 16.5% in spike trains of cells from MEC-lesion compared to control rats (from 2.49 to 2.9 spikes, $n = 101$ CTRL(MEC) and 158 LESION(MEC) cells, z-statistic = − 2.74, $p = 0.006$, MW test; Fig. 4b). Both lesion groups therefore showed disinhibition at the onset of the spike train, but this was accompanied by more early-phase spikes in only cells from DG-lesion rats (see Figs. 3c, 4a).

In additional analyses that consider the distance traveled during a spike train, we binned the normalized distance during each spike train into 10 bins and considered the joint distribution of spike phase and normalized distance (Fig. 4c). Here, we found that DG lesions resulted in a particularly pronounced redistribution of spikes to earlier theta phases in early bins. This redistribution reduced the proportion of late-phase spikes that are normally observed in the early section of the path such that the theta phase when spikes occurred was now remarkably similar irrespective of distance along the path. In addition, the circular variance of CA3 spike theta phase was selectively increased in the first half of the path (Fig. 4d, top; pairwise comparisons between CTRL(DG) and LESION(DG) are significant in the first five of ten bins, Holm-Bonferroni corrected $p < 0.05$, see Supplementary Table S2 for detailed statistics) such that the variance in all bins with DG lesion

reached levels that are in controls only observed in the second half. In contrast, circular variance took on similar values in MEC control and lesion rats (Fig. 4d, bottom; pairwise comparisons between CTRL(MEC) and LESION(MEC), all bins not significant, see Supplementary Table S2 for detailed statistics). The broadening of the onset phase distribution at the onset of the train together with the increase in firing rate in the initial theta cycle (Fig. 4c) is consistent with the possibility that DG to CA3 input is critically involved in restricting the spiking in the beginning of the train to late-theta phase ("prospective") CA3 spiking by inhibiting spurious spikes in the early phases of the theta cycle.

### Analytically reducing phase variability partially recovered phase precession of CA3 cells from MEC-lesion rats

If a lesion impairs phase precession by increasing the variance of the phase distribution, it might be feasible to recover phase precession by analytically reducing the phase variance. Phase variance can be reduced by replacing, within each theta cycle, all spikes with their mean timestamp. However, if the main effect of a lesion is a reduction in slope or phase range, replacing the spikes with their cycle mean should not restore phase precession. When we replaced the spikes of each theta cycle with their mean phase within the cycle ('cycle-mean analysis'), the effect of the lesion could be largely recovered in MEC-lesion rats, but not in DG-lesion rats. The rescue of the MEC lesion was observed both when all trains were combined as well as when individual trains were analyzed separately (Supplementary Fig. S8). With the cycle-mean analysis, the mean of the cells' slopes in MEC-lesion rats was no longer distinguishable from control rats (CTRL(MEC): $n = 101$ cells, − 97.4° per traversal vs. LESION(MEC): $n = 158$ cells, − 84.6° per traversal, z-statistic = − 1.44, $p = 0.151$, MW test), while it remained different from controls in DG-lesion rats

(CTRL(DG): $n = 84$ cells, $-96.6°$ per traversal vs. LESION(DG): $n = 68$ cells, $-5.6°$ per traversal, z-statistic $= -4.23$, $p = 2.4 \times 10^{-5}$, MW test) (Fig. 5a–c). In all cases, however, the mean slope was less than zero for each of the groups (CTRL(DG): $n = 84$ cells, z-statistic $= -6.49$, $p = 4.4 \times 10^{-11}$, LESION(DG): $n = 68$ cells, z-statistic $= -2.33$, $p = 0.01$; CTRL(MEC): $n = 101$ cells, z-statistic $= -7.42$, $p = 5.9 \times 10^{-14}$, LESION(MEC): $n = 158$ cells, z-statistic $= -7.56$, $p = 2 \times 10^{-14}$; one-sided sign tests), as without cycle averaging (Fig. 5c; compare with Figs. 1 and 2). We also tested to what extent the calculation of the cycle mean reduced phase variability compared to the original spike trains. In DG-lesion rats, there was only a minor difference in the variance of the phase probability over all trains of each neuron between the original and cycle-mean analyses (circular variance $= 0.733$, cycle-mean circular variance $= 0.661$, z-statistic $= 2.04$, $p = 0.041$, MW test; Cohen's $d = -0.308$). However, in the two control groups and in the MEC-lesion group variance was substantially reduced by taking the cycle mean (CTRL(DG): circular variance $= 0.707$, cycle-mean circular variance $= 0.557$; CTRL(MEC): circular variance $= 0.782$, cycle-mean circular variance $= 0.603$; LESION(MEC): circular variance $= 0.805$, cycle-mean circular variance $= 0.634$; tests of difference between original and cycle-mean variances: all $p$-values $< 10^{-6}$, MW tests; Fig. 5d: compare, for each panel, the difference of medians indicated by the solid and dashed vertical lines; Cohen's $d$ for CTRL(DG), CTRL(MEC), and LESION(MEC), respectively: $-0.809$, $-0.917$, $-0.779$). Taken together, these results indicate that increased within-cycle variability makes a major contribution to diminished phase precession in cells from MEC-lesion animals. In contrast, increased within-cycle variability was not identified as the key source for reducing phase precession in DG-lesion animals.

## Theta-scale temporal correlations of CA3 cells were preserved with MEC lesions but not with DG lesions

Phase precession is associated with the occurrence of ordered neuronal firing patterns within each theta cycle[37] such that sequential firing within a theta cycle ('theta sequence') corresponds to the order in which place fields are traversed, but with the timing in the theta cycle compressed compared to the behavioral time scale. For example, two adjacent place cells are activated one after another within milliseconds in a theta cycle, while the rise and fall in firing rates when traversing the fields occurs on a much slower time scale. Phase precession is thought to link the slower behavioral sequence to the faster pairwise temporal correlation in theta cycles. However, dissociations between theta sequences and phase precession have been reported. Phase precession without theta sequences can be observed in a novel environment and without CA3 inputs to CA1[41,59], and theta sequences have been observed with impaired phase precession[60]. Given that theta phase precession and theta sequences can be dissociated, we asked whether the precise pairwise timing at the theta scale was diminished when phase precession was impaired by the reduction of either DG or MEC inputs to CA3. To measure whether time-compressed ordering within the theta cycle occurred, we measured whether there is a relation between the timing on the theta scale (i.e., the temporal difference in the spike cross-correlation of cell pairs) and the behavioral scale (i.e., the distance between the peak firing locations of place fields; see Methods)[37].

While there was a strong correlation between the spatial separation of place fields and the theta phase difference of cell pairs in controls (CTRL(DG): $n = 30$ pairs, $r = 0.651$, $p = 9.87 \times 10^{-5}$, Pearson's correlation), we found that the behavioral order of firing was not reflected on the theta-cycle time scale in cell pairs from DG-lesion rats (Fig. 6a–d; LESION(DG): $n = 14$ pairs, $r = -0.187$, $p = 0.523$, Pearson's correlation). This effect was observed even though the proportion of neuron pairs with phase precession in DG-lesion rats was comparable to that in control rats (Fig. 6d, right). Contrary to the result with DG lesions, the CA3 pairs in the MEC-lesion rats maintained their spiking

order in theta cycles compared to their firing order on the maze (Fig. 6e–h; CTRL(MEC) $n = 19$ pairs, $r = 0.654$, $p = 0.0024$; LESION(MEC) $n = 27$ pairs, $r = 0.624$, $p = 5.02 \times 10^{-4}$; Pearson's correlations), despite reduced phase precession in comparison to neuron pairs from control rats (Fig. 6h, right) and different from the loss of ordering that has been reported in CA1[54]. Thus, it seems that the MEC is not critical in maintaining theta-scale spike ordering when there are DG inputs that are sufficient for organizing temporal relationships of CA3 spiking within theta cycles.

## A phenomenological computational model of phase precession in CA3 cells revealed distinct effects of DG and MEC inputs on the inhibitory signal

To gain a further mechanistic understanding of whether and how the effects on phase precession observed in lesion animals can arise from single-cell integration of the two excitatory theta-modulated inputs to CA3, we devised a minimal phenomenological model based on oscillatory interference. We chose to minimize the number of free parameters (see below) of the model to be able to quantitatively fit simulations of the spiking dynamics to the phase precession statistics observed in the data sets. Although our analyses of experimental data had to be limited to the two excitatory inputs to CA3 that were manipulated in lesion experiments, we reasoned that if a computational model based on oscillatory interference were to emulate the lesions, it must account for inputs beyond the manipulated inputs. Inhibition has been shown to mediate input gain control, precise spike timing, and enhanced coding in networks[61–64] and can thus be considered essential for controlling the theta phase of pyramidal cell spikes[49]. Therefore, we included an inhibitory oscillation in the model that can be viewed as corresponding to the observed oscillations of a large fraction of hippocampal interneurons at the LFP theta frequency[65,66]. Based on recordings from DG and MEC principal cells that are known to project to CA3, the excitatory inputs from each of these two regions were considered to oscillate at frequencies slightly above the LFP theta frequency[67,68]. In addition, the relative contributions of DG and MEC inputs varied along the place field to reflect the proposal that entorhinal inputs provide sensory cues at the true place field location while intrahippocampal circuits govern the prospective spiking[30,69–71]. We allowed the model to have four free parameters: a phase shift between the two excitatory inputs denoted by $\psi$, a phase shift between excitation and inhibition denoted by $\varphi_{inh}$, the oscillatory amplitude of inhibition denoted by $A$, and a DC component for the inhibitory oscillation (baseline inhibition) denoted by $I_{DC}$. Although the full range of possible $\psi$ parameters was tested, it is relevant to note that experimental data[68] suggest that neurons with direct MEC inputs to DG (i.e., MEC layer II neurons) and DG inputs to CA3 show activity over $\sim 90°$ ranges of the theta cycle that are approximately overlapping. Even if inputs were to originate from neurons at the extremes of these distributions, the difference in $\psi$ would, therefore, typically not exceed $\pm 90°$.

The output of the model CA3 cell was determined by the place modulated[72,73] combination of the three inputs (two excitatory and one inhibitory) from which a threshold value that was constant across the place field was subtracted. Spikes were generated stochastically via an inhomogeneous Poisson point process with an intensity measure defined by the total excitatory drive minus the threshold. The simulated spike phases were extracted with respect to an 8 Hz oscillation representing the LFP theta oscillation, which was considered to be phase-locked to the inhibitory oscillation (see Methods). The difference angle between intracellular and LFP oscillation was chosen as 180°, since it produced the largest spike rates at the uninhibited phase (e.g., ref. 46). Accordingly, the largest spike rates in the data would occur at the minimum inhibition phase of the model. Variations of LFP phase shift by $\pm 45°$ from 180° did not qualitatively alter the result (Supplementary Fig. S9a).

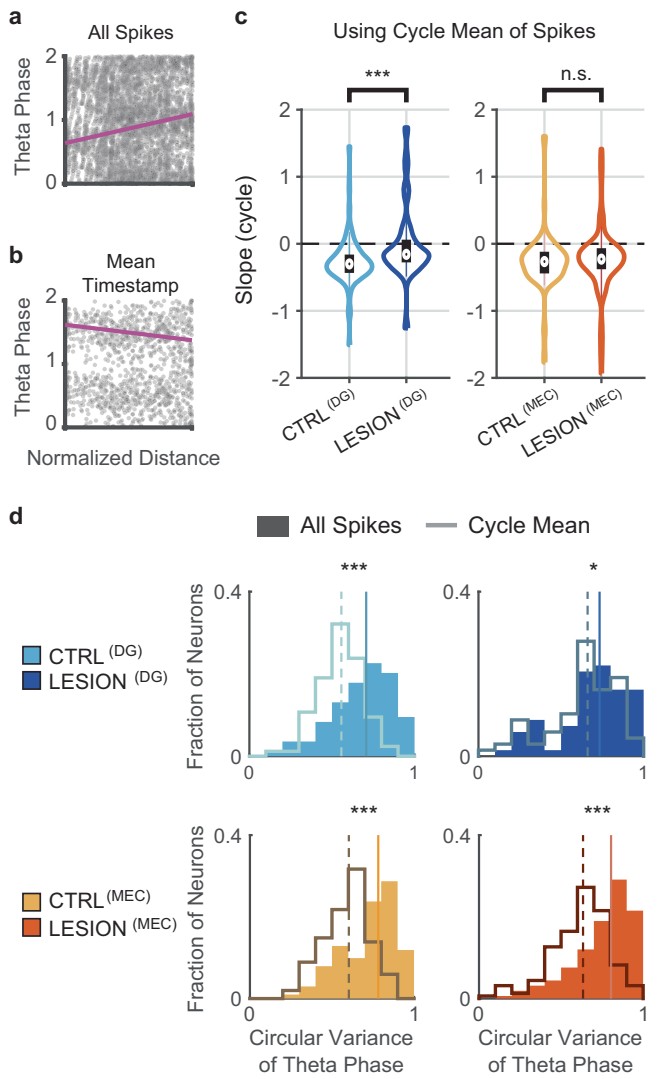

**Fig. 5 | Distinct patterns of temporal reorganization of CA3 spiking after DG and MEC lesions. a** Phase-distance plot of spikes from an example CA3 cell from an MEC-lesion rat. All spikes of the cell's trains are included. **b** Phase-distance plot of the same cell after replacing the spikes within each theta cycle with the mean phase of the spikes within each cycle. Using the cycle mean yielded a negative precession slope. An inconsistent distribution of spikes within theta-cycles after MEC lesions may thus have precluded the detection of phase precession. **c** Distribution (violin plot) of circular-linear regression slopes calculated from the cycle means. With regression slopes from cycle means there was no difference between cells from control and MEC-lesion rats ($n = 101$ and 158 cells, z-statistic $= -1.44$, $p = 0.151$, MW test), while the difference between cells from control and the DG-lesion rats was retained ($n = 84$ and 68 cells, z-statistic $= -4.23$, $p = 2.4 \times 10^{-5}$, MW test). However, as in analyses without within-cycle phase averaging (see Figs. 1 and 2), all groups showed some level of remaining phase precession (median slope less than zero: CTRL$^{(DG)}$: $n = 84$ cells, z-statistic $= -6.49$, $p = 4.4 \times 10^{-11}$, LESION$^{(DG)}$: $n = 68$ cells, z-statistic $= -2.33$, $p = 0.01$; CTRL$^{(MEC)}$: $n = 101$ cells, z-statistic $= -7.42$, $p = 5.9 \times 10^{-14}$, LESION$^{(MEC)}$: $n = 158$ cells, z-statistic $= -7.56$, $p = 2 \times 10^{-14}$; one-sided sign tests). **d** Circular variance of theta phase with either all spikes (filled bars) or with each cycle's spike mean (solid lines). In all groups, the circular variance decreased after replacing each cycle's spikes with their mean, though the effect is least pronounced for CA3 cells of DG-lesion rats. The median circular variance of cycle mean spikes (dashed vertical lines) and all spikes (solid vertical lines), respectively [CTRL$^{(DG)}$, 0.557 and 0.707 (effect size $= -0.809$; z-statistic $= -5.22$, $p = 1.8 \times 10^{-7}$); LESION$^{(DG)}$, 0.661 and 0.733 (effect size $= -0.308$; z-statistic $= -2.04$, $p = 0.041$); CTRL$^{(MEC)}$, 0.603 and 0.782 (effect size $= -0.917$; z-statistic $= -6.16$, $p = 7.5 \times 10^{-10}$); LESION$^{(MEC)}$, 0.634 and 0.805 (effect size $= -0.779$; z-statistic $= -7.26$, $p = 4.0 \times 10^{-13}$; MW tests)]. Violin plots: Outline, distribution; error bars: 1.5 times the interquartile interval above the third and below the first quartile. n.s., not significant, * $p < 0.05$, *** $p < 0.001$. These results were confirmed when repeating the same analyses by using the cells' place fields rather than the cells' spike trains (Supplementary Fig. S7iii). Source data are provided as a Source Data file.

We simulated CA3 model neuron spikes for a broad range of parameter values. We observed for the full model (Fig. 7a) that phase precession can be consistently obtained in single trials and on trial average (Fig. 7b, c) but was less prominent when either of the two excitatory components was removed (averages shown for each lesion in Fig. 7c). To determine how parameter values under this model corresponded to the experimental data from the control and lesion groups, we calculated four phase precession measurements – slope, explained variance R², onset phase, and offset phase – from the model spike data in the full A versus $\varphi_{inh}$ parameter space. We then identified the region of the parameter space in which the model generated phase precession measurements that corresponded to those obtained in our empirical data (i.e., the middle 80th-percentile of the empirical observations in the experiments). This was done separately by matching the control data to the control model and the lesion data to each of the respective lesion models. In the control model, only a limited range of phase shifts ($\varphi_{inh}$) between the inhibitory input and the excitatory inputs generated phase precession measurements that corresponded to data from control animals. When the DG input was set to zero in the model, the model-generated phase precession data matched with the empirical data over a broader and shifted set of $\varphi_{inh}$ values compared to controls (Fig. 7d). The set of parameter A values was also shifted downwards with the lesion, which is expected when decreasing the total amount of excitation by setting one of the excitatory inputs to zero. In contrast, when MEC-lesion empirical data were

matched with model data with the MEC input set to zero, the model data could reproduce the empirical data with values of $\varphi_{inh}$ that were largely unchanged, along with downward-shifted A values compared to the control empirical/model data match (Fig. 7e). This suggests that the loss of the excitatory MEC inputs requires a compensatory reduction of inhibitory amplitudes, but that the increased phase variability observed in MEC-lesion data requires no major adjustments in the timing of inhibition.

Interestingly, the other two free parameters in the model – phase differences between the two excitatory inputs (i.e., ψ) up to at least ±90° and the addition of an inhibitory DC component up to a value of 1.1 – did not produce overly distinct ranges of match with empirical data (Supplementary Figs. S9b, S10a). In addition, major imbalances between excitatory input amplitudes of DG and MEC (75/125 or 125/75, Supplementary Fig. S10b) did not result in substantial variation in the state space for allowable empirical data. With complete lesions of each of the excitatory inputs to the model, there is an expected compensatory response in inhibition amplitude, which is approximately equal with the loss of MEC and DG inputs (Fig. 7e). However, accounting for the match of empirical data to the model after each of the two lesions required different adjustments in the $\varphi_{inh}$ dimension. We found that the inhibition phase needed to be broader after the loss of DG inputs, while it remained approximately in the control range after the loss of MEC inputs. Taken together, the lack of responsiveness of the model to changes in the ψ parameter (i.e., within the physiological range of ±90°) compared to its dependence on $\varphi_{inh}$ is interesting because it shows that phase precession is determined to a larger degree by the phase differences between excitatory and inhibitory inputs than by phase differences between two excitatory inputs. Overall, our simulations demonstrate that the range of observed effects in the CA3 circuit can, in principle, be generated by the interaction of major excitatory and inhibitory theta-modulated inputs to a CA3 cell (Fig. 7 and Supplementary Fig. S11).

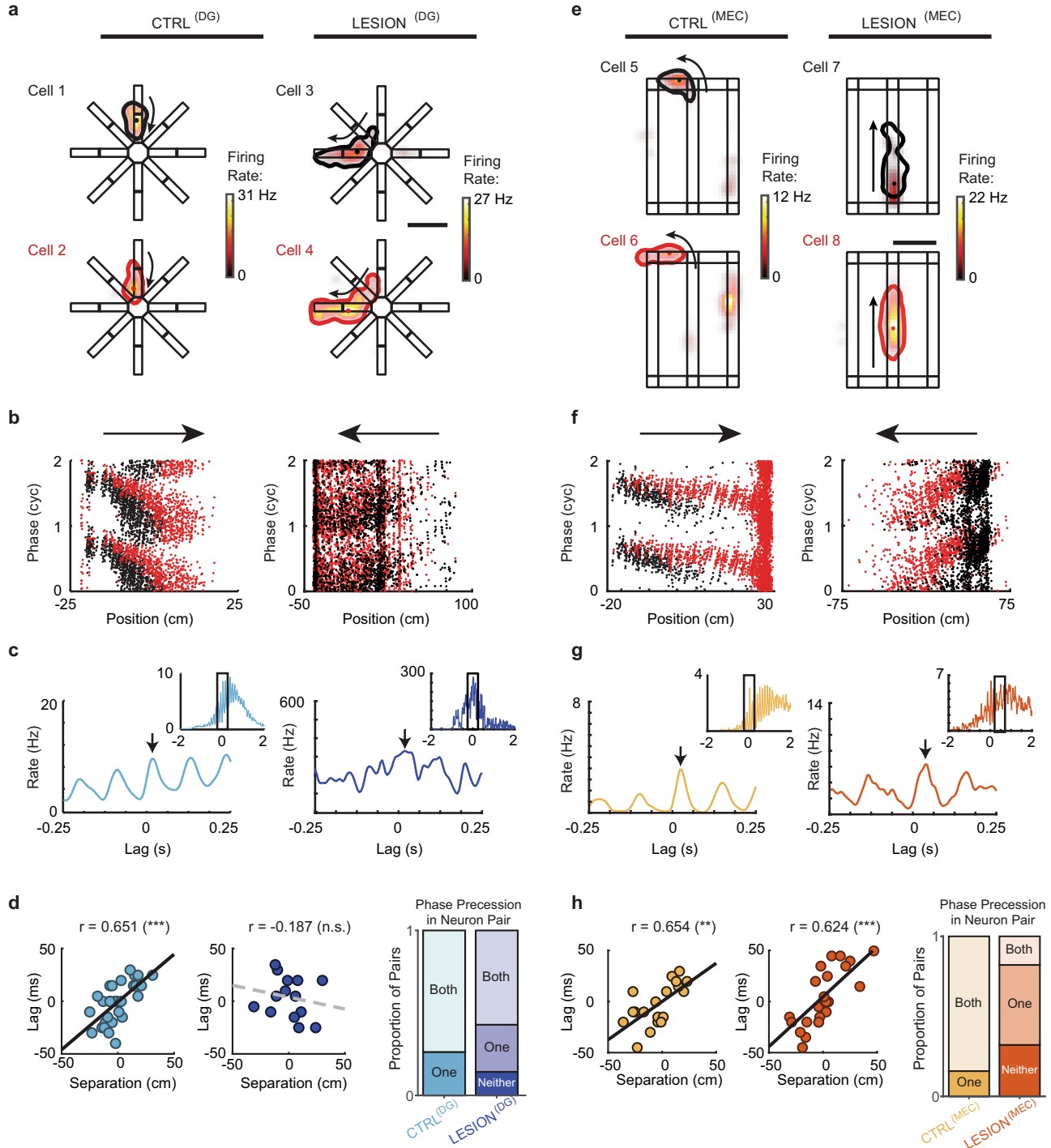

**Fig. 6 | DG but not MEC inputs are required for the temporal organization of CA3 cell pairs at the theta-cycle time scale. a** Pairs of simultaneously recorded CA3 cells with overlapping place fields were selected for the analysis of spiking in shared theta cycles. Cells 1 and 2 are from a control rat, and cells 3 and 4 are from a DG-lesion rat. Each place field is delineated by a contour that corresponds to 20% of the maximum firing rate (dot inside contour, location of peak firing). Each pair of overlapping fields is depicted with the one that is entered first in black and the one entered second in red. Black arrow, running direction. Scale bar, 50 cm. **b** Phase-position plots of the cell pairs' spikes (red and black dots, from the cells depicted in red and black in **a**) while running in the direction indicated by the horizontal arrows (corresponding to the direction in **a**). **c** Cross-correlation of the spikes (arrow, peak of the cross-correlation function nearest zero lag; inset, cross-correlogram for a window width of 4 seconds). **d** Left, Phase shift at the theta-cycle time scale plotted

against the distance between place field peaks (solid and dashed lines, linear regression for data from control and DG-lesion rats; CTRL$^{(DG)}$: $n = 30$ pairs, $r = 0.651$, $p = 9.87 \times 10^{-5}$, Pearson's correlation). Right, Proportion of cell pairs where neither, one, or both neurons in the pair displayed phase precession. Although a substantial number of pairs in the DG-lesion group were phase precessing, the phase precession did not yield a relation between pairwise spike-time difference and distance between fields (LESION$^{(DG)}$: $n = 14$ pairs, $r = -0.187$, $p = 0.523$, Pearson's correlation). **e–h**, Same as **a–d** except that the CA3 data are from the MEC-lesion group and their respective controls. Despite marked deficits in phase precession, there is a strong correlation between spike-time difference and place field distance (CTRL$^{(MEC)}$ $n = 19$ pairs, $r = 0.654$, $p = 0.0024$; LESION$^{(MEC)}$ $n = 27$ pairs, $r = 0.624$, $p = 5.02 \times 10^{-4}$; Pearson's correlations). n.s., not significant, ** $p < 0.01$, *** $p < 0.001$. Source data are provided as a Source Data file.

## Discussion

The DG is the first processing stage in the intrahippocampal circuit and is considered to perform a number of specialized computations that are critical for memory such as spatial and temporal pattern separation as well as novelty detection[3–10]. Furthermore, computational models also emphasize that the dentate-CA3 network forms a loop that could be used for generating and storing sequences[30,53], which in turn can be used for guiding ongoing behavior and decisions[20,21,23,28]. While there is recent evidence for a contribution of DG to the activation of CA3 ensembles during SWRs[12], the role of DG inputs to CA3 during periods when theta oscillations are predominant has not been established. Here, we show that diminished DG inputs to CA3 cells resulted in a substantial disruption of precise spike timing within theta cycles and in reduced theta phase precession. The reduced phase precession was accompanied by a disrupted temporal order of the spiking within a theta cycle for cells with overlapping place fields. It is possible that effects on the temporal activity patterns of CA3 cells are not specific to DG inputs but might emerge when any of the theta-modulated excitatory inputs to CA3 are diminished. We, therefore, compared the effects from reduced DG inputs to CA3 with the effects of reducing MEC inputs to CA3. Similar to our observation with DG lesions, we found that loss of MEC inputs resulted in reduced phase precession. Despite the phenomenological similarity when considering standard phase precession measurements, we identified profound differences in the effects of each manipulation of the major excitatory inputs on precise timing. Only DG but not MEC lesions precluded spikes from selectively occurring late in the theta cycle at the onset of spike trains, and only DG but not MEC lesions disrupted the pairwise timing between cells that are co-active. Given that manipulations of each of the two theta-modulated inputs to CA3 resulted in distinct effects of spike timing at the theta scale, we generated a minimalistic computational model that allowed direct comparisons to data and identified distinct coupling of each of the excitatory inputs to local inhibition as a plausible mechanism of how these differences emerge. By comparing the model to empirical data, we recognized that the effects that resemble DG lesions were more readily achieved by varying the amplitude and phase of the inhibition while the effects that resemble MEC lesions were more readily achieved by varying only the amplitude of the inhibition. Taken together, DG inputs to CA3, therefore, have a particularly pronounced role in generating the preferential spiking of CA3 cells during late theta phases and for the temporal order of spiking within a theta cycle.

While standard measurements of phase precession can broadly indicate that spike timing is altered, the more detailed measurements in our study provide further insight into the pattern of disruption. The selective effect on late spiking during the initial theta cycles is evident in the finding that diminished DG inputs preferentially broadened the phase of spiking within a theta cycle at the onset, but not at the offset of a spike train. We note that these phase shifts were unlikely a result of traveling wave theta phase differences due to tetrode placement[74] as this would have altered onset and offset distributions to the same extent, contrary to our observations. Rather, in-depth analyses of spiking during theta cycles revealed that DG lesions resulted in a broadening of the phase window during which spikes are generated during the initial theta cycles. In addition, we also found that the relative timing of CA3 cell pairs on a theta-cycle time scale depended on DG. Importantly, neither the pronounced broadening of the onset phase nor the selective effects on spike timing at the onset of spiking were observed with MEC lesions, which nonetheless reduced phase precession in CA3 to a similar extent as the DG lesions. Given that MEC layer II does not only project directly to CA3 but also to DG[75], it might have been expected that MEC lesions result in larger deficits when direct effects on CA3 and indirect effects via DG on CA3 combine. For example, it could have caused the minor decrease in CA3 firing rates with MEC lesions. However, the preserved pairwise spike timing of CA3

after MEC lesions and the moderately preserved phase precession in CA3 after MEC lesions differ from the profound disruption of pairwise spike timing with DG lesions and from the previously reported profound disruption of the temporal order in pairs of CA1 cells and of phase precession in CA1 cells after MEC lesions[54,76]. Our results also exclude the interpretation that the MEC lesions have effects over a broader range of theta phases than the DG lesions. The onset phases broadened with DG lesions, but were precisely matched to controls with MEC lesions. Similarly, circular variance increased selectively in the first half of trains with DG lesions, and not in any part of the trains with MEC lesions.

The less pronounced effect on the timing of CA3 firing patterns with MEC lesions compared to DG lesions could be a consequence of remaining LEC and medial septal inputs to DG, which preserve critical aspects of DG firing patterns. Similarly, the more severely disrupted CA1 than CA3 firing patterns with MEC lesions could be a consequence of a more major role of MEC projections to CA1 than to CA3 and/or a more minor role of the second external excitatory inputs – CA3 inputs to CA1 as opposed to DG inputs to CA3. As a consequence, the additional preserved inputs from DG are sufficient to preserve temporal organization in CA3 while CA3 inputs to CA1 are not[54]. The DG inputs thus confer the CA3 circuit with the propensity to generate sequential activity patterns, such that this computation – even when MEC inputs are diminished – can emerge with remaining DG projections to CA3[54]. Our data, therefore, suggest the broader DG-CA3 circuit is required to support the computations that generate theta sequences. While this observation is inconsistent with an early phase precession model that uses asymmetric synaptic weights in the recurrent CA3 network to generate phase precession[47], it has more recently been shown that recurrent networks need to be combined with external inputs or with mechanisms that lead to firing frequency adaptation to robustly generate phase precession or predictive coding[51,53,77]. While our data or any data that we are aware of do not directly test the role of recurrent collaterals, our findings support the notion that external inputs to CA3 are needed in addition to or in lieu of recurrent circuits for the generation of phase precession and for precise theta-scale spike timing. To test the suggested role of DG, spiking network models will need to be developed that consider separate DG and MEC inputs to CA3 in conjunction with a separate role of somatic and dendritic inhibition.

How is the DG-CA3 projection specialized to support the emergence of precise spike timing? Initial models of the DG contribution to phase precession have emphasized the strong facilitation at mossy fiber synapses from DG granule cells to CA3 pyramidal cells[52]. In this scenario, an initially weak excitatory input would be facilitated by repeated activation of the synapse across theta cycles and become increasingly more effective in overcoming rhythmic inhibition, such that spiking occurs at progressively earlier theta phases across theta cycles. However, this straightforward model is not consistent with recent data, which show that inputs from granule cells to inhibitory interneurons in CA3 will result in feedforward inhibition that at least matches, if not exceeds, the facilitation at mossy fiber synapses to CA3 pyramidal cells, in particular at the time scale across theta cycles[78,79], which is particularly relevant to phase precession. Accordingly, our analyses suggest that DG and MEC circuits make qualitatively different contributions to the organization of the precise temporal profiles of CA3 spiking, consistent with a model in which DG inputs effectively promote spiking at late theta phases early in the spike train of a CA3 neuron (i.e., upon entry into the place field) (Fig. 8a). In contrast, later in the spike train (i.e., in the middle and near the exit from the place field), MEC inputs appear to ensure an appropriate mean theta phase of CA3 spiking by driving CA3 neurons in time windows around a monotonically decreasing mean theta phase over successive theta cycles (Fig. 8b).

Given the major role of inhibition in shaping the spike timing in intact neural circuits, we used a phenomenological model and made

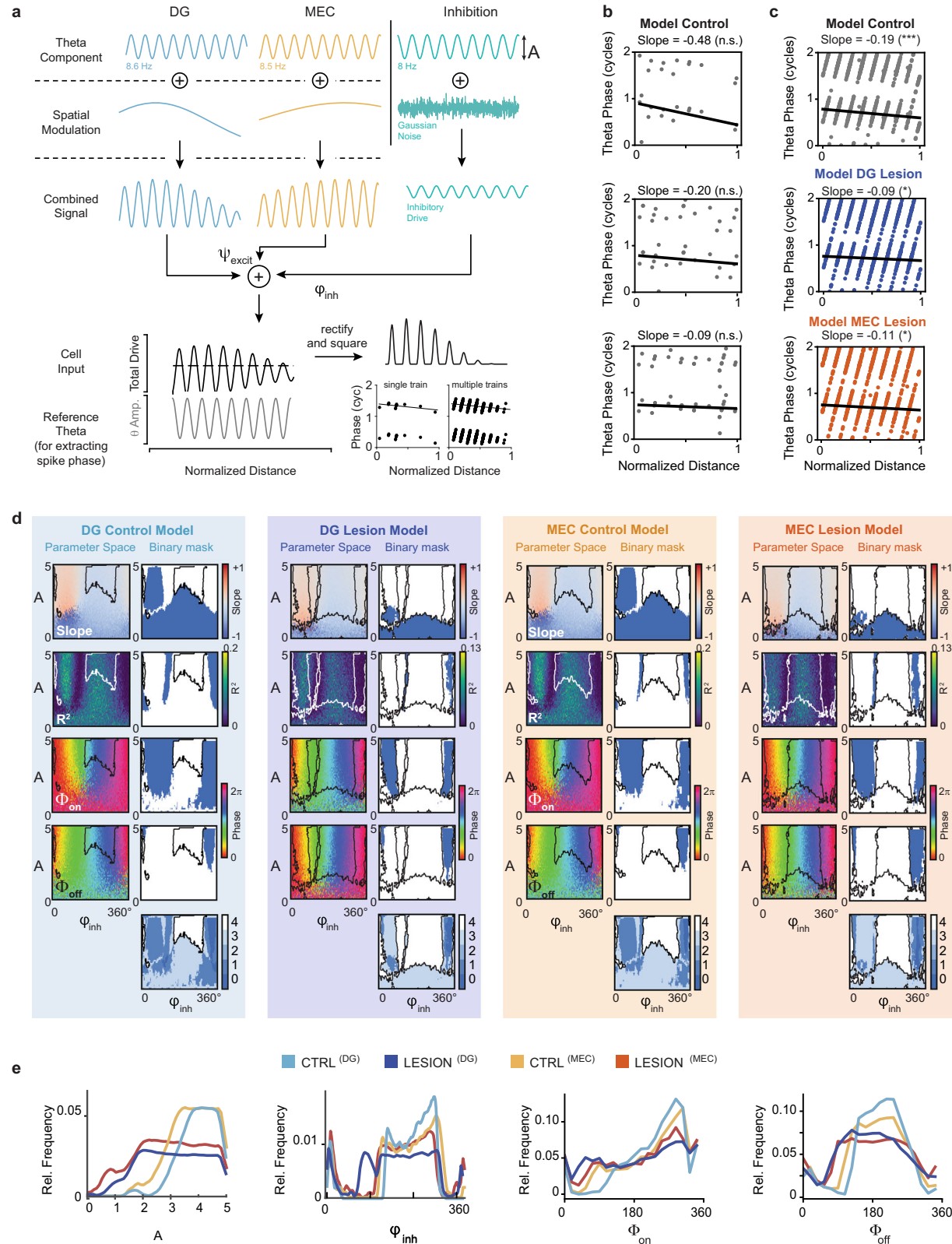

sure that it was sufficiently minimalistic (i.e., had few free parameters) to allow comparisons to experimental data. While keeping the parameters to a minimum, we reasoned that it was essential to add the well-established oscillating inhibitory inputs in addition to the two oscillating excitatory inputs that were tested in our analyses of experimental data. The model is, therefore, conceptually related to previous models of phase precession that have considered oscillatory interference between an inhibitory somatic drive and dendritic excitation[35,49,80] except that two independent excitatory inputs rather than a single input were used here. Model parameters included the theta phase difference between the excitatory and inhibitory inputs and the oscillatory amplitude of inhibition. By exhaustively searching the parameter space of the model, we identified model parameters that gave rise to phase precession measurements that corresponded to

**Fig. 7 | A single-cell model with two oscillating excitatory inputs and an inhibitory input reproduced the main empirical results. a** Model construction. The three inputs are modeled after DG, MEC, and local inhibition converging onto the CA3 model neuron. The excitatory DG and MEC inputs oscillate at faster-than-LFP frequencies ($v_{DG}$ = 8.6 Hz, $v_{MEC}$ = 8.5 Hz) with DG inputs more prominent early in the field and MEC inputs more prominent later in the field. The inhibitory input oscillates at 8 Hz throughout the place field, corresponding to the LFP theta. Small Gaussian noise is added to the inhibitory input to ensure robustness against minor perturbations. The excitatory inputs contribute positively at the fixed phase difference ψ, which is taken to be 0° from published findings[68] on DG and MEC population activity. The inhibitory input contributes negatively to the total drive at a phase differential $\varphi_{inh}$ relative to excitation at place field entry. Finally, the total drive is rectified. A reference 8 Hz oscillatory inhibition is displayed at the bottom left, which is used to extract the phase of the simulated spikes. The phase-distance relationship is then depicted as for experimental data. Not all steps are displayed for brevity (see "Methods" for full details). **b** Phase-versus-normalized distance plots of spikes generated by the model. Three randomly selected single-pass examples show phase precession. The values of A/$\varphi_{inh}$ = 3/260° (inhibitory oscillation amplitude and phase), $I_{DC}$ = 0 (inhibition DC component) and ψ = 0° (excitatory phase differential) are the same across the three plots. The measured slopes from the simulated data are displayed at the top of each panel along with the statistical significance based on circular-linear regression. **c** Phase-versus-normalized distance plot for spikes across multiple passes. Panels from top to bottom are generated by the control, DG lesion, and MEC lesion models (1

cycle = 360°). The measured slope and significance of phase precession are displayed on top of each panel based on circular-linear regression. n.s., not significant, * $p < 0.05$, *** $p < 0.001$. Lesion experiments were simulated by setting the DG or MEC input to zero, and all other parameters as in b except for A lesion = 2. In both lesion cases, phase precession slopes are reduced. **d, e** DG and MEC lesions alter CA3 phase precession in qualitatively different ways. **d,** Values of the slope, explained variance ($R^2$), onset phase ($\Phi_{on}$), and offset phase ($\Phi_{off}$) are shown for combinations of A and $\varphi_{inh}$ parameters. The color scale in each panel is according to the color bar to the right. The range of model A-$\varphi_{inh}$ parameter space that corresponds to empirical phase precession values are shown in blue and white plots to the right with white areas depicting the space that yields 80% of the empirical measurements. The intersection between white areas for multiple phase precession measurements is displayed in the overlap plots at the bottom (dark blue to white, 0 to 4 measurements overlap). The zone of overlap for all four measurements is delineated with outlines (black or white lines) that are projected back onto all other panels. **e,** left, Distribution of parameter values A and $\varphi_{inh}$ that result in a match with the empirical control and lesion data. To match the empirical phase precession measurements, the DG-lesion but not the MEC-lesion model is forced to take on a shifted set of $\varphi_{inh}$ values. **e,** right, Distributions of $\Phi_{on}$ and $\Phi_{off}$ values that are generated by the A-$\varphi_{inh}$ parameter space that corresponds to the overlap areas of each model type. The DG lesion model generates the broadest $\Phi_{on}$ distribution. DG control, light blue; DG lesion, dark blue; MEC control, yellow; MEC lesion, red. Source data are provided as a Source Data file.

the empirical observations. We then eliminated either the DG or the MEC input to the model and repeated the search for a feature space of rhythmic inhibition that corresponded to the empirical data. The resulting phenomenological models can reveal patterns of rhythmic inhibition that are compatible with the observed changes in the temporal firing patterns of CA3 cells. By mapping the experimental findings to the computational model, we found that the results suggest that the DG and MEC input pathways could be exerting two different types of effects on the inhibitory subnetwork. The consequences of loss of DG inputs can be explained by a combined expansion in the inhibition phase and decrease in the inhibition amplitude, whereas the consequences of loss of MEC inputs can be explained solely by a decrease in inhibition amplitude (see Fig. 7e). The shifts in amplitude are expected because the loss of one of the excitatory pathways results in diminished excitation that is offset by a lower level of inhibition. Interestingly, the model without DG input is compatible with a less constrained input to fast-spiking interneurons that are targeted by granule cell projections to CA3[81,82]. In contrast, MEC inputs are known to not only target pyramidal cells but also somatostatin interneurons that predominantly control dendritic inhibition, and manipulations of dendritic inhibition have been shown to be without effect on the average spike phase throughout the place field[82], which resembles our observation that the model without MEC inputs does not need major adjustments to the inhibitory phase to explain data. Although these predictions from the model are yet to be confirmed by recording from identified dendrite-targeting and soma-targeting interneurons, these data suggest that effects from manipulating excitatory inputs do not only arise from diminished direct connectivity to principal cells, but also from how these inputs engage inhibitory interneurons. In particular, our data are consistent with the mossy fiber inputs to CA3 more strongly engaging somatic inhibition, which determines the theta phase of spikes, and with MEC more strongly engaging dendritic inhibition which does not directly set the theta phase. The different functions of input pathways imply that manipulations and models that link phase precession to theta sequences and theta sequences to behavior will need to consider multiple input pathways and perhaps even a much larger circuit that includes other brain regions with phase precession, such as MEC and CA1.

It is generally assumed that firing at precisely timed theta phases is a prerequisite for generating sequences of neuronal firing patterns[41,83], but the combinations of inputs that are necessary for implementing

these computations have not been established. For example, one model of phase precession proposes that phase precession can emerge by combining two excitatory inputs that each have a different phase preference with respect to the theta cycle[50]. When the strength of one input increases and of the other decreases along the animal's path, the spiking of a cell that integrates these inputs would show a phase shift over successive theta cycles. We tested versions of our model in which we offset the phases of the two excitatory inputs with respect to each other, and we show that there are no major qualitative differences compared to versions in which the two excitatory inputs are in phase. Rather, we observe that the inhibitory phase continues to constrain the match to control data for any of these models, which suggests that the phase of the inhibition with respect to excitation rather than a phase difference between excitatory inputs is the most critical parameter.

Taken together, our data and phenomenological model identify that it is the spikes in the initial theta cycles and at late phases of the theta cycle that are particularly dependent on DG input and on precisely timed inhibitory oscillations. The late-phase spikes are thought to emerge from internally stored patterns of synaptic strength that generate prospective neuronal activity[30], and our results thus suggest a critical contribution of DG for such activity patterns to emerge in CA3 networks. This is conceptually aligned with our previous report that the DG network contributes to prospective neuronal activity patterns during SWRs[12] and is consistent with the hypothesis that DG inputs are essential for sequence coding, future planning, and for generating intrinsic "look-ahead" spikes in CA3[30,71,84,85]. While such a function has been proposed by numerous computational models, our results provide experimental evidence for the role of DG beyond the previously established functions of pattern separation and novelty detection[3,8].

## Methods

### Subjects and surgical procedures

All experimental procedures can be found in previous publications describing data analyzed in the present study[12,55]. We reanalyzed and compared CA3 activity patterns from these data, including a total of 31 rats (Supplementary Table S1). These included 4 control and 9 dentate-lesion rats (7 and 16 sessions; DG lesion experiment) with CA3 or dual CA3-DG (2 of the 4 control rats) single-unit recordings, 7 control and 8 MEC-lesion rats (18 and 20 sessions; MEC lesion experiment) with CA3 single-unit recordings, and 3 control rats with only DG recordings.

Male Long-Evans rats between the ages of 3 and 6 months (300–350 g) were used as subjects. The animals were kept on a 12-hour light-dark cycle (7 AM to 7 PM dark) and housed individually. In vivo recordings were conducted in the dark phase. Rats were restricted to 85% of their ad libitum weight and given full access to water. All procedures were conducted in accordance with the University of California, San Diego Institutional Animal Care and Use Committee, at the University of California, San Diego according to National Institutes of Health guidelines.

## Experimental procedures and brain lesions

The details of the DG lesion and MEC lesion experiments were published previously[12,55]. In brief, rats in the DG lesion experiment were trained on the 8-arm radial maze to perform a spatial-working memory task (see Behavioral tasks section). Rats initially designated to receive DG lesions ($n = 9$ animals; LESION[(DG)]) underwent a surgical procedure during which colchicine was bilaterally infused along the septal-temporal axis of the DG. The remaining rats ($n = 4$ animals; CTRL[(DG)]) were subjected to a sham lesion. During the same surgical procedure, a hyperdrive of 14 independently moving tetrodes was implanted above the right hippocampus for electrophysiology as described below. Rats in the MEC lesion experiment were trained on a working memory task, the figure-8 continuous spatial alternation task (see Behavioral tasks section). These rats underwent a surgical procedure in which the control rats ($n = 7$ animals; CTRL[(MEC)]) received a sham lesion (injection of the vehicle) and the experimental rats ($n = 8$ animals; LESION[(MEC)]) received an excitotoxic lesion of MEC by an injection of NMDA.

In post-mortem histological material, the final position of the recording tetrodes was confirmed by performing cresyl violet staining of the sectioned brain tissue. In the DG Lesion experiment, the loss of dentate granule input to CA3 cells was confirmed by TIMM stains as previously described in detail[12]. The extent of DG granule cell damage was quantified in a localized fashion. Specifically, each tetrode ending location was scored based on the intensity of the TIMM-positive staining in histological sections. Scores of 0 (~ 0% TIMM-positive signal), 1 (< 30% signal), or 2 (< 70% signal), and 3 (> 70% signal) were assigned to each of the tetrodes in DG-lesion animals, and only tetrodes with scores of 0, 1, and 2 were included in the LESION[(DG)] data set. The extent of MEC lesions was confirmed quantitatively[55], and on average, 93.0% of the total MEC volume was ablated (95.3% of layer II, 92.4% of layer III, and 91.4% of deep layers) with any sparing typically observed in the most ventral portions of MEC (Supplementary Fig. S1).

## Hyperdrive implants

An array of 14 independently movable tetrodes was implanted over the right hippocampus in all 31 rats (control group: 4.0 mm posterior and + 2.7 to + 2.9 mm lateral to bregma; DG lesion group: 3.5–4.4 mm posterior and + 2.8 to + 3.2 mm lateral to bregma; MEC lesion group: − 4.0 mm posterior and + 2.8 mm lateral to bregma). The hyperdrive was secured with skull screws and dental cement to prevent mechanical instability. The tetrodes (with tips platinum plated to 150–300 kΩ at 1 kHz) were slowly lowered each day over a period of 2–4 weeks to ensure recording stability and minimize damage to the brain. Depth records, LFP signals, and neural spiking markers were used to estimate tetrode distance from the target region. After an initial period of larger advances, the tetrodes were moved only in small increments over several days until a satisfactory signal (i.e., low-amplitude multiunit activity) was observed. Once near CA3, the tetrodes were allowed to settle inside the stratum pyramidale of the CA3 of the hippocampus with no further active movement of the tetrodes to maximize recording quality (i.e., high-amplitude multiunit activity). In 3 of the 31 rats all tetrodes were lowered to the dentate granule layer.

## Electrophysiological recordings

A Neuralynx Cheetah recording system with a multichannel head-mounted preamplifier was used for LFP and single-unit recordings. A signal from a skull screw was used as animal ground, and a reference signal from the neocortex was subtracted from the hippocampal signals to increase the hippocampal signal-to-noise ratio. Unit recordings were filtered at 600 Hz to 6 kHz, and spike waveforms above an amplitude of 40 μV were time-stamped and recorded at 32 kHz for 1 ms. LFP recordings were filtered between 1 and 425 Hz in the DG-lesion experiment and between 1 and 450 Hz in the MEC-lesion experiment.

## Behavioral tasks

**DG lesion experiment (spatial working memory on the 8-arm radial maze).** The rats in the DG lesion experiment were trained to perform a DG-dependent spatial working memory task[12]. The task used a maze with a central platform and 8 radial arms that each had a proximal segment that could be lowered and raised. The rats were first placed on the central platform (i.e., "stem") of the 8-arm maze with all 8 arms lowered such that the reward cups at the end of each arm were inaccessible to the animal (Supplementary Fig. S2a). Next, the experimenter raised one arm at a time, for four arms, following a previously generated pseudorandom sequence. The rat was allowed to run down each raised arm, and upon its return to the stem, that arm was lowered, and the next arm in the sequence was raised. Once the rat had visited all four experimenter-forced arms ("forced choice" phase), all 8 arms were raised and available for the rat to visit ("free choice" phase). The optimal strategy would consist of the rat visiting every one of the four arms unvisited during the forced-choice phase without reentering any arm. A total of 16 rats ($n = 4$ CTRL[(DG)]; $n = 9$ LESION[(DG)], $n = 3$ with only DG recordings) were trained and tested in this task while performing single-unit recordings in CA3 and/or DG.

**MEC lesion experiment (spatial working memory on the figure-8 maze).** In the MEC lesion experiment, the rats were trained to perform a hippocampus-dependent alternation task on the figure-8 maze[55]. In this task, a rat is placed in a delay zone at the base of the figure-8 maze (Supplementary Fig. S2b) and is required to run up the "stem" of the maze toward a T-junction from where it can choose between reward locations on either the left or right before returning on a side arm to the delay zone. Blocks of trials with and without delays were performed. In non-delayed trial blocks the delay site is not used to restrict the animal's movements. In delayed trial blocks a barrier restricted the rat's progress for 2, 10, or 60 s for each trial. These blocks were not distinguished for the analyses presented here. The first lap ("trial 0") is discarded. However, it is used to determine the success of the animal in choosing the right or left reward on the following lap. From the second lap onwards (trial 1 and later), a trial is "correct" if and only if the animal chooses to visit the reward location not visited on the preceding trial. If the animal chooses the same side more than once, it will not receive a reward at the visited reward site. It will, however, continue to receive reward items at the appropriate reward sites as soon as it chooses the side not chosen on the previous trial. A total of 15 rats ($n = 7$ CTRL[(MEC)]; $n = 8$ LESION[(MEC)]) were trained and tested on this task while CA3 recordings were performed.

## Data analyses

All statistical tests were chosen to appropriately match the underlying data distributions. In the case of testing proportions, $\chi^2$ tests were used. For testing of differences between central tendencies, first the normality and homoscedasticity were tested with Anderson-Darling and F-test, respectively. Data were not normally distributed, and Mann-Whitney (MW) or Kruskal-Wallis tests were applied. For testing of differences between circular means, one-factor MANOVA was used. For comparing distributions, the Kolmogorov-Smirnov test was used. The

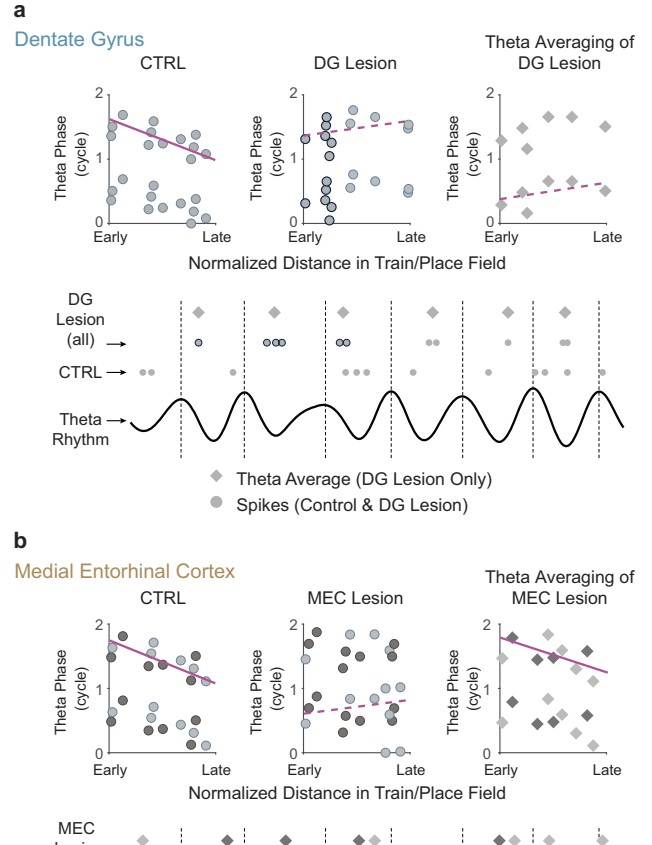

**a**

Dentate Gyrus

CTRL · DG Lesion · Theta Averaging of DG Lesion

**b**

Medial Entorhinal Cortex

CTRL · MEC Lesion · Theta Averaging of MEC Lesion

Fig. 8 | **Schematic of how DG and MEC lesions can disrupt CA3 temporal coding. a** Top left, In control animals, initial spikes of the place field occur late in the theta cycle and progressively advance to earlier phases further into the place field. Spikes are depicted by gray circles. Top middle, Compared to the control condition, DG lesions result in redistributed/additional CA3 spikes early in the theta cycle at the onset of trains (gray circles with black outline). Top right, theta cycle averaging (gray diamonds) is unable to rescue phase precession from this pattern of results. Bottom, Schematic of spiking with reference to the theta cycle, corresponding to the above plots. **b** Schematic of how MEC lesions can disrupt CA3 temporal coding. Top left to right, Two spike trains are depicted by different shades of gray. Increased spike time variability within and across spike trains in each cycle lowers the likelihood of detecting phase precession with MEC lesions in the spike-by-cell analysis. Theta averaging rescues phase precession because, in contrast to (**a**), it at least partially reverses the heightened variability across successive traversals of the place field (i.e., trains 1 and 2). MEC thus supports the consistency of theta phase coding in CA3 pyramidal cells.

α level was set to 0.05 for all experiments and tests. All hypothesis testing was performed two-sided unless noted otherwise. Multiple comparisons correction was performed only in the case of Fig. 4d and is presented in Supplementary Table S2. All other $p$-values are presented uncorrected for multiple comparisons.

**Spike sorting.** Spike sorting was performed offline using a custom version of MClust 3.5 (Redish, A.D., http://redishlab.neuroscience.umn.edu/MClust/MClust.html). Clusters were selected in the sleep

sessions before and after a behavior and matched to the data recorded during the behavior to ensure consistency and reliability. The cross-correlogram was used as an additional criterion to ensure cluster independence. Only well-separated clusters were retained for analysis.

**Spatial firing properties.** The measurements included in Supplementary Fig. S3 are defined as follows. Let $N$ be the total number of spikes of a given cell, and the number of such spikes that were part of a detected train (as defined below) denoted $N_t$. The proportion of spikes assigned to a train is defined $\frac{N_t}{N}$. The firing rate is defined as $\lambda = \frac{N}{T}$ where $T$ is the summed total duration in seconds of all behavioral trials in a given session. The number of spikes per train is calculated for each cell as the $N/N_{tr}$ where $N_{tr}$ is the number of detected trains for that cell. To calculate the train length we first found the physical position of the first and last spikes of a train on the maze (points $P_1 = (x_1, y_1)$ and $P_N = (x_N, y_N)$). The train length $L$ is calculated as $L = \sum_{i=2}^{N} ||P_i - P_{i-1}||$, where $||\cdot||$ is the $L_2$ norm. The number of bins covered was calculated as the total number of square bins of size 2 by 2 cm that contained at least one point from the path of a detected train. The information content measure was adapted from ref. 86 and was calculated as $H = \sum_x \lambda(x) \log_2 \frac{\lambda(x)}{\lambda} p(x) \delta x$, where $\lambda$ is the mean firing rate, and $p$ is the probability mass function of the rat's position over the spatial bins of the maze. The selectivity and sparsity measurements were adapted from ref. 46 and are defined, respectively, as $S = \frac{\mathbb{E}[\lambda]}{\mathbb{E}[\lambda^2]}$ and $s = \frac{\lambda_{max}}{\lambda}$, where $\mathbb{E}$ is the expected value over the spatial bins and $\lambda$ is the mean rate as defined above.

**Rate map construction.** First, intervals during the periods delimited by trial timestamps in which the velocity of the animal exceeded 2 cm/s were selected. For each cell, all spikes that occurred outside these intervals were excluded for the construction of rate maps. Next, the environment was divided into square bins of side length 5 cm and the spikes that occurred in each such bin were counted. The occupancy matrix was constructed similarly by counting the number of position tracking points falling in each spatial bin multiplied by the tracking acquisition rate (29.97 FPS). The rate map was the result of the element-wise division of the spike count matrix by the occupancy matrix, spatially smoothed with a 2-d Gaussian kernel size 15 cm.

**Spike train detection.** A spike train was defined as a set of 5 or more consecutive spikes with a maximal inter-spike interval of 500 ms. Additional criteria were imposed on the selection of spike trains for analysis (Supplementary Fig. S2c). A spike train (a.k.a. "pass") was deemed valid for analysis if it was at least 300 ms and no more than 2500 ms in duration, its corresponding path was at least 20 cm long, its corresponding path endpoints were at least 10 cm apart in physical space, and if the average velocity of the animal during the pass exceeded 2 cm/s. All of a cell's detected trains were discarded if its mean firing rate over the duration of the behavior was smaller than 0.1 Hz or greater than 5 Hz.

**LFP analysis and theta phase extraction.** Local field potentials were recorded from one of the electrodes for each tetrode. The raw LFP signal was filtered in the theta range (6–10 Hz), and the channel with the largest theta rectified RMS power was selected as the reference for phase precession analysis. The phase estimate was obtained by $f = \text{atan2} \frac{\Im m(H(s))}{\Re e(H(s))}$ under linear interpolation, where $H(\cdot)$ is the Hilbert transform and $s$ is the 6 Hz to 10 Hz filtered LFP signal.

For power analyses of frequency bands in addition to the theta range (6–10 Hz), the raw LFP signal was filtered in delta (1–4 Hz), slow gamma (25–50 Hz), and fast gamma (50–100 Hz) and band power was calculated using Matlab's 'bandpower' function. Phase-amplitude coupling of theta and gamma oscillations was performed using

published Matlab code (https://github.com/tortlab/phase-amplitude-coupling)[87]. In addition, the speed dependence of theta frequency was analyzed by first calculating the time-resolved spectrogram (5 s moving window in steps of 0.5 s) of the LFP using Chronux (`mtspec-gramc_fast` function, 1–20 Hz with time-bandwidth product TW = 3 and number of tapers $K = 5$), by then finding the frequency with the peak power in the 6–10 Hz band at each position, and by finally estimating the theta frequency associated with each position by linearly interpolating the spectrogram-based frequency estimates. Next, the speed in each session was binned with a resolution of 2 cm/s, and a regression analysis was performed for the data points of each session as well as for the data points of all sessions of a group.

To confirm that the phase estimate is not unduly biased by asymmetric theta waves, as previously shown for CA1 recordings[56], we examined theta wave asymmetry at all CA3 recording sites that were selected as the reference for phase precession analysis. To calculate an asymmetry index, we adapted the methods of ref. 56 with minor modifications to account for the higher gamma amplitude in CA3 compared to CA1. We began by bandpass filtering the raw LFPs in the theta band (6–10 Hz) and in the broader 1–80 Hz band, as in ref. 56. Using the 1–80 Hz bandpass-filtered signal, we then identified the maximum within the first half the theta cycle (0–180° of the 6–10 Hz filtered signal) and the minimum within the second half of the theta cycle (180°–360° of the 6–10 Hz filtered signal) and marked these extrema as peaks and troughs, respectively (Supplementary Fig. S5a). The asymmetry index was then the ratio of the duration of the ascending wave segment (trough to peak) divided by the duration of the descending wave segments (peak to trough) on a logarithmic scale. Using this scale, zero corresponds to a symmetric wave shape. To then quantify the asymmetry for each recording session, we calculated the mean asymmetry index over all theta cycles of a recording session (Supplementary Fig. S5b).

**Quantification of phase precession.** The distance and theta phase variables were extracted from detected trains. For each train, the last sampled $(x, y)$ coordinate before the first spike and the first sampled $(x, y)$ coordinate after the last spike was marked as the start and end of that train's corresponding trajectory. Next, spike positions were normalized with respect to the start and end points, yielding vector d of normalized distances. Taken together with the theta phase vector $\vartheta$ described above, the circular-linear regression was then computed on the $(d, \vartheta)$ pairs for each train or cell, as described below. The circular-linear regression produced a slope value $s$ and the estimate of explained variance $R^2$. The onset phase $\Phi_{on}$ was calculated for each train as the circular mean theta phase of the spikes occurring in the first (possibly truncated) theta cycle of the train. The offset phase $\Phi_{off}$ was defined similarly, except over the last theta cycle of the train. These four values are referred to as phase precession "measurements" in the modeling section. For statistical analysis of onset and offset phases, the median of the mean phases of each cell's trains was taken, which resulted in one value per cell.

**Population level quantification of remaining phase precession**
Two methods were employed for the quantification of the phase precession in each data set: slope-by-cell analysis and slope-by-train analysis.

**Slope-by-cell analysis.** In this method, for each CA3 cell, we performed the circular-linear regression analysis on the set of all spikes that belonged to a detected train from that cell. A cell was deemed to exhibit "phase precession" if the circular-linear regression $p$-value was less than $\alpha = 0.05$ and the slope was negative[88]. The quantification of proportions was performed on the values thus obtained.

**Slope-by-train analysis.** Here, the circular-linear regression was performed on the spikes from individual trains $i$ of each cell, to produce phase precession measurements. To obtain a cell-specific slope value, the $s^{(i)}$ values were averaged for each cell. The statistical comparisons of proportions were then performed on the cell-averaged values (such that each cell contributed a single slope value regardless of the number of its trains). The $\Phi_{on}$ and $\Phi_{off}$ values were only defined for trains, though instead of averaging them per train, they were directly used to find the distributions used in Fig. 3.

**Calculation of circular variance and statistical tests of circular data (e.g., theta phase values).** The circular statistics toolbox CircStats[89] was used for the computation of statistical quantities, such as the circular variance of theta phase (`circ_var` function of the CircStats toolbox). The CircStats toolbox[89] also implements many statistical recipes by ref. 90 and was used for statistical tests of circular data.

**Spike phase variance analysis.** For Fig. 5d, the circular variance was calculated either across all spikes of each neuron or across the time-stamps resulting from replacing each cycle's spikes with their mean. Notice that even though this operation reduces the total number of spikes, it will not necessarily reduce the variance; this would depend on the distribution of spikes within the theta cycle and the phase reliability firing windows of a cell over multiple trains. As shown, the outcome of this analysis, therefore, depends on the experimental conditions (CTRL, DG lesions, or MEC lesions).

**Standard place field definition.** Standard place fields of neurons were defined as the area within the 20% contour of the place maps, and this definition was used for Supplementary Figs. S6 and S7. Normalized distances along the path from the entry to the exit were calculated, and the distances were divided into 10 bins for further analysis. For example, we calculated the circular standard deviation across all spikes of each cell in each bin and plotted the mean values together with the error bars representing the standard error of the mean (Supplementary Fig. S6c).

**Onset, offset, and binned theta phase estimation.** The onset firing phase was defined as the circular mean phase of the spikes occurring in the first (partial) theta cycle of each train. The offset firing phase was analogously defined as the circular mean phase of the spikes occurring in the last (partial) theta cycle of each train. The histograms in Fig. 3 are obtained from the single train onset and offset phase values for each group. To estimate firing probability in the binned theta cycle (Fig. 4), we assigned a bin label (early, mid, or late, corresponding to [0, 2π/3), [2π/3, 4π/3), [4π/3, 2π), respectively) to each spike and plotted the resulting discrete probability distribution (panels a and b). This approach was repeated for each of 10 equal bins of the normalized position of spikes within a train to get a "position-resolved" theta bin firing probability estimate (panel c).

**Test for common mean in circular data (Circular MANOVA).** As suggested by ref. 91, we used a one-factor MANOVA test to test for differences in the mean of circular data (e.g., theta phase). Each phase $\theta$ was treated as a single observation with two response variables $\cos(\theta)$ and $\sin(\theta)$, with the experimental group (control or lesion) as the sole factor. The $p$-value was automatically obtained by Matlab's `manova1` function by comparing the test statistics with the chi-square distribution with 2 degrees of freedom. This procedure is referred to in the main text as "circular MANOVA." We avoided using the Watson-Williams test due to its inapplicability when the mean resultant length of the pooled data is < 0.45, as well as the superior performance of MANOVA[91]. Other tests (permutation tests, Kuiper test, Watson's $U^2$, CircStats[89] test for common medians) returned similar patterns of results (not shown).

**Test for common concentration parameter in circular data.** As suggested by ref. 91, we used the concentration test from the Directional toolbox[92]. However, we translated this code to Matlab before applying it to data.

**Calculation of effect size.** To quantify the effect size of linear ratio scale data (linear data), we used Cohen's $d$ defined as $d = \frac{m_1 - m_2}{s_{12}}$ where $s_{12} = \sqrt{\frac{(n_1-1)\cdot \nu_1 + (n_2-1)\cdot \nu_2}{n_1 + n_2 - 2}}$, and $n_i$ are the number of data points and $\nu_i$ are the data variances. To quantify effect size for circular data, we used the same formula but using the circular version of mean and variance functions.

**Cell pair sequence analysis.** Two simultaneously recorded units were considered a "pair" for the purpose of sequence analyses if (a) there were at least 50 theta cycles in which both units spiked, (b) at least 20% of each unit's spikes throughout the session occurred in theta cycles in which the other unit also spiked, and (c) at least 10% of all theta cycles in which a unit spiked included spikes from the other unit as well. For each unit pair, the cross-correlogram was computed and the relative time, $\tau$, of the peak closest to the zero time lag was found. The physical separation of the peaks of the two units' place fields, $d$, was computed and used to make the tuple $(d, \tau)$. In the case of the 8-arm maze where runs in opposite outbound and inbound directions were possible, each unit pair was treated twice—once for the inbound run and once for the outbound. We confirmed that in all cases, only one of the two-run directions had enough spikes to reliably assess pair co-modulation (i.e., the unreliable direction did not contribute a $(d, \tau)$ tuple). Once all $(d, \tau)$ tuples were obtained for each experimental group (CTRL[(DG)], LESION[(DG)], CTRL[(MEC)], and LESION[(MEC)]), a linear regression model was fit to the data to assess the significance of the relationship between place field separation ($d$) and theta co-modulation ($\tau$), as reported in Fig. 6. For the phase precession plots in Fig. 6b and Fig. 6f, trains from each cell were first mapped to the animal's path. The path equivalent of each train was then projected onto the line segment connecting the place field peaks of the two cells via a dot product. The midpoint of this line segment was considered the origin ($x = 0$) and was used on the $x$-axis of the phase-position plots. The sign of the direction of travel was defined to be positive if Cell 1's place field was visited before Cell 2's place field; otherwise, it was considered negative.

**Computational model**
**Model description.** The model CA3 neuron received three distinct inputs. Two of these were excitatory (i.e., positively contributed to the model neuron's total drive) and one was inhibitory (i.e., negatively contributed to the total drive). The excitatory inputs modeled the DG and MEC monosynaptic excitatory drive that CA3 principal cells receive, while the inhibitory input modeled the total inhibitory input that these neurons receive. The equations governing the value of each of the three functions took the following forms:

$$G_{DG}(t) = \gamma_{DG}\left(1 + \cos\left(2\pi\nu^{DG}t\right)\right) \quad (1)$$

$$G_{MEC}(t) = \gamma_{MEC}\left(1 + \cos\left(\psi + 2\pi\nu^{MEC}t\right)\right) \quad (2)$$

$$G_{INH}(t) = I_{DC} + A\cos\left(\varphi + 2\pi\nu^{INH}t\right) \quad (3)$$

where $\nu^{DG} = 8.6\,\text{Hz}$, $\nu^{MEC} = 8.5\,\text{Hz}$, and $\nu^{INH} = 8\,\text{Hz}$, and $I_{DC}$ represented the DC (baseline) component of inhibition. In the above equations, $\psi$ controls the theta phase difference of the two excitatory inputs by essentially timing only the MEC input while the DG input remains the same. Effectively, this could cause the place field to shift

around slightly which we shall ignore. Since the mean firing phase in DG and MEC is roughly similar in empirical data[68], we chose $\psi = 0$ for the main analysis. Variation of $\psi$ up to ±90 degrees, however, did not qualitatively alter the result (Supplementary Fig. S9). The values for the oscillation frequencies in the theta range (8.6 Hz for DG and 8.5 Hz for MEC) were estimated from Supplementary Fig. 15 in ref. 68. $I_{DC}$ was drawn from a Gaussian with a constant mean between 0.5 and 25 (with specific values given in Results) and a variance of 0.025.

To produce the total drive, the inputs were combined as follows:

$$G(t) = H\left(M_{DG}(t)\cdot G_{DG}(t) + M_{MEC}(t)\cdot G_{MEC}(t) - \frac{1}{8}G_{INH}(t)\right) \quad (4)$$

$H$ is the Heaviside function ("rectify" step in Fig. 7) and $M$ represents a spatial modulation function to each of these inputs to mimic the influence of DG and MEC inputs to the early and late portions of place fields, respectively (see Fig. 4)[71]. These functions are defined as follows:

$$M_{DG}(x) = \mathcal{N}_x(0.3, 0.45)^{\eta_{DG}} \quad (5)$$

$$M_{MEC}(x) = \mathcal{N}_x(0.7, 0.75)^{\eta_{MEC}} \quad (6)$$

where $\eta_{DG} = 0.128$ and $\eta_{MEC} = 1.88$ are concentration parameters and $N_x(\mu, \sigma)$ is the Gaussian function with mean $\mu$ and standard deviation $\sigma$.

In addition, $\varphi$ denotes the excitatory-inhibitory phase differential, which in the text is referred to by $\varphi_{inh}$ for clarity, and $A$ denotes the amplitude of the inhibitory modulation. These two parameters were systematically varied by simulating all combinations (250 steps of $A$ in the range [0, 5] and 1000 steps of $\varphi$ in the range [0°, 360°]). Gain coefficients $\gamma_{DG}$ and $\gamma_{MEC}$ were set to 0 to simulate DG and MEC lesion experiments, respectively. In addition, we also simulated that each of the excitatory inputs is 60% percent of the other (ratio: 75/125) while leaving the total excitation intact to show that the model is robust to substantial variations in the relative strength of the two excitatory inputs (Supplementary Fig. S10b).

**Spike generation.** The total drive obtained in the previous step was normalized to define a probability distribution and used as an intensity function for an inhomogeneous Poisson process to generate the spikes. The phase precession measurements were calculated for these simulated trains as described for the empirical data, but by assuming that the animal moves at a constant velocity.

**Mapping of model output to empirical data.** The phase precession measurements (slope, variance explained, onset phase, offset phase) obtained by simulating the model with various combinations of free parameters ($A$ and $\varphi_{inh}$) were individually compared to those obtained by analyzing the experimental data from control, DG-lesion, or MEC-lesion rats. To match the model inhibitory phase with theta phase in the empirical data, we aligned the maximal CA3 firing in the data with the time of minimum inhibition in the model (LFP phase = 180°). We confirmed that this alignment between the empirical and model LFP phase resulted in a broad range of $A$ and $\varphi_{inh}$ parameter combinations to reproduce data (Supplementary Fig. S9a). Each set of empirical phase precession measurements was compared to the phase precession measurements obtained from analyzing the corresponding model instantiation (CTRL to full model (no terms set to 0), LESION[(DG)] to "DG-lesion" model, and LESION[(MEC)] to "MEC-lesion" model). Each measurement from the model that was within the 80th percentile centered on the median of the empirical data was considered admissible (white regions in the binary plots of Fig. 7d). Free parameter combinations that included all four admissible measurements were accepted ('Overlap' plots, bottom panels in Fig. 7d). Finally, the accepted free parameters were compared between lesion and control

instantiations of the model by plotting the histogram of their distribution (Fig. 7e). To produce a sufficiently large sample size when determining $\Phi_{on}$ and $\Phi_{off}$ values that the model assumes within the allowable empirical $A$ / $\varphi_{inh}$ parameter space, we expanded the admissible space to correspond to the $90^{th}$ percentile centered on the median of empirical data.

### Reporting summary

Further information on research design is available in the Nature Portfolio Reporting Summary linked to this article.

## Data availability

Additional data that support the findings of this study are not in standardized formats but can be made available without restrictions upon request to the corresponding authors. Source data are provided in this paper.

## Code availability

All custom code for processing the data and for generating the model is freely available from a GitHub repository (https://github.com/cleibold/CA3phaseprecession) and from Zenodo (https://doi.org/10.5281/zenodo.13907008). Any additional information required to reanalyze the data reported in this paper is available from the corresponding authors upon request.

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

## Acknowledgements
We thank M. Wong and A.-L. Schlenner for their technical assistance. This work was supported by NIH grant R01 MH119179 and the Walter F. Heiligenberg Professorship to J.K.L., NIH grants R01 NS102915, R21 MH100354, R01 NS084324, and R01 NS097772 to S.L., and DFG (German Research Association) grant LE2250/13-1 to C.L. M.S. was supported by NIH training grant T32 AG 00216.

## Author contributions
S.A., S.L., and J.K.L. conceived experiments, designed study, and interpreted data. S.L. and J.K.L. managed the project. T.S. and M.S. prepared and provided data from previously published experimental work. S.A. analyzed data. S.A. created the computational model in collaboration with C.L and S.L. Figures were prepared by S.A. and J.K.L., and S.A., S.L., and J.K.L. wrote the manuscript with feedback and editing from C.L.

## Competing interests
The authors declare no competing interests.
