## [Transparent Peer Review file · Nature Communications]

Distinct roles of dentate gyrus and medial entorhinal cortex inputs for phase precession and temporal correlations in the hippocampal CA3 area

Corresponding Author: Dr Jill Leutgeb

Version 0:

Reviewer comments:

Reviewer #1

(Remarks to the Author)

The manuscript titled, 'Distinct roles of dentate gyrus and medial entorhinal cortex inputs for phase precession and temporal correlations in the hippocampal CA3 area' describes new empirical and modeling results based on reanalysis of existing data. The data consist of recordings of spiking activity from hippocampal region CA3 in behaving rats performing spatial memory tasks with or without medial entorhinal (MEC) or dentate gyrus (DG) lesions. The key empirical results include: 1) demonstration that both types of lesions reduce CA3 theta phase precession, 2) DG but not MEC lesions disrupt the timing of the first spikes in a sequence, and 3) the phase precession disruption observed following MEC but not DG lesions can be mitigated by alternate analysis methods that are less sensitive to the precise timing of individual spikes. The key modeling results, obtained by examining the patterns of parameters which could account for the empirical data, provide evidence that DG input modulate the timing of spiking with respect to the theta cycle and that MEC input modulates the magnitude of inhibition. Together, this set of results substantially advances what is known about the mechanisms of CA3 phase precession and the functional importance of the DG.

Review summary: I am very enthusiastic about this manuscript. The results are of high scientific significance, the data provide strong support for the conclusions, and the presentation is clear. While I do believe there are issues that require attention before it is ready for publication, I expect they are addressable. In the remainder of the review I unpack why I see this work to be significant under Strengths, list the issues that I see as critical to address in Concerns, and have noted additional relatively minor points under Minor concerns.

Strengths: There are two primary reasons that I believe this work is of high scientific significance. The first is the mechanistic insights it brings to our understanding of CA3 theta phase precession. Theta phase precession has long held the attention for many interested in hippocampal physiology and/or neural information processing. It is widely hypothesized to have high functional importance (e.g., for memory encoding). There are also numerous hypotheses regarding the mechanistic basis of theta phase precession. The present work provides strong constraining data relevant for understanding the mechanistic basis by showing the distinct but consequential contributions of the DG and MEC to CA3 phase precession. This advancement in mechanistic understanding, in turn, will be constraining and informative for new theories of the functional interpretations of phase precession. The second reason that I believe this work is of high scientific significance hinges on the apparent functional role the current results ascribe to DG. In the broad scheme of hippocampal research, relatively little is known about the DG and what role is attributed to it has been fairly constrained to its putative importance for 'pattern separation' in mnemonic processing. I agree with the authors' statements that these new results demonstrate a broader importance of DG processing in hippocampal processing in that the intact DG is necessary for CA3 temporal ordering of spatially tuned neurons. While it remains to be shown that this specific contribution is necessary for intact hippocampal function, it is reasonable to presume that it does.

Weaknesses:

The primary concern impeding my overall enthusiasm for this manuscript is that an alternate hypothesis goes unacknowledged and unaddressed: the possibility that the reduction of phase precession, for example after the DG lesion, is the result of the loss of excitatory input that reduces the spatial range over which the neuron fires. By this perspective, the

observed spiking in the lesion condition could be just the late portion of the spatial field, a phase which ordinarily has weak phase locking and weak intrinsic phase precession. Falsifying this hypothesis warrants more attention. While there are likely better approaches, the first question that came to my mind to dissociate this alternative hypothesis with the reduced inhibition hypothesis put forward in the manuscript was to consider the spatial extent of the sequences. The logic being that reduced inhibition might extend the spatial extent of the firing whereas reduced excitation might reduce the spatial extent.

A second important point to address is the units of analysis used in the statistics. Most crucially, at no point is rat the unit of analysis, instead it is typically cell. The challenge with this is that it fails to address the reliability of a statement regarding the likelihood that lesioning the DG, for example, will have effect X. It is technically possible, for example, that one rat yielded an effect and contributed a sufficient number of cells to the overall analysis to drive the main effect. There are two specific occasions where the unit of analysis caught my attention. The first is in the analysis of 'first spike in a train.' As I understand it, the analysis was performed by computing the most likely phase of first spike for each cell and then examining the distribution of those phases. This is arguably relevant, but I don't see it being as relevant as asking if the distribution of phases across sequences for each cell increases. By this alternate framing, one would compute the dispersion or circular variance of the phase of first spike across sequences for each cell separately and then perform the hypothesis testing on the distribution of dispersion scores. The second is in the analysis of mean phase of spiking (~line 252) where individual spikes were taken as the unit of analysis, yielding values of n in the tens of thousands. Each spike is not independent, a core assumption in any statistical analysis. Again, while 'rat' would be the most conservative level of binning, already, 'cell' would be far superior and more interpretable than 'spike.'

Finally, no attention is given to possible changes in the theta waveform and the consequences of this on the metrics of interest. It is well established that theta is not sinusoidal and asymmetries in the wave can impact resulting analysis linking to phase (Belluscio et al., 2012). Thus, it stands to reason that lesions of upstream inputs may have changed the waveform and that those changes could impact the phase precession values. Methods exist for controlling for this, testing that the conclusions do not change (or documentation of how they do) when those are applied would strengthen this work.

Minor concerns:

Statements included in the results with no empirical support should be moved to the discussion or removed entirely. Two examples include, 1) the statement at lines 176-178 concerning the importance of DG input for generating sufficient phase precession to support 'real-time encoding and retrieval of episodic memories' is applying more interpretation than the data warrant given that phase precession persisted though weakened and no empirical evidence was provided regarding either encoding or retrieval of memories, and 2) statement at lines 239-241 regarding DG being essential for generating intrinsic 'look-ahead' spikes is too strong given that no overt analysis of look ahead was performed. While both examples are valid connections to existing literature, they are not factual accounts of the data and insert a fair amount of interpretation and require a fair amount of inductive reasoning and thus better belong in the discussion. Relatedly, the concluding sentence of the manuscript says that the present work showed that emergence of look ahead depends on DG but without concrete clear demonstration.

The analysis of number of spikes in the first full theta cycle (starting ~line 259) was troubling for two reasons. First, the effect is driven by a difference in the DG control condition rather than the experimental condition. The DG lesion, MEC lesion and MEC control groups were all effectively the same. Second, the DG control condition had a mean of less than 1, suggesting it is possible to have the first theta cycle in the sequence have less than 1 spike. But I struggle to understand how it is considered as the first cycle of the sequence if there is not yet spiking. I looked to clarify in the methods regarding how this cycle was defined but failed to find the relevant method.

No attention is given to the fact that lesions of the MEC will also impact the DG.

Does the model account for the differential effects of the lesion types on phase of first spike and last spike? Overt description of this in either case would help. Inclusion of this in the figure would too.

Line 112 – clarify if bouts are defined on a per-cell basis or population basis?

Line 359 – Venditto et al., 2019 examined CA1 sequences in the context of reduced phase precession

Line 975 – specific measure of variance not defined, inclusion of equation or citation would be more transparent

Reviewer #2

(Remarks to the Author)

Distinct roles of dentate gyrus and medial entorhinal cortex inputs for phase precession and temporal correlations in the hippocampal CA3 area

Siavash Ahmadi¹, Takuya Sasaki^{1,2}, Marta Sabariego¹, Christian Leibold³, Stefan Leutgeb^{1,4,*}, Jill K. Leutgeb^{1,4*}
Nature Communications

In this paper Ahmadi et al. re-analyze 2 previously published dataset, exploring the contributions of DG and MEC to CA3 activity at the fine-scale temporal level. They provide a computational model that accounts for the role DG and MEC inhibitory subnetworks in generating phase precession and sequences in CA3. The investigation relies on solid premises.

The manuscript is well written.

“line 134 ... it is not clear whether DG inputs are necessary for phase precession in its direct target cells in CA3.” I appreciate the premises of the investigation. There is a need for studying temporal coding at the single cell and population levels in other hippocampal subfields aside CA1.

Few major points that need to be addressed:

Major points:

- I understand details of the lesions have been reported in previously published papers, but there is the need for including histology picture here as well (Figure 1 and 2). Readers need to understand the level of damage of the lesions. This directly connects to the following point.
- Both MEC and DG are layered structures. Each layer contributes specifically to local and up-/down-stream computations. The major MEC lesions reported (95.3% of layer II, 92.4% of 824 layer III, and 91.4% of deep layers) interrupt both direct connections to CA1 and the classic tri-synaptic circuit. The computational model and the discussion are based on careful and delicate assumptions on the anatomy of the circuitry, but the same level of detail is not mirrored in the intervention. “The mechanisms by which DG and MEC circuits contribute to the organization of the precise temporal profiles of CA3 spiking (Fig. 5e-f) are therefore distinct and are consistent with a model in which DG inputs effectively restrict spiking to late phases of the theta cycles early in the spike train of a CA3 neuron (i.e., upon entry into the place field) (Fig. 5e). In contrast, later in the spike train (i.e., in the middle and near the exit from the place field), MEC inputs appear to ensure an appropriate mean theta phase of CA3 spiking by driving CA3 neurons in time windows around a monotonically decreasing mean theta phase over successive theta cycles” This is a key statement for this paper. But again, it is difficult to exclude unwanted effects of the lesions - given that they destroy the majority of the MEC. It would be nice if the authors can elaborate on this.
- The choice of using spike-trains without previously identifying place fields is well explained. However, at least in one of the previously published papers (Sasaki et al. 2018, figure 6 and 7) Place Fields were well isolated. Does a strict selection of place field yield any differences in the analysis of figure 3,4 and 5 of the manuscript? Reports of place field dynamics (rate code) are also interesting, when comparing the two manipulations.
- LFP analysis are missing. At least one (supplementary) figure needs to address changes in theta, slow gamma and fast gamma parameters (θ frequency and amplitude, mean phase preference of slow and fast gammas), comparing the 2 datasets and lesions. Measures should also be normalized by running speed. This should be considered as a control measure to rule out that changes might derive from alterations of the theta rhythm.
- On a broader level of discussion: if DG contributes as well to sequence coding and future planning as proposed by the authors (in line 566) what are key differences with the CA3-CA1 circuit (Treves and Rolls, 1992)?
- The main behavioral effects of the lesions are already reported in the papers published previously with these dataset (Sasaki et al. 2018, Sabariego et al. 2019). Is it possible to relate behavioral performances with either CA3 phase precession or temporal correlations of CA3 cells in DG vs MEC lesions?

Minor points:

- Line 101-130: having a clear description of the methodological details used to analyze the data is extremely useful; however, I would suggest moving portions of this sections to the method part, as the flow of the section itself would benefit.
- Line 290 “Putative granule cells exhibited a narrow theta phase preference at the onset of spiking”. Supplementary figure S3a,b,c should be mentioned before Figure 4. This section can be included in the previous paragraph.
- Figure 7 Panel d: plots are not easy to read. I understood the idea to show and compare phase shifts, but maybe different color coding can be used (or number of panels decreased).
- FigureS3: how are place field defined here? After the selection of spike-trains or in the classical way?

Reviewer #3

(Remarks to the Author)

While the existence of theta phase precession has been known for ~30 years, its functional role in cognitive function and cellular mechanisms supporting it still remain largely unknown. Therefore, empirical data supporting its role and mechanisms are of importance to the field. In this manuscript, the authors focused on the mechanism of theta phase precession in the hippocampal CA3 cells, and demonstrated different roles of DG and MEC inputs to CA3 on phase precession. They report that DG input contributes to the phase of CA3 spikes especially at the initial part of a place field, while the MEC input contributes to the precision of the phase throughout the place field. In addition, while DG input contributed to theta sequence, MEC input did not. Using a computational model based on oscillatory interference, the authors demonstrated that DG lesion could have involved a shift of inhibitory oscillatory input, while the MEC lesion could have involved a decrease in the amplitude of inhibitory oscillatory input. While these findings are novel and interesting, both the functional relevance and mechanisms supporting these differences remain unclear, limiting the significance of this study.

Major:

As the authors pointed out, it has been known that MEC input to CA1 is necessary for CA1 phase precession. It is therefore not surprising to find that DG and MEC inputs to CA3 are necessary for CA3 phase precession.

While it is interesting that DG input and MEC input differently affect phase precession, there is no empirical insight as to how these two areas exert different effects. The model points to the inhibition phase and amplitudes. However, this is just that the model will not fit the data without these modifications. The authors point to the potential difference of somatic (DG) vs dendritic (MEC) suppressions as the potential source of these differences, which is interesting. However, there is no empirical data backing this up. Without empirical data supporting this, there is no practical advance regarding the phase precession mechanism. It is necessary to conduct, for example, specific manipulation of these two types of inhibitory circuit to support this view. Without empirical observations, this manuscript remains to be descriptive.

To support the hypothesis that inhibitory input is differently modulated by the DG and MEC lesions, authors could have attempted to quantify the activity of putative inhibitory cells (rate and timing). Or, cross correlations of spike timings between putative excitatory and inhibitory cells could be used in addition to describe possible effects on the inhibitory network.

The recurrent excitatory network in CA3 would be an important candidate as a mechanism on phase precession as the authors mention in the manuscript. Although they claim that the recurrent CA3 network by itself is not sufficient based on the disrupted phase precession in the case of DG and MEC lesions, it is self-evident that recurrent network without input to CA3 cannot support phase precession. Therefore, whether the recurrent network plays a crucial role or not needs to be tested by, for example, suppressing specifically the recurrent network. Furthermore, their computational model did not include this recurrent network. In summary, although the recurrent network would be a strong candidate mechanism supporting phase precession as suggested by other researchers, this study does not provide insight into this potentially important mechanism.

It is not clear what the different effect of the MEC vs DG lesions on phase precession means in terms of memory function. Although the authors discuss the possibility that DG is more important for sequence in the discussion, the fact that MEC is upstream of both CA3 and DG, and the fact that MEC cells show phase precession speaks against the idea that DG is more important for sequence than the MEC. Empirical data backing up the importance of the different effect of DG vs MEC lesions, for example a specific DG lesion impairing CA3 dependent sequential tasks, is needed to reveal the importance of this difference.

The different effect of DG vs MEC lesions could reflect differences in the magnitude of decreased excitability. MEC input synapses on the more distal part of dendrites than the DG input. Therefore, the MEC lesion might have had a weaker effect on the reduction of excitatory input. This possibility should be tested for example by looking into the change of excitatory cell firing. In addition, this possibility could be tested in the computational model by changing the magnitude of DG or MEC input.

The computational models are often used to provide an explanation for the empirical data. In contrast, the model provided in this manuscript rather demonstrates that the model does not fit to the observed data unless additional changes in the inhibitory circuit are introduced. While these required changes of inhibitory input (A and phy_inf) could indeed indicate additional changes that occurred associated with DG and MEC lesions, it should first be considered that the model could be improved. There might be other models which could explain the data without considering additional inhibitory components. An alternative approach could be to implement a hypothetical inhibitory circuit that receives input from the DG or MEC, and then simulate lesioned conditions to show that inclusion of a specific inhibitory circuit can explain the empirical data for a wide range of parameters.

Minor:

The introduction section does not explain the aim of study very well. For example, it remains unclear why it was important to compare DG vs MEC lesions. Also, the authors spent much text on replay which was not the focus of this manuscript. On the other hand, the introduction of the known and proposed mechanisms of phase precession is rather scarce.

Fig3a and c: sorting the order of cells using their mean phase might make sense?

Fig4 will benefit from example spike trains that are distributed to a wide range of theta phase in DG lesion rats, as opposed to late narrower range in control.

Fig7d and f should be presented and explained better. The four columns of panels should be clearly indicated on the figure that they correspond to the onset phase, offset phase, etc. Also, onset phase is written as phy_o here but phy_on in Fig7b? In addition, in the panels with DG lesion, what are the black (not pink) dots?

It would be better if the model can provide a possible mechanism as to how relative timing of place cells can be preserved in the case of MEC lesion.

Version 1:

Reviewer comments:

Reviewer #1

(Remarks to the Author)

The results of this revised manuscript, through much enriched, largely support the same set of conclusions as the initial submission. This new version is an extremely nice piece of work. I only had three concerns that I considered to be substantial before and each have been thoroughly addressed. Incidentally, I had also raised a number of what I called 'minor concerns' and these, too, were beautifully handled. I was enthusiastic about this work already and that enthusiasm has only grown. I have no hesitations in recommending that this manuscript be advanced to publication.

(Remarks on code availability)

Reviewer #2

(Remarks to the Author)

I want to compliment the authors for carefully addressing all my concerns, with a set of extensive reviews. The manuscript is significantly clearer and improved in quality. I don't have any further comments and, in my opinion, the paper is now suitable for publication.

(Remarks on code availability)

Reviewer #3

(Remarks to the Author)

The key observation of this work is that MEC and DG differently affect phase precession in CA3. Unfortunately, the revision could not provide any additional support for the proposed mechanism behind this difference. The authors responded to my major comment #2 that the data analysis remains descriptive. In addition, it is difficult to eliminate the possibility that this main observation stems from a different extent of the lesions as pointed out by my major comment 6 as well as by the reviewer 1. Their additional analysis on this aspect indicates that the MEC lesion might have affected the activity of CA3 more strongly, suggesting different extent of the lesions. Together with the lack of correlation between phase precession change and behaviour, scientific advance made by this work might be limited. Therefore, my main concerns were not resolved overall.

Major comment 3: "To support the hypothesis that inhibitory input ..."

It is unfortunate that there are not enough putative interneurons to analyse. However, there are multiple papers which analysed putative interneurons recorded from the electrodes in the CA3 pyramidal cell layer. In this sense, I must point out that it is a short-coming of the current work that the main mechanisms the authors demonstrate from the model is relying on recordings they don't have. In my opinion, such recordings will be a minimum requirement to suggest inhibition as the main mechanisms underlying different phase precession they observe.

Major comment 4: "The recurrent excitatory network in CA3..."

It might be ok not to include CA3 recurrent network if empirical data of the authors suggest the importance of inhibition. However, that is not the case here. In this case, a more convincing modelling approach would be to test multiple known models and come to a conclusion that inhibition can but recurrent excitation, for example, cannot explain their data.

Major comment 5: "It is not clear what ..."

Unfortunately, the revision could not improve this aspect and there still is no link between lesion types causing different phase precession deficits and behaviour.

Major comment 6: "The different effect of DG vs MEC lesions"

Their additional analysis on this was useful and additional simulation of different excitation strength strengthened the model. However, I think the larger effect of MEC lesion they found could have caused a larger effect of MEC lesion on phase precession (affecting entire phase).

Major comment 7: "The computational models are..."

Please see my comment on the comment #4 above.

(Remarks on code availability)

Point-by-point response to reviewers, with reviewer comments retained in black and our responses added in blue. A manuscript copy with key revisions highlighted in red is included with the resubmission.

We thank the three reviewers for their time and expertise and for the positive feedback on our manuscript, in particular for mentioning that ‘this set of results substantially advances what is known about the mechanisms of CA3 phase precession’, ‘the investigation relies on solid premises’, ‘the manuscript is well written’ and ‘these findings are novel and interesting’. We appreciate the detailed set of comments and have thoroughly addressed each concern, as described in detail in the point-by-point responses.

We apologize for the long turn-around time. This was in part a consequence of needing to coordinate with a first-author who has moved from a PhD to a postdoc position, but more importantly, emerged from the need and implementation of extensive revisions of the data analysis and the modeling portions.

-In response to reviewer comments, we expanded many of the main figures (e.g., panels c and f in Fig. 3) and increased the number of supplementary figures from 5 to 11. Taken together, the revisions approximately double the extent of previously already comprehensive data analyses. For example, new measurements of phase dispersion are added, and all of the key analyses that were previously presented for spike trains are now also included for standard place fields. Our revisions also include extensive new LFP analyses of recording data that were requested by a reviewer and are now shown in new supplementary figures (Fig. S4 and S5).

-Although the model has remained conceptually the same as in the original manuscript, we have much more comprehensively analyzed the parameter space and have included numerous new series of plots such as of the phase difference between the empirical LFP and the model inhibition, of the phase difference between the two excitatory input pathways, of different levels of inhibition, and of different levels of excitation. These extensive new surveys of model parameters have resulted in three new supplementary figures (Supplementary Figs. 9, 10, and 11) in addition to major updates to main Fig. 7. To make these new, comprehensive analyses feasible, we had to substantially reduce the run time for model implementation, which required us to completely rewrite the code for the model in Python. We will of course make the code available to the research community and anticipate that the code will not only have been useful for the additional analysis that is reported here, but will also be used widely by others.

Reviewer #1 (Remarks to the Author):

The manuscript titled, ‘Distinct roles of dentate gyrus and medial entorhinal cortex inputs for phase precession and temporal correlations in the hippocampal CA3 area’ describes new empirical and modeling results based on reanalysis of existing data. The data consist of recordings of spiking activity from hippocampal region CA3 in behaving rats performing spatial memory tasks with or without medial entorhinal (MEC) or dentate gyrus (DG) lesions. The key empirical results include: 1) demonstration that both types of lesions reduce CA3 theta phase precession, 2) DG but not MEC lesions disrupt the timing of the first spikes in a sequence, and 3) the phase precession disruption observed following MEC but not DG lesions can be mitigated by alternate analysis methods that are less sensitive to the precise timing of

individual spikes. The key modeling results, obtained by examining the patterns of parameters which could account for the empirical data, provide evidence that DG input modulate the timing of spiking with respect to the theta cycle and that MEC input modulates the magnitude of inhibition. Together, this set of results substantially advances what is known about the mechanisms of CA3 phase precession and the functional importance of the DG.

Review summary: I am very enthusiastic about this manuscript. The results are of high scientific significance, the data provide strong support for the conclusions, and the presentation is clear. While I do believe there are issues that require attention before it is ready for publication, I expect they are addressable. In the remainder of the review I unpack why I see this work to be significant under Strengths, list the issues that I see as critical to address in Concerns, and have noted additional relatively minor points under Minor concerns.

Strengths: There are two primary reasons that I believe this work is of high scientific significance.

The first is the mechanistic insights it brings to our understanding of CA3 theta phase precession. Theta phase precession has long held the attention for many interested in hippocampal physiology and/or neural information processing. It is widely hypothesized to have high functional importance (e.g., for memory encoding). There are also numerous hypotheses regarding the mechanistic basis of theta phase precession. The present work provides strong constraining data relevant for understanding the mechanistic basis by showing the distinct but consequential contributions of the DG and MEC to CA3 phase precession. This advancement in mechanistic understanding, in turn, will be constraining and informative for new theories of the functional interpretations of phase precession.

The second reason that I believe this work is of high scientific significance hinges on the apparent functional role the current results ascribed to DG. In the broad scheme of hippocampal research, relatively little is known about the DG and what role is attributed to it has been fairly constrained to its putative importance for 'pattern separation' in mnemonic processing. I agree with the authors' statements that these new results demonstrate a broader importance of DG processing in hippocampal processing in that the intact DG is necessary for CA3 temporal ordering of spatially tuned neurons. While it remains to be shown that this specific contribution is necessary for intact hippocampal function, it is reasonable to presume that it does.

We appreciate the recognition of the high significance, and the elegant summary of the two main strengths of our research.

Weaknesses:

(1) The primary concern impeding my overall enthusiasm for this manuscript is that an alternate hypothesis goes unacknowledged and unaddressed: the possibility that the reduction of phase precession, for example after the DG lesion, is the result of the loss of excitatory input that reduces the spatial range over which the neuron fires. By this perspective, the observed spiking in the lesion condition could be just the late portion of the spatial field, a phase which ordinarily has weak phase locking and weak intrinsic phase precession. Falsifying this hypothesis warrants more attention. While there are likely better approaches, the first question that came to my mind to dissociate this alternative hypothesis with the reduced inhibition hypothesis put forward in the manuscript was to consider the spatial extent of the sequences. The logic being that reduced inhibition might extend the spatial extent of the firing whereas reduced excitation might reduce the spatial extent.

We agree that the loss of excitatory input to CA3, when lesioning the majority of DG or MEC neurons, could result in reduced excitation, and as a consequence, in smaller fields that may correspond to only the late portion of the field. Given that any excitatory inputs also provide feedforward and feedback inhibition, it seems reasonable to also expect that any reduced excitation goes along with reduced inhibition. The precise balance of these effects will obviously determine whether there is ultimately some degree of excess excitation or inhibition with the lesions. We agree that measurements of firing rates and of field sizes of principal cells in the target area (i.e., CA3) are perhaps the most obvious readout to determine whether lesion effects are net excitatory or inhibitory. The hypothesis that there is a net reduction of inhibition would predict higher rates and larger fields, while the alternative hypothesis that there is a net reduction of excitation would predict lower rates and smaller firing fields.

To address this point, we updated Figure S3 (previously Figure S2) to not only show comparisons between the two control groups, but now also comparisons of each of the control groups with the corresponding lesion group. Compared to controls, DG lesions did not alter firing rate, sparsity, or spatial information, but decreased selectivity and yielded longer path lengths during trains. On balance, there is therefore not a major change in excitation compared to inhibition, but if there are any effects, they are generally in the direction of broader fields. The results are therefore consistent with the notion of net reduction in inhibition, which makes it unlikely that only the late portion of the spatial field is observed in the lesion condition. To address this point explicitly in the manuscript, we expanded the text to now include a more detailed description of the effects of lesions on firing rate and spatial measurements.

Line 129-141:

‘Furthermore, we also examined whether lesioning a large proportion of DG or MEC inputs to CA3, which are each excitatory, had major effects on firing rates and spatial firing characteristics. Compared to controls, DG lesions did not alter firing rate, sparsity, or spatial information, but decreased selectivity and yielded longer path lengths during trains (Fig. S3b). On balance, there is therefore not a major overall change in excitation relative to inhibition, but if there are any effects, they are generally in the direction of broader fields. The results are therefore not consistent with the notion of a merely reduced excitation, such that only the late (i.e., higher rate) portion of the spatial field is observed in the lesion condition. Performing the same analyses for MEC lesions compared to controls yielded lower selectivity and spatial information, longer path lengths during trains and increased sparsity (Fig. S3c). Again, the less precise spatial firing is inconsistent with the notion that only a portion of the firing field is retained. However, the minor decrease in firing rate with MEC lesions suggests that loss of excitation may be more predominant with MEC lesions compared to DG lesions.’

In addition to extending the figure panels describing the differences between control and lesion data from experiments (Fig. S3), analyses of model parameters showed that the model is surprisingly robust to the level of inhibitory DC component (Fig. S10a). In the current implementation of the model, we therefore considered a baseline level of inhibition as one of the least relevant parameters and do not use an inhibitory DC component. Furthermore, the $A-\phi_{inh}$ parameter space that is depicted in Fig. 7 shows that the model can robustly generate the phase precession that is observed in control data with a wide range of inhibitory oscillatory amplitudes (i.e., parameter A). Of course, in model variants that omit either DG or MEC input (‘lesions’), the excitation is substantially lower such that lower levels of inhibition (A) are permissible for reproducing experimental data. To account for the finding that lower levels of excitation are balanced by lower levels of inhibition, the example model data in Fig. 7c use lower A values (i.e., 2 instead of 3) for the lesion condition compared to the control condition.

(2) A second important point to address is the units of analysis used in the statistics. Most crucially, at no point is rat the unit of analysis, instead it is typically cell. The challenge with this is that it fails to address the reliability of a statement regarding the likelihood that lesioning the DG, for example, will have effect X. It is technically possible, for example, that one rat yielded an effect and contributed a sufficient number of cells to the overall analysis to drive the main effect.

There are two specific occasions where the unit of analysis caught my attention.

(2.1) The first is in the analysis of ‘first spike in a train.’ As I understand it, the analysis was performed by computing the most likely phase of first spike for each cell and then examining the distribution of those phases. This is arguably relevant, but I don’t see it being as relevant as asking if the distribution of phases across sequences for each cell increases. By this alternate framing, one would compute the dispersion or circular variance of the phase of first spike across sequences for each cell separately and then perform the hypothesis testing on the distribution of dispersion scores.

The comment raises concerns about the unit of analysis as well as the method that was used for calculating the dispersion. These two issues are addressed as described in (a) and (b).

(a) With regard to the unit of analysis in the calculation of dispersion, we stated in lines 216-219 of the original manuscript that

‘..., we calculated the onset and offset theta phase of CA3 spike trains. The onset phase of trains – defined as the circular median phase of the spikes in the first cycle of each train – no longer consistently occurred at late phases in CA3 cells of DG-lesioned rats compared to control rats (Fig. 3a).’

The original method is characterized by the reviewer as ‘relevant’ and is therefore retained. However, we may previously have not been sufficiently clear in pointing out that the unit of analysis was the cell and that the sampling of cells was distributed over many rats (Table S1). In particular, we first calculated the circular mean of first-cycle spikes of each train and then – **for each cell** – took the median over all of the cell’s trains (as shown in Fig. 3a and in the dispersion of these ‘medians by cell’ in Fig. 3b).

To more explicitly report that the unit of analysis was the cell, the statement is now updated, so that it is apparent that the phase was averaged over all spike trains of a cell before performing the statistics.

Line 254-257:

‘we calculated the onset and offset theta phase of CA3 spike trains. The onset phase of trains – defined by first calculating the circular mean of first-cycle spikes of each train and by then taking the median over all of the cell’s trains – no longer consistently occurred at late phases in CA3 cells of DG-lesion rats (Fig. 3a,b).’

In the figure legend, we did state that the median onset phase across spike trains of a cell is shown in Fig. 3a, but again, did not explicitly state that these values were used for sorting and are the values that were used for the statistics in Fig. 3b.

This is now updated in the Fig. 3 legend, which includes these statements:

‘Cells within each panel are sorted from top to bottom by their median onset/offset phase.’

'b, The cells' median onset and offset phases (shown in blue in panel a) were compared between control and DG-lesion rats.'

In the methods, we only mentioned that the mean spike phase of the first cycle was taken. This is now updated to also state that we then calculated the median over all trains of each cell, which results in one value per cell.

Line 1065-1066:

'For statistical analysis of onset and offset phases, the median of the mean phases of each cell's trains was taken, which resulted in one value per cell.'

To better report the number of rats and the distribution of recorded units across rats, we now report that the rat and session numbers refer to the corresponding table (Table S1).

Line 873-877:

'We reanalyzed and compared CA3 activity patterns from these data, including a total of 31 rats (Table S1). These included 4 control and 9 dentate-lesion rats (7 and 16 sessions; DG lesion experiment) with CA3 or dual CA3-DG (2 of the 4 control rats) single-unit recordings, 7 control and 8 MEC-lesion rats (18 and 20 sessions; MEC lesion experiment) with CA3 single-unit recordings, and 3 control rats with only DG recordings.'

(b) We added the analysis that is suggested by the reviewer as even more relevant, **and note that this is an insightful comment because the new analysis makes the intended point much more directly**. We calculated the dispersion of the onset phase across each cell's trains as suggested (i.e., by taking the mean phase of spikes in the first theta cycle of each train and by then computing the dispersion of phases across the cell's trains). The corresponding analysis was also performed for the offset phases by using the spikes in the last theta cycle of each of the cell's trains.

The figure panels from this analysis are included below for inspection and are included as new figure panels (c and f) in Fig. 3.

As predicted by the reviewer, these analyses are well suited for detecting changes in the dispersion of onset and offset phases. The dispersions were strikingly increased for the first spikes, but not for last spikes with DG lesions (onset: z-statistic = -4.11, $U = 4187$, $p = 3.9 \times 10^{-5}$; offset: z-statistic = -1.33, $U = 4564$, $p = 0.18$, MW tests). For MEC lesions, the dispersions of both the first and last spikes were moderately increased (onset: z-statistic = -2.30, $U = 9879$, $p = 0.021$; offset: z-statistic = -2.16, $U = 10230$, $p = 0.031$, MW tests), which is consistent with the overall increase in variability of phase precession that is also detected with other analyses. This analysis therefore further confirms the selective effect of DG lesions only on the onset timing and is now described in the main text.

Line 269-274:

'Selective effects on the timing of the onset phases, but not of the offset phases were further confirmed by measuring the dispersion of the phases of the first spikes and of the phases of the last spikes across spike trains of each cell. With DG lesions, the dispersion of the first spikes, but not of the last spikes increased (Fig. 3c; first: z-statistic = -4.11, U = 4187, p = 3.9 × 10⁻⁵; last: z-statistic = -1.33, U = 4564, p = 0.18, MW tests).'

Line 282-285:

'Furthermore, effects on the dispersions of first and last spikes across spike trains were moderate and approximately matched (Fig. 3f; first: z-statistic = -2.30, U = 9879, p = 0.021; last: z-statistic = -2.16, U = 10230, p = 0.031, MW tests), which is consistent with an overall increase in variability in theta phase preference.'

Line 288-291:

'These results were also confirmed when repeating the same analyses by using the cells' place fields rather than the cells' spike trains (Fig. S7), which confirms that qualitative differences in the effect of DG and MEC lesions on temporal firing patterns in CA3 cannot be attributed to methodological details.'

Legend of Fig. 3c (line 726-741):

*'c, The dispersion of the phases of the first and of the phases of the last spike was calculated across spike trains of each cell, and the cells' dispersions were compared between control and DG-lesion rats. Dispersions of the cells' onset phases but not of the cells' offset phases increased with DG lesions (onset: z-statistic = -4.11, U = 4187, p = 3.9 × 10⁻⁵; offset: z-statistic = -1.33, U = 4564, p = 0.18, MW tests). d-f, As (a-c), but for the cells from MEC-lesion rats (LESION^(MEC)) and their corresponding controls (CTRL^(MEC)). Onset phase values again peaked in the ascending phase of the theta cycle, but their distribution was not altered by the lesion (n = 101 CTRL^(MEC) and 158 LESION^(MEC) cells, $\chi^2 = 1.18$, p = 0.56, circular MANOVA; phase concentration: CTRL^(MEC) $\kappa = 1.49$, LESION^(MEC) $\kappa = 1.33$, U = 0.44, p = 0.51, concentration test). Similarly, the distribution of offset phases did not differ between cells of MEC-lesion and control rats ($\chi^2 = 2.18$, p = 0.34, circular MANOVA; concentration: CTRL^(MEC) $\kappa = 1.08$, LESION^(MEC) $\kappa = 1.16$, U = 0.15, p = 0.70, concentration test). Dispersions of onset and offset phases were moderately broadened by MEC lesions (onset: z-statistic = -2.30, U = 9879, p = 0.021; offset: z-statistic = -2.16, U = 10230, p = 0.031, MW tests), which is consistent with an overall increase in variability in theta phase preference. n.s., not significant, *** p < 0.001.'*

(2.2) The second is in the analysis of mean phase of spiking (~line 252) where individual spikes were taken as the unit of analysis, yielding values of n in the tens of thousands. Each spike is not independent, a core assumption in any statistical analysis. Again, while 'rat' would be the most conservative level of binning, already, 'cell' would be far superior and more interpretable than 'spike.'

This refers to the data in Fig. 4a, where we plot the fraction of spikes early in the theta cycle. We agree that this should not have been reported with the number of spikes as the unit of analysis. We now first calculated the fractions by averaging over the spikes of each cell and by then performing a comparison across groups **with cells as the unit of analysis**. With the analyses appropriately powered, we observed that only DG lesions resulted in a redistribution of firing phases of CA3 cells. The updated statistics are now included in the main text and in the Fig. 4 legend.

Line 299-305:

'Spike phase distributions across theta cycles shifted towards earlier phases of the theta cycle with DG lesions while the shift was in the opposite direction with MEC lesions compared to their respective controls (Fig. 4a; mean theta phase: CTRL^(DG) n = 84 cells, mean phase = 173.4°, LESION^(DG) n = 68 cells, mean phase = 74.1°, χ^2 statistic = 18.4, p = 0.0001; CTRL^(MEC) n = 101 cells, mean phase = 157.1°, LESION^(MEC) n = 158 cells, mean phase = 172.3°, χ^2 statistic = 0.74, p = 0.69; χ^2 test for proportion of spikes contained in each of the three theta bins).'

Figure legend of Fig. 4 (line 743-767):

'Figure 4. With reduced DG input, peak probability of CA3 spiking shifts to earlier theta phases. a, Left, For control CA3 cells, distribution of spike incidence across early, middle, and late theta phases. The middle third of the theta cycle contains more spikes than early or late phases. Top right, The proportion of CA3 spikes early in the theta cycle was markedly increased with DG lesions (mean theta phase: CTRL^(DG) n = 84 cells, mean phase = 173.4°, LESION^(DG) n = 68 cells, mean phase = 74.1°, χ^2 statistic = 18.4, p = 0.0001, χ^2 test for proportion of spikes in each of the three thirds). Dotted lines, data from control CA3 cells, as shown to the left. Bottom right, with MEC lesions (LESION^(MEC)), there was no change in the phase distribution of CA3 spikes (CTRL^(MEC) n = 101 cells, mean phase = 157.1°, LESION^(MEC) n = 158 cells, mean phase = 172.3°, χ^2 statistic = 0.74, p = 0.69). b, Average number of spikes over a 360°-cycle from the first spike in the train. In trains from cells of DG-lesion rats, the median number of initial-cycle spikes increased 23.8% compared to control cells (from 2.61 to 3.23, n = 84 CTRL^(DG) and 68 LESION^(DG) cells, z-statistic = -4.38, U = 5243, p = 1.2 × 10⁻⁵, MW test). In MEC-lesion rats, the median number of initial-cycle spikes increased by 16.5% compared to control trains (from 2.49 to 2.9 spikes, n = 101 CTRL^(MEC) and 158 LESION^(MEC) cells, z-statistic = -2.74, U = 11516, p = 0.006, MW test). c, Fraction of CA3 spikes at early, middle, and late phases of the theta cycle as a function of normalized (%) distance during the train. At the onset of control trains, a low proportion of spikes is typically observed at early phases, but this proportion increased with DG lesions. d, Top, DG lesions selectively increased the variability of spike phase in the first half of a spike train (from 5% to 45% of normalized distance, MW tests; Holm-Bonferroni corrected; see Table S2 for statistics). See Fig. 1b for example spike trains that show this effect. Bottom, MEC lesions did not increase the variability of theta phase along the distance through the field (MW tests; Holm-Bonferroni corrected; Table S2). * p < 0.05, ** p < 0.01, * p < 0.001. For most theta cycle-based analyses, similar results were obtained when repeating the analyses by using the cells' place fields rather than the cells' spike trains (Fig. S7ii).'**

In parallel, we also performed corresponding place-field based analyses, which were requested by reviewer #2, with cell as the unit of analysis (Fig. S7).

Finally, no attention is given to possible changes in the theta waveform and the consequences of this on the metrics of interest. It is well established that theta is not sinusoidal and asymmetries in the wave can impact resulting analysis linking to phase (Belluscio et al., 2012). Thus, it stands to reason that lesions of upstream inputs may have changed the waveform and that those changes could impact the phase precession values. Methods exist for controlling for this, testing that the conclusions do not change (or documentation of how they do) when those are applied would strengthen this work.

This is an important consideration, and we followed the suggestion to use the methods in Belluscio et al. (2012) to calculate asymmetry. Because their methods were previously used for CA1 recordings, we introduced minor modifications to account for the higher gamma amplitude in CA3 recordings. Compared to the strong asymmetry of theta waves that was detected for CA1 recordings (Belluscio et

al., 2012), our key finding is that the asymmetry at CA3 recording sites is minor (asymmetry index, DG-control group: -0.05; MEC-control group: -0.03). Asymmetry was further reduced after DG and MEC lesions, with essentially no remaining asymmetry (DG-lesion: -0.0094; MEC-lesion: -0.0050). Without waveform asymmetry, Hilbert transforms are an appropriate and perhaps the least biased method to estimate phase. To show that theta oscillations that were recorded in CA3 show only minor degrees of asymmetry, we added the analyses of waveform asymmetry as a supplementary figure (Fig. S5), which is also included for inspection below.

In addition, we updated the methods section to provide a description of the calculations of the asymmetry index, and we inserted a statement in the main text that refers to the new supplementary figure:

Line 1039-1052:

To confirm that the phase estimate is not unduly biased by asymmetric theta waves, as previously shown for CA1 recordings⁵⁶, we examined theta wave asymmetry at all CA3 recording sites that were selected as the reference for phase precession analysis. To calculate an asymmetry index, we adopted the methods of ref. 56 with minor modifications to account for the higher gamma amplitude in CA3 compared to CA1. We began by bandpass filtering the raw LFPs in the theta band (6-10 Hz) and in the broader 1 Hz to 80 Hz band, as in ref. 56. Using the 1-80 Hz bandpass-filtered signal, we then identified the maximum within the first half the theta cycle (0-180° of the 6-10 Hz filtered signal) and the minimum

within the second half of the theta cycle (180°-360° of the 6-10 Hz filtered signal) and marked these extrema as peaks and troughs, respectively (Fig. S5a). The asymmetry index was then the ratio of the duration of the ascending wave segment (trough to peak) divided by the duration of the descending wave segments (peak to trough) on a logarithmic scale. Using this scale, zero corresponds to a symmetric wave shape. To then quantify the asymmetry for each recording session, we calculated the mean asymmetry index over all theta cycles of a recording session (Fig. S5b).'

Line 156-160:

'We also examined the symmetry of the wave shape of theta oscillations because notable asymmetry has been reported in CA1⁵⁶. In our CA3 LFP recordings, we did not find major asymmetry of theta waves, and there was no added asymmetry with either DG or MEC lesions (Fig. S5). The shape of theta waves is therefore not a source of bias for any of the phase measurements.'

Minor concerns:

Statements included in the results with no empirical support should be moved to the discussion or removed entirely. Two examples include, 1) the statement at lines 176-178 concerning the importance of DG input for generating sufficient phase precession to support 'real-time encoding and retrieval of episodic memories' is applying more interpretation than the data warrant given that phase precession persisted though weakened and no empirical evidence was provided regarding either encoding or retrieval of memories, and 2) statement at lines 239-241 regarding DG being essential for generating intrinsic 'look-ahead' spikes is too strong given that no overt analysis of look ahead was performed. While both examples are valid connections to existing literature, they are not factual accounts of the data and insert a fair amount of interpretation and require a fair amount of inductive reasoning and thus better belong in the discussion. Relatedly, the concluding sentence of the manuscript says that the present work showed that emergence of look ahead depends on DG but without concrete clear demonstration.

The statements that were previously included on lines 176-178 and on lines 239-241 are now deleted from the results, and we rewrote the concluding sentences of the manuscript, which now much more broadly states that our data are consistent with several hypotheses that implicate DG in several forms of predictive coding:

Line 636-646:

'Taken together, our data and phenomenological model identify that it is the spikes in the initial theta cycles and at late phases of the theta cycle that are particularly dependent on DG input and on precisely timed inhibitory oscillations. The late-phase spikes are thought to emerge from internally stored patterns of synaptic strength that generate prospective neuronal activity³⁰, and our results thus suggest a critical contribution of DG for such activity patterns to emerge in CA3 networks. This is conceptually aligned with our previous report that the DG network contributes to prospective neuronal activity patterns during SWRs¹² and is consistent with the hypothesis that DG inputs are essential for sequence coding, future planning, and for generating intrinsic "look-ahead" spikes in CA3^{30, 71, 84, 85}. While such a function has been proposed by numerous computational models, our results provide experimental evidence for a role of DG beyond the previously established functions of pattern separation and novelty detection^{3, 8}.'

The analysis of number of spikes in the first full theta cycle (starting ~line 259) was troubling for two

reasons. First, the effect is driven by a difference in the DG control condition rather than the experimental condition. The DG lesion, MEC lesion and MEC control groups were all effectively the same. Second, the DG control condition had a mean of less than 1, suggesting it is possible to have the first theta cycle in the sequence have less than 1 spike. But I struggle to understand how it is considered as the first cycle of the sequence if there is not yet spiking. I looked to clarify in the methods regarding how this cycle was defined but failed to find the relevant method.

We realize that this was neither calculated in the most effective way nor described comprehensively. In the original manuscript, we used the number of spikes in the first full theta cycle (with full cycles defined as beginning and ending at the peak LFP) after the cycle that followed the first spike, which could have been imprecise for a number of reasons. We now updated the measurement to a more accurate one, and count the number of spikes within a full cycle (i.e., 360°) from the time of the first spike. This always includes the first spike, and spike counts over the cycle are thus always equal to or greater than 1.

With the updated measurement, the difference between the control groups is small (2.61 in DG-Controls and 2.49 in MEC-Controls), and the average fraction of spikes in the first cycle is now shown to increase moderately with both DG lesion and MEC lesions (by 23.8% and 16.5%, respectively). We note that this measurement does not indicate whether the phase of spiking has changed, and is therefore not critical for the argument that spike onset phase is changed by the DG, but not the MEC lesion. To report the new analyses, we replaced panel b in Fig. 4 and provided an updated description of the results in the main text.

Line 311-318:

'The number of initial-cycle spikes increased 23.8% in trains from cells of DG-lesion compared to control rats (from 2.61 to 3.23, $n = 84$ CTRL^(DG) and 68 LESION^(DG) cells, z -statistic = -4.38, $U = 5243$, $p = 1.2 \times 10^{-5}$, MW test). Similarly, the median number of initial-cycle spikes increased by 16.5% in spike trains of cells from MEC-lesion compared to control rats (from 2.49 to 2.9 spikes, $n = 101$ CTRL^(MEC) and 158 LESION^(MEC) cells, z -statistic = -2.74, $U = 11516$, $p = 0.006$, MW test; Fig. 4b). Both lesion groups therefore showed disinhibition at the onset of the spike train, but this was accompanied by more early-phase spikes in only cells from DG-lesion rats (see Figs. 3c and 4a).'

No attention is given to the fact that lesions of the MEC will also impact the DG.

As pointed out by the reviewer, MEC lesions result in the loss of a major direct entorhinal input to not only CA3, but also DG. It is therefore possible that some of the effects that emerge with the MEC lesion are mediated indirectly by the loss of MEC inputs to DG. However, if this were the most relevant pathway for spike timing, the effects on CA3 firing patterns in the MEC and DG lesion groups should either be largely overlapping, or MEC lesions should have a more pronounced effect than DG lesions as a consequence of the loss of both direct and indirect projections. Because this is not what we find, it can be concluded that the same effects on timing as with DG lesions do not arise from a loss of MEC inputs

to DG, presumably because the function of DG remains partially intact when DG continues to receive inputs from LEC in MEC-lesion rats.

To more comprehensively discuss the reasons why DG and MEC lesion effects differ despite the MEC inputs to DG, we now amended the discussion to include the points that are mentioned above.

Line 540-550:

'Given that MEC layer II does not only project directly to CA3, but also to DG⁷⁵, it might have been expected that MEC lesions result in larger deficits when direct effects on CA3 and indirect effects via DG on CA3 combine. However, the preserved pairwise spike timing of CA3 after MEC lesions and the moderately preserved phase precession in CA3 after MEC lesions differ from the profound disruption of pairwise spike timing with DG lesions and differ from the previously reported profound disruption of the temporal order in pairs of CA1 cells and of phase precession in CA1 cells after MEC lesions^{54, 76}. The less pronounced effect on the timing of CA3 firing patterns with MEC lesions compared to DG lesions could be a consequence of remaining LEC and medial septal inputs to DG, which preserve critical aspects of DG firing patterns.'

Does the model account for the differential effects of the lesion types on phase of first spike and last spike? Overt description of this in either case would help. Inclusion of this in the figure would too.

Indeed, a key feature of our model is that there is a difference in how first-spike phase is constrained after the model DG lesion compared to the model MEC lesion (third row of Fig. 7d). In particular, Fig. 7e visualizes that a broader distribution of onset phase of spike trains emerges from a less constrained phase of inhibition (φ_{inh}) after DG lesions. In addition, we now added figure panels to Fig. 7e and simulated spike trains to Fig. S11 that show that the allowable parameter space of the DG lesion model results in onset spike times that are distributed over a wide range of the theta cycle.

The effects of the model lesion on onset phases and on the correspondence to experimental data are now pointed out more clearly in the results and discussion sections, and we specifically refer to the figure panels that show the key effect (i.e., Fig. 7d and e).

Line 465-475:

'When the DG input was set to zero in the model, the model-generated phase precession data matched with the empirical data over a broader and shifted set of φ_{inh} values compared to controls (Fig. 7d). The set of parameter A values was also shifted downwards with the lesion, which is expected when decreasing the total amount of excitation by setting one of the excitatory inputs to zero. In contrast, when MEC-lesion empirical data were matched with model data with the MEC input set to zero, the model data could reproduce the empirical data with values of φ_{inh} that were largely unchanged, along with downward-shifted A values compared to the control empirical/model data match (Fig. 7e). This suggests that the loss of the excitatory MEC inputs requires a compensatory reduction of inhibitory amplitudes, but that the increased phase variability observed in MEC-lesion data requires no major adjustments in the timing of inhibition.'

Line 599-606:

'The resulting phenomenological models can reveal patterns of rhythmic inhibition that are compatible with the observed changes in the temporal firing patterns of CA3 cells. By mapping the experimental findings to the computational model, we found that the results suggest that the DG and MEC input pathways could be exerting two different types of effects on the inhibitory subnetwork. The

consequences of loss of DG inputs can be explained by a combined expansion in the inhibition phase and downward shift in the inhibition amplitude, whereas the consequences of loss of MEC inputs can be explained solely by a downward shift in inhibition amplitude (see Fig. 7e).'

Line 112 – clarify if bouts are defined on a per-cell basis or population basis?

In the abbreviated version of the description of the methods to identify activity bouts, we now clearly state that we initially selected bouts of increased neuronal activity on a per cell basis:

Line 114-116:

'We therefore directly identified bouts of each cell's increased neuronal activity by selecting spike trains from the temporal firing patterns (see Fig. S2c for details on the criteria).'

Line 359 – Venditto et al., 2019 examined CA1 sequences in the context of reduced phase precession

Thank you for bringing this reference to our attention, which is now included. The corresponding statement has now been updated to correctly state that theta sequences have been observed along with impaired phase precession. Please note that this is also an observation that we confirm here for CA3 recordings in the MEC lesion group where phase precession is diminished while pairwise timing remains intact.

Line 382-389:

'Phase precession is thought to link the slower behavioral sequence to the faster pairwise temporal correlation in theta cycles. However, dissociations between theta sequences and phase precession have been reported. Phase precession without theta sequences can be observed in a novel environment and without CA3 inputs to CA1^{41,59}, and theta sequences have been observed with impaired phase precession⁶⁰. Given that theta phase precession and theta sequences can be dissociated, we asked whether the precise pairwise timing at the theta scale was diminished when phase precession was impaired by the reduction of either DG or MEC inputs to CA3.'

Line 975 – specific measure of variance not defined, inclusion of equation or citation would be more transparent

We now expanded the preceding paragraph, which describes the methods that were used for the quantification of circular data, and made sure to not only include the citation of the toolbox by Berens but also the name of the function in the toolbox and the citation of the book that the toolbox is based on.

Line 1084-1088:

'Calculation of circular variance and statistical tests of circular data (e.g., theta phase values). The circular statistics toolbox CircStats⁸⁸ was used for the computation of statistical quantities, such as the circular variance of theta phase ('circ_var' function of the CircStats toolbox). The CircStats toolbox⁸⁸ also implements many statistical recipes by ref. 89 and was used for statistical tests of circular data.'

Reviewer #2 (Remarks to the Author):

In this paper Ahmadi et al. re-analyze 2 previously published dataset, exploring the contributions of DG and MEC to CA3 activity at the fine-scale temporal level. They provide a computational model that accounts for the role DG and MEC inhibitory subnetworks in generating phase precession and sequences in CA3. The investigation relies on solid premises. The manuscript is well written. "line 134 ... it is not clear whether DG inputs are necessary for phase precession in its direct target cells in CA3." I appreciate the premises of the investigation. There is a need for studying temporal coding at the single cell and population levels in other hippocampal subfields aside CA1.

Few major points that need to be addressed:

Major points:

- I understand details of the lesions have been reported in previously published papers, but there is the need for including histology picture here as well (Figure 1 and 2). Readers need to understand the level of damage of the lesions. This directly connects to the following point.

As suggested, we included the previously published histology pictures in a new supplementary figure (Fig. S1) and refer to the new figure in the main text.

Line 108-110:

'In these datasets, CA3 cells were recorded in hippocampus-dependent working memory tasks after lesioning either dentate granule neurons or the MEC (Fig. S1, Fig. S2a,b), and each lesion group was paired with a respective control group (Table S1).'

- Both MEC and DG are layered structures. Each layer contributes specifically to local and up-/down-stream computations. The major MEC lesions reported (95.3% of layer II, 92.4% of 824 layer III, and 91.4% of deep layers) interrupt both direct connections to CA1 and the classic tri-synaptic circuit. The computational model and the discussion are based on careful and delicate assumptions on the anatomy of the circuitry, but the same level of detail is not mirrored in the intervention. "The mechanisms by which DG and MEC circuits contribute to the organization of the precise temporal profiles of CA3 spiking (Fig. 5e-f) are therefore distinct and are consistent with a model in which DG inputs effectively restrict spiking to late phases of the theta cycles early in the spike train of a CA3 neuron (i.e., upon entry into the place field) (Fig. 5e). In contrast, later in the spike train (i.e., in the middle and near the exit from the place field), MEC inputs appear to ensure an appropriate mean theta phase of CA3 spiking by driving CA3 neurons in time windows around a monotonically decreasing mean theta phase over successive theta cycles" This is a key statement for this paper. But again, it is difficult to exclude unwanted effects of the lesions - given that they destroy the majority of the MEC. It would be nice if the authors can elaborate on this.

Although our MEC lesions are not layer specific, effects from the loss of direct MEC layer II projections to CA3 should be much more relevant than effects from the direct MEC layer III projections to CA1 because there is very limited direct feedback from CA1 to CA3. It could be argued that we also disrupt a longer feedback loop from CA1 through MEC to CA3, but this would also need to involve the projections from MEC layer II to CA3, such that layer II projections are again most relevant.

An important additional point to consider is that MEC layer II is a major input to not only CA3, but also to DG (see also our response to reviewer #1). It is therefore possible that some of the effects that

emerge with the MEC lesion are mediated indirectly by the loss of MEC inputs to DG. However, if this were the most relevant pathway for spike timing, the effects on the MEC and DG lesion groups on CA3 firing patterns should either be largely overlapping or MEC lesions should have a more pronounced effect than DG lesions as a consequence of the combined loss of direct and indirect projections to CA3. Because this is not what we find, it can be inferred that the same effect on timing as with DG lesions does not arise from a loss of MEC inputs to DG, presumably because the function of DG remains partially intact when DG continues to receive inputs from LEC in MEC-lesioned rats.

To now provide a more complete picture of the conclusion that can be drawn from lesions that include all MEC layers, we expanded the discussion of MEC inputs to DG and CA1.

Line 540-558:

'Given that MEC layer II does not only project directly to CA3, but also to DG⁷⁵, it might have been expected that MEC lesions result in larger deficits when direct effects on CA3 and indirect effects via DG on CA3 combine. However, the preserved pairwise spike timing of CA3 after MEC lesions and the moderately preserved phase precession in CA3 after MEC lesions differ from the profound disruption of pairwise spike timing with DG lesions and differ from the previously reported profound disruption of the temporal order in pairs of CA1 cells and of phase precession in CA1 cells after MEC lesions^{54, 76}. The less pronounced effect on the timing of CA3 firing patterns with MEC lesions compared to DG lesions could be a consequence of remaining LEC and medial septal inputs to DG, which preserve critical aspects of DG firing patterns. Similarly, the more severely disrupted CA1 than CA3 firing patterns with MEC lesions could be a consequence of a more major role of MEC projections to CA1 than to CA3 and/or a more minor role of the second external excitatory inputs – CA3 inputs to CA1 as opposed to DG inputs to CA3. As a consequence, the additional preserved inputs from DG are sufficient to preserve temporal organization in CA3 while CA3 inputs to CA1 are not⁵⁴. The DG inputs thus confer the CA3 circuit with the propensity to generate sequential activity patterns, such that this computation – even when MEC inputs are diminished – can emerge with remaining DG projections to CA3⁵⁴. Our data therefore suggest the broader DG-CA3 circuit is required to support the computations that generate theta sequences.'

- The choice of using spike-trains without previously identifying place fields is well explained. However, at least in one of the previously published papers (Sasaki et al. 2018, figure 6 and 7) place fields were well isolated. Does a strict selection of place field yield any differences in the analysis of figure 3, 4 and 5 of the manuscript? Reports of place field dynamics (rate code) are also interesting, when comparing the two manipulations.

As suggested, we redid analyses of key measurements that are shown with spike trains in Figs. 3, 4 and 5 with using place fields. The key result that DG lesions result in less restricted initial spiking at late phases while MEC lesions do not result in a corresponding effect is reproduced. However, there are also some differences in measurements over the duration of the spike trains compared to the length of the place field. With DG lesions, the circular variance is higher early in the spike train (i.e., Fig. 4d), while this effect is not pronounced early in the field (Fig. S7iid). These differences for distance-based measures are perhaps expected given that the alignment of each train with the field is variable, and that effects on temporal organization of spiking should more readily emerge with temporal rather than spatial alignment.

The new supplementary figure (Fig. S7) is now referred to in the text and in figure legends on multiple occasions.

Line 282-291:

'Taken together, the selective effects on onset, but not offset phase with DG, but not with MEC lesions indicate that it is predominantly the DG input rather than the MEC input to CA3 that is involved in setting the narrow, late onset theta phase of CA3 spikes. These results were also confirmed when repeating the same analyses by using the cells' place fields rather than the cells' spike trains (Fig. S7), which confirms that qualitative differences in the effect of DG and MEC lesions on temporal firing patterns in CA3 cannot be attributed to methodological details.'

- LFP analysis are missing. At least one (supplementary) figure needs to address changes in theta, slow gamma and fast gamma parameters (θ frequency and amplitude, mean phase preference of slow and

fast gammas), comparing the 2 datasets and lesions. Measures should also be normalized by running speed. This should be considered as a control measure to rule out that changes might derive from alterations of the theta rhythm.

We now added a new supplementary figure (Fig. S4) that comprehensively describes lesion effects on LFP delta, theta and gamma oscillations as well as on the running speed modulation of theta oscillation frequency.

In summary, we did not observe effects on delta, theta, and slow gamma power. For fast gamma power, we observed an increase in power with MEC lesions, but not with DG lesions. In addition, the increase in fast gamma power was accompanied by a ~3-fold increase in the modulation of fast-gamma amplitude by theta phase, which indicates that theta and gamma oscillations continued to be well coordinated with diminished MEC input. In contrast, while DG lesions did not have an effect on the overall high-gamma amplitude, they reduced theta-phase modulation of fast gamma. Interestingly, the loss of coordination with theta was specific for fast-gamma amplitude, as modulation of slow-gamma amplitude was retained.

Although our patterns of results might be considered inconsistent with the notion that fast gamma is forwarded by MEC inputs to hippocampus (Colgin et al. 2009), the previous models on gamma generation in hippocampus are predominantly based on analysis of direct MEC layer III inputs to the hippocampal CA1 area, while our data are from CA3 recording sites. Our findings of a major effect of MEC lesions on CA3 fast gamma amplitude and of DG lesions on the coordination of theta and fast gamma calls for more detailed models on gamma generation across hippocampal subregions. Furthermore, the selective effects on the modulation of fast-gamma amplitude by theta phase indicate that theta oscillations remained able to time other types of oscillations after the DG lesion and that the effects are thus specific for one type of phase-amplitude modulation.

In addition to the effects on fast gamma, MEC lesions also diminished the extent to which theta frequency was modulated by running speed and DG lesions yielded small, but detectable effects on the theta frequency. However, theta frequency did not differ after DG and MEC lesions, which makes it unlikely that any differences in precise timing of CA3 cells between the lesion groups can be attributed to a difference in LFP patterns.

In addition to the analysis on the basic LFP properties that are requested here, we also performed – in response to a comment by reviewer #1 – an analysis of the symmetry of the theta waves, which did not show any substantive differences between the two lesions.

All of the effects of lesions on LFP are now summarized in a new paragraph in the results section.

Line 143-160:

In addition to examining the changes in the spatial firing patterns and firing rates with the lesions, we also examined whether LFP oscillations were changed by either of the two lesions (Fig. S4). We did not observe any major effects of either lesion on the power of delta, theta, and slow gamma oscillations, but the power of fast gamma oscillations was increased with the MEC lesions and theta phase-fast gamma amplitude co-modulation was markedly reduced with DG lesions. In addition, the DG lesions resulted in a minor decrease in theta oscillation frequency [DG control vs. lesion: 7.92 ± 0.21 Hz and 7.24 ± 0.48 Hz, median \pm interquartile range (iqr), $n = 6$ and 10 sessions, $U = 98$, $p < 0.001$, Mann-Whitney (MW) test]

and MEC lesions in a loss of speed modulation of theta frequency (MEC control and lesion: $r = .361$, $p < 0.001$ and $r = 0.075$, $p = 0.11$, linear regression). However, the theta oscillation frequencies after DG and MEC lesions was indistinguishable (DG vs. MEC lesion: 7.24 ± 0.48 Hz and 6.92 ± 0.90 Hz, median \pm iqr, $n = 10$ and 19 sessions, $U = 173$, $p = 0.30$, MW test), such that any differences in precise timing of CA3 cells between the lesion groups cannot be attributed to a difference in LFP patterns. We also examined the symmetry of the wave shape of theta oscillations because notable asymmetry has been reported in CA1⁵⁶. In our CA3 LFP recordings, we did not find major asymmetry of theta waves, and there was no added asymmetry with either DG or MEC lesions (Fig. S5). The shape of theta waves is therefore not a source of bias for any of the phase measurements.'

Methods for LFP analyses were added.

Line 1025-1052:

'For power analyses of frequency bands in addition to the theta range (6-10 Hz), the raw LFP signal was filtered in delta (1-4 Hz), slow gamma (25-50 Hz), and the fast gamma (50-100 Hz) and band power was calculated using Matlab's 'bandpower' function. Phase-amplitude coupling of theta and gamma oscillations was performed using published Matlab code (<https://github.com/tortlab/phase-amplitude-coupling>)⁸⁷. In addition, the speed dependence of theta frequency was analyzed by first calculating the time-resolved spectrogram (5 second moving window in steps of 0.5 s) of the LFP using Chronux (`mtspecgramc_fast` function, 1-20 Hz with time-bandwidth product $TW = 3$ and number of tapers $K = 5$), by then finding the frequency with the peak power in the 6-10 Hz band at each position, and by finally estimating the theta frequency associated with each position by linearly interpolating the spectrogram-based frequency estimates. Next, the speed in each session was binned with a resolution of 2 cm/s, and a regression analyses was performed for the data points of each session as well as for the data points of all sessions of a group.

To confirm that the phase estimate is not unduly biased by asymmetric theta waves, as previously shown for CA1 recordings⁵⁶, we examined theta wave asymmetry at all CA3 recording sites that were selected as the reference for phase precession analysis. To calculate an asymmetry index, we adopted the methods of ref. 56 with minor modifications to account for the higher gamma amplitude in CA3 compared to CA1. We began by bandpass filtering the raw LFPs in the theta band (6-10 Hz) and in the broader 1-80 Hz band, as in ref. 56. Using the 1-80 Hz bandpass-filtered signal, we then identified the maximum within the first half the theta cycle (0-180° of the 6-10 Hz filtered signal) and the minimum within the second half of the theta cycle (180°-360° of the 6-10 Hz filtered signal) and marked these extrema as peaks and troughs, respectively (Fig. S5a). The asymmetry index was then the ratio of the duration of the ascending wave segment (trough to peak) divided by the duration of the descending wave segments (peak to trough) on a logarithmic scale. Using this scale, zero corresponds to a symmetric wave shape. To then quantify the asymmetry for each recording session, we calculated the mean asymmetry index over all theta cycles of a recording session (Fig. S5b).'

- On a broader level of discussion: if DG contributes as well to sequence coding and future planning as proposed by the authors (in line 566) what are key differences with the CA3-CA1 circuit (Treves and Rolls, 1992)?

We are somewhat uncertain whether we are correctly interpreting the request to comment on ‘key differences with the CA3-CA1 circuit’. In the cited paper, Treves and Rolls (1992) focus on comparing the perforant path and the mossy fiber input to CA3. This is corresponding to the key comparison in our data analysis, but their work focuses entirely on novelty, and is now cited (ref. 8) in our introduction and discussion.

Line 24-27:

‘This is supported by the observation that spatial firing patterns of DG cells show pattern separation³⁻⁵, which is in turn consistent with the general conceptual framework that the DG mossy fiber projections to CA3 support memory by promoting the generation of new and distinct hippocampal firing patterns⁶⁻¹⁰.’

Line 497-499:

‘The DG is the first processing stage in the intrahippocampal circuit and is considered to perform a number of specialized computations that are critical for memory such as spatial and temporal pattern separation as well as novelty detection³⁻¹⁰.’

Line 641-646:

‘This is conceptually aligned with our previous report that the DG network contributes to prospective neuronal activity patterns during SWRs¹² and is consistent with the hypothesis that DG inputs are essential for sequence coding, future planning, and for generating intrinsic “look-ahead” spikes in CA3^{30, 71, 84, 85}. While such a function has been proposed by numerous computational models, our results provide experimental evidence for a role of DG beyond the previously established functions of pattern separation and novelty detection^{3, 8}.’

While Treves also published many computational models that include the CA1 network, we assume that Treves (2004) is perhaps the most relevant. Their model considers inputs from entorhinal cortex, and for the implementation of the differentiated CA3-CA1 network, also collaterals from CA3 to CA1. While their model did not go as far as considering the relation of spiking to theta oscillations, it shows that spike adaptation was critical for predictive coding, which is now mentioned in the discussion.

Line 558-562:

While this observation is inconsistent with an early phase precession model that uses asymmetric synaptic weights in the recurrent CA3 network to generate phase precession⁴⁷, it has more recently been shown that recurrent networks need to be combined with external inputs or with mechanisms that lead to firing frequency adaptation to robustly generate phase precession or predictive coding^{51, 53, 77}.’

- The main behavioral effects of the lesions are already reported in the papers published previously with these dataset (Sasaki et al. 2018, Sabariego et al. 2019). Is it possible to relate behavioral performances with either CA3 phase precession or temporal correlations of CA3 cells in DG vs MEC lesions?

Because hippocampal CA3 cells project to CA1, which is in turn, the main output pathway from hippocampus, it is our assessment that any findings from such an analysis would be inconclusive. For the MEC lesion, we previously reported that phase precession is almost completely abolished in CA1 (Schlesiger et al. 2015). It is therefore likely that the behavioral deficits emerge as a consequence of the temporal disruption of CA1 firing patterns rather than as a consequence of retained temporal sequences in CA3 firing patterns. For DG lesions, we do not have a comprehensive CA1 dataset that would allow us to assess whether the disrupted temporal order in CA3 is also reflected in CA1. We would therefore

suggest that a comprehensive analysis of the relation of phase precession to behavioral performance cannot be meaningfully performed with the current dataset and requires further recording data, which are beyond the scope of the current study.

Minor points:

- Line 101-130: having a clear description of the methodological details used to analyze the data is extremely useful; however, I would suggest moving portions of this sections to the method part, as the flow of the section itself would benefit.

As suggested, this section was substantially shortened (now 22 compared to 30 lines in the original manuscript). The section now reads as follows:

Line 106-127:

'To test the contribution of DG and MEC inputs to CA3 phase precession, two previously published datasets with recordings of neuronal activity in the rat hippocampal CA3 region were analyzed^{12,55}. In these datasets, CA3 cells were recorded in hippocampus-dependent working memory tasks after lesioning either dentate granule neurons or the MEC (Fig. S1, Fig. S2a,b), and each lesion group was paired with a respective control group (Table S1). Because the working memory tasks required the rats to follow chosen trajectories, coverage of space was inevitably non-uniform. We therefore reasoned that the method of defining spike trains by first identifying place fields and then identifying passes through fields may not be precise as a result of uneven coverage. We therefore directly identified bouts of each cell's increased neuronal activity by selecting spike trains from the temporal firing patterns (see Fig. S2c for details on the criteria). We only considered neuronal activity during locomotion (Fig. S2d) when there is reliable occurrence of theta oscillations. Given that we included only spikes within trains and during movement periods, only a proportion of all recorded spikes (34.3% vs. 20.4% DG control and lesion; 30.6% vs. 22.5% MEC control and lesion) were included as qualifying trains and further analyzed (Fig. S3). While spike trains were identified solely by timing and velocity criteria, we confirmed that the trains clustered preferentially at one or few spatial locations, as would be expected for CA3 place cells. In addition, we confirmed that there were only minor differences in firing rate and spatial precision measurements when comparing cells between the control group for DG lesions and the control group for MEC lesions, even though the number of spikes per train and the path length during trains differed between these groups (Fig. S3a). These analyses confirm that the spatial firing characteristics of the two control datasets are comparable even though they were taken from two different spatial working memory tasks^{12,55}.'

- Line 290 "Putative granule cells exhibited a narrow theta phase preference at the onset of spiking". Supplementary figure S3a,b,c should be mentioned before Figure 4. This section can be included in the previous paragraph.

The text is now updated so that the paragraph on putative granule cells is moved up (line 237-246) and precedes the paragraphs that describe Figs. 3 and 4 (line 248-337). With the moved paragraph, the references to all panels of Fig. S3 are now mentioned before any references to Fig. 4.

- Figure 7 Panel d: plots are not easy to read. I understood the idea to show and compare phase shifts, but maybe different color coding can be used (or number of panels decreased).

As justly mentioned by the reviewer, some of the graphical choices and the labelling in Fig. 7d were not conducive to its readability.

-We now consistently use the symbols Φ_{on} and Φ_{off} for onset and offset phase, and φ_{inh} for the phase of the inhibitory oscillation in the model

-We abandoned the use of pink outlines (which had the appearance of pink dots for small parameter spaces) and consistently use black/white outlines to highlight the model parameter space that explains data. We also increased the line weight of the outlines to make the zone that explains the data more prominent when plotted on top of color-coded plots.

-We updated the color code for Φ_{on} , Φ_{off} , Slope, and R^2 to make the appearance of the patterns in the parameter space more distinct.

-We rearranged the panels such that the parameters are now ordered vertically, which allowed us to place the masks for the correspondence between data and model to the right ('admissible parameter space').

- We reconsidered how to depict the overlap of admissible parameter spaces. Admissible space for each parameter is now shown in white and non-admissible space in blue. This allows us to render the zone of overlapping admissible spaces across parameters (Φ_{on} , Φ_{off} , Slope, and R^2) in white and zones with an increasing number of overlapping non-admissible spaces with increasingly darker shades of blue. This color scheme yields a more intuitive way of reading the summary panel on the bottom.

- The order of the parameters (Φ_{on} , Φ_{off} , Slope, and R^2) were adjusted such that Slope is now on top to provide a more intuitive progression of details.

-The figure legend of Fig. 7 is substantially updated.

Line 814-852:

Figure 7. A single-cell model with two oscillating excitatory inputs and an inhibitory input reproduced the main empirical results. **a**, Model construction. The three inputs are modeled after DG, MEC, and local inhibition converging onto the CA3 model neuron. The excitatory DG and MEC inputs oscillate at faster-than-LFP frequencies ($\omega_{DG} = 8.6$ Hz, $\omega_{MEC} = 8.5$ Hz) with DG inputs more prominent early in the field and MEC inputs more prominent later in the field. The inhibitory input oscillates at 8 Hz throughout the place field, corresponding to LFP theta. Small Gaussian noise is added to the inhibitory input to ensure robustness against minor perturbations. The excitatory inputs contribute positively at the fixed phase difference ψ , which is taken to be 0° from published findings⁶⁸ on DG and MEC population activity. The inhibitory input contributes negatively to the total drive at a phase differential φ_{inh} relative to excitation at place field entry. Finally, the total drive is rectified. A reference 8-Hz oscillatory inhibition is displayed at the bottom left, which is used to extract the phase of the simulated spikes. The phase-distance relationship is then depicted as for experimental data. Not all steps are displayed for brevity (see Methods for full details). **b**, Phase-versus-normalized distance plots of spikes generated by the model. Three randomly selected single pass examples show phase precession. The values of $A/\varphi_{inh} = 3/260^\circ$ (inhibitory oscillation amplitude and phase), $I_{DC} = 0$ (inhibition DC component) and $\psi = 0^\circ$ (excitatory phase differential) are the same across the three plots. The measured slopes from the simulated data are displayed at the top of each panel along with the significance. **c**, Phase-versus-normalized distance plot for spikes across multiple passes. Panels from top to bottom are generated by the control, DG lesion, and MEC lesion models (1 cycle = 360°). The measured slope and significance of phase precession are displayed on top of each panel. n.s., not significant, * $p < 0.05$, *** $p < 0.001$. Lesion experiments were simulated by setting the DG or MEC input to zero, and all other parameters as in **b** except for A lesion = 2.

In both lesion cases, phase precession slopes are reduced. (d and e) DG and MEC lesions alter CA3 phase precession in qualitatively different ways. d, Values of slope, explained variance (R^2), onset phase (Φ_{on}), and offset phase (Φ_{off}) are shown for combinations of A and φ_{inh} parameters. The color scale in each panel is according to the color bar to the right. The range of model A- φ_{inh} parameter space that corresponds to empirical phase precession values are shown in blue and white plots to the right with white areas depicting the space that yields 80% of the empirical measurements. The intersection between white areas for multiple phase precession measurements is displayed in the overlap plots at the bottom (dark blue to white, 0 to 4 measurements overlap). The zone of overlap for all four measurements is delineated with outlines (black or white lines) that are projected back onto all other panels. e, left, Distribution of parameter values A and φ_{inh} that result in a match with the empirical control and lesion data. To match the empirical phase precession measurements, the DG-lesion but not the MEC-lesion model is forced to take on a shifted set of φ_{inh} values. e, right, Distributions of Φ_{on} and Φ_{on} values that are generated by the A- φ_{inh} parameter space that corresponds to the overlap areas of each model type. The DG lesion model generates the broadest Φ_{on} distribution. DG control, light blue; DG lesion, dark blue; MEC control, yellow; MEC lesion, red.'

- FigureS3: how are place field defined here? After the selection of spike-trains or in the classical way?

In this figure (now Fig. S6) and in the new figures that repeat all of the key analyses with a map-based place field definition (Fig. S7), we used the classical definition. Place fields were defined as the area within the 20% contour of the place maps. This information is now added to the figure legend and described in more detail in the methods under the subheader '**Standard place field definition**'.

Line 1098-1103:

'Standard place field definition. Standard place fields of neurons were defined as the area within the 20% contour of the place maps, and this definition was used for Figs. S6 and S7. Normalized distances along the path from the entry to the exit were calculated, and the distances were divided into 10 bins for further analysis. For example, we calculated the circular standard deviation across all spikes of each cell in each bin and plotted the mean values together with the error bars representing the standard error of the mean (Fig. S6c).'

Reviewer #3 (Remarks to the Author):

While the existence of theta phase precession has been known for ~30 years, its functional role in cognitive function and cellular mechanisms supporting it still remain largely unknown. Therefore, empirical data supporting its role and mechanisms are of importance to the field. In this manuscript, the authors focused on the mechanism of theta phase precession in the hippocampal CA3 cells, and demonstrated different roles of DG and MEC inputs to CA3 on phase precession. They report that DG input contributes to the phase of CA3 spikes especially at the initial part of a place field, while the MEC input contributes to the precision of the phase throughout the place field. In addition, while DG input contributed to theta sequence, MEC input did not. Using a computational model based on oscillatory interference, the authors demonstrated that DG lesion could have involved a shift of inhibitory oscillatory input, while the MEC lesion could have involved a decrease in the amplitude of inhibitory oscillatory input. While these findings are novel and interesting, both the functional relevance and mechanisms supporting these differences remain unclear, limiting the significance of this study.

Major:

Because most of the major points of the reviewers seem to be conceptual rather than a criticism of our data analyses, our responses are predominantly providing additional background and justification. However, we made every effort to not only include these clarifications in the response to the reviewer, but also in the text of the manuscript, in particular in the discussion section.

In addition to addressing a number of conceptual points, we also much more comprehensively explored the parameter space of our model. With these additions, we hope that it is now much more convincing that an approach that focuses on the convergence of different types of excitatory and inhibitory inputs onto a single cell has valuable explanatory power. Because this type of model allows for a much more limited number of tunable parameters than network models, it is particularly well-suited for comparisons between model and experimental data. Furthermore, the parameter space of the model can be much more comprehensively tested than for network models, and we added many of these analyses. However, these additional analyses required us to completely rewrite the code so that it would run more efficiently, to then extensively test parameter ranges, such as the full range of LFP phase difference angles and of phase angles between the two model excitatory inputs (Fig. S9). Even with the improved run time, the additional computational work turned out to be time-consuming, and contributed to the additional time to complete the revisions.

However, we believe that the new supplementary figures that are now included provide additional important insights into dynamics that emerge from the combination of two excitatory inputs with an inhibitory input, and that the ease of comparison between the data and the model continues to justify an otherwise simple model of phase precession. We are not aware of other phase precession models that have combined three types of oscillatory inputs, and the model therefore spearheads a novel approach despite its simplicity.

Eventually, models of neurons with multiple types of oscillatory inputs will need to be combined with circuit models, but we hope that it is apparent that even the full exploration of our multi-input cellular model is beyond the scope of a single manuscript. In particular, network models have numerous tunable parameters that currently cannot be biologically constrained, and a simpler model with fewer parameters is therefore a more candid approach for performing meaningful comparisons to experimental data. We hope that the dynamics of phase precession that are now reported already provide interesting initial insights and will inspire a new generation of phase precession models that

more comprehensively consider the multitude of different types of excitatory and inhibitory input pathways and the diversity of mechanisms for generating and inheriting phase precession.

As the authors pointed out, it has been known that MEC input to CA1 is necessary for CA1 phase precession. It is therefore not surprising to find that DG and MEC inputs to CA3 are necessary for CA3 phase precession.

There is certainly consensus that any major excitatory input to a brain region will determine the firing patterns of the target cell population, and this also holds up for the necessity of excitatory inputs for phase precession. In particular, inputs that have similar properties (i.e., theta modulated, excitatory) can be hypothesized to have similar effects, such that phase precession is reduced if the input is diminished. This is essentially the result of our first analysis steps (Figs. 1 and 2) before performing further in-depth analyses of spike timing.

Importantly, the additional in-depth analyses lead to the key insight that the simple conjecture of just observing reduced phase precession after reducing excitation does not hold up in at least two different ways. (1) Inputs from one brain region (i.e., MEC) to two different target regions (i.e., CA1 and CA3) have substantially different roles on phase precession and spike timing. With MEC lesions, pairwise ordering is abolished in CA1 (Schlesiger et al., 2015), but not in CA3 (Fig. 6h). (2) Inputs from two different brain regions to one target region have different effects on spike timing (Figs. 3-6).

Taken together, our results might therefore be characterized as unsurprising if only considering that there is reduction of phase precession after loss of any excitatory inputs, but as surprising and novel in showing that each input makes a unique contribution to spike timing. Although selective effects on only some features of phase precession (e.g., late phase spiking) are often considered in computational models, we are not aware of previous analyses of data that have shown equally selective effects as we show here with DG lesions compared to controls.

While it is interesting that DG input and MEC input differently affect phase precession, there is no empirical insight as to how these two areas exert different effects. The model points to the inhibition phase and amplitudes. However, this is just that the model will not fit the data without these modifications. The authors point to the potential difference of somatic (DG) vs dendritic (MEC) suppressions as the potential source of these differences, which is interesting. However, there is no empirical data backing this up. Without empirical data supporting this, there is no practical advance regarding the phase precession mechanism. It is necessary to conduct, for example, specific manipulation of these two types of inhibitory circuit to support this view. Without empirical observations, this manuscript remains to be descriptive.

We agree that our data analysis has to remain descriptive and that follow-up experiments are needed to further determine how each of the manipulated excitatory inputs engages feedforward and feedback inhibition. To combine manipulations of inhibitory sub-circuits that are engaged by a particular excitatory pathway with an assessment of spike timing and phase precession, experiments in not-yet well-defined interneuron subpopulations in freely behaving rodents will be needed. While it is common for computational models to suggest future experimental studies, follow-up studies of this nature would far exceed the scope of our study.

However, we agree that we previously did not explore our computational model as extensively as would be needed to gain additional key insights. As mentioned above, we therefore expanded our analysis to

include a larger range of parameters (Figs. S9 and S10). These analyses revealed that the two lesion cases (no model DG input or no model MEC input) typically required an adjustment in inhibitory amplitude. This is intuitive when considering that a reduction in excitation should go along with a reduction in inhibition. However, we also consistently observed that DG lesion data could be explained by a less constrained phase difference between the model excitatory and inhibitory input. The same was not the case for the MEC lesion data, where the phase angle between the excitatory and inhibitory input had to essentially match the range that also explained the control data (Fig. 7e).

In addition, we expanded our analysis of model parameters to examine how phase differences between two excitatory inputs and an inhibitory input combine to generate phase precession. Briefly, we show that phase difference can be generated over a wide range of phase differences of two excitatory inputs (Fig. 9b) as long as there is an oscillatory inhibitory input at a relatively constant and narrow phase with respect to excitation. In all cases the phase range of the inhibitory input with respect to the excitatory input is relatively narrow, except for a broadening that is necessary to explain DG lesions. One of the key insights that can be gained from combining the analyses of the experimental data (from lesioning two different excitatory inputs) with a simple computational model of adding three oscillating inputs (two excitatory and one inhibitory) is therefore the idea that the oscillatory inputs combine in ways such that one input (DG excitation) mostly controls late phase spiking while the phase of another (inhibition) controls whether phase precession can even emerge.

The new analyses and the conclusions that emerge from the new analysis are now comprehensively discussed in substantially rewritten paragraphs in the results and discussion.

Line 410-494:

'A phenomenological computational model of phase precession in CA3 cells revealed distinct effects of DG and MEC inputs on the inhibitory signal

To gain further mechanistic understanding of whether and how the effects on phase precession observed in lesion animals can arise from single-cell integration of the two excitatory theta-modulated inputs to CA3, we devised a minimal phenomenological model based on oscillatory interference. We chose to minimize the number of free parameters (see below) of the model to be able to quantitatively fit simulations of the spiking dynamics to the phase precession statistics observed in the data sets. Although our analyses of experimental data had to be limited to the two excitatory inputs to CA3 that were manipulated in lesion experiments, we reasoned that if a computational model based on oscillatory interference were to emulate the lesions, it must account for inputs beyond the manipulated inputs. Inhibition has been shown to mediate input gain control, precise spike timing and enhanced coding in networks⁶¹⁻⁶⁴ and can thus be considered essential for controlling the theta phase of pyramidal cell spikes⁴⁹. Therefore, we included an inhibitory oscillation in the model that can be viewed as corresponding to the observed oscillations of a large fraction of hippocampal interneurons at the LFP theta frequency^{65, 66}. Based on recordings from DG and MEC principal cells that are known to project to CA3, the excitatory inputs from each of these two regions were considered to oscillate at frequencies slightly above the LFP theta frequency^{67, 68}. In addition, the relative contributions of DG and MEC inputs varied along the place field to reflect the proposal that entorhinal inputs provide sensory cues at the true place field location while intrahippocampal circuits govern the prospective spiking^{30, 69-71}. We allowed the model to have four free parameters: a phase shift between the two excitatory inputs denoted by ψ , a phase shift between excitation and inhibition denoted by φ_{inh} , the oscillatory amplitude of inhibition denoted by A , and a DC component for the inhibitory oscillation (baseline inhibition) denoted by I_{DC} .

Although the full range of possible ψ parameters was tested, it is relevant to note that experimental data⁶⁸ suggest that neurons with direct MEC inputs to DG (i.e., MEC layer II neurons) and DG inputs to CA3 show activity over $\sim 90^\circ$ ranges of the theta cycle that are approximately overlapping. Even if inputs were to originate from neurons at the extremes of these distributions, the difference in ψ would therefore typically not exceed $\pm 90^\circ$.

The output of the model CA3 cell was determined by the place modulated^{72, 73} combination of the three inputs (two excitatory and one inhibitory) from which a threshold value that was constant across the place field was subtracted. Spikes were generated stochastically via an inhomogeneous Poisson point process with an intensity measure defined by the total excitatory drive minus the threshold. The simulated spike phases were extracted with respect to an 8 Hz oscillation representing the LFP theta oscillation, which was considered to be phase locked to the inhibitory oscillation (see Methods). The difference angle between intracellular and LFP oscillation was chosen as 180° , since it produced the largest spike rates at the uninhibited phase (e.g., ref. 46). Accordingly, the largest spike rates in the data would occur at the minimum inhibition phase of the model. Variations of LFP phase shift by $\pm 45^\circ$ from 180° did not qualitatively alter the result (Fig. S9a).

We simulated CA3 model neuron spikes for a broad range of parameter values. We observed for the full model (Fig. 7a) that phase precession can be consistently obtained in single trials and on trial average (Fig. 7b and c), but was less prominent when either of the two excitatory components was removed (averages shown for each lesion in Fig. 7c). To determine how parameter values under this model corresponded to the experimental data from the control and lesion groups, we calculated four phase precession measurements – slope, explained variance R^2 , onset phase, and offset phase – from the model spike data in the full A versus φ_{inh} parameter space. We then identified the region of the parameter space in which the model generated phase precession measurements that corresponded to those obtained in our empirical data (i.e., the middle 80th-percentile of the empirical observations in the experiments). This was done separately by matching the control data to the control model and the lesion data to each of the respective lesion models. In the control model, only a limited range of phase shifts (φ_{inh}) between the inhibitory input and the excitatory inputs generated phase precession measurements that corresponded to data from control animals. When the DG input was set to zero in the model, the model-generated phase precession data matched with the empirical data over a broader and shifted set of φ_{inh} values compared to controls (Fig. 7d). The set of parameter A values was also shifted downwards with the lesion, which is expected when decreasing the total amount of excitation by setting one of the excitatory inputs to zero. In contrast, when MEC-lesion empirical data were matched with model data with the MEC input set to zero, the model data could reproduce the empirical data with values of φ_{inh} that were largely unchanged, along with downward-shifted A values compared to the control empirical/model data match (Fig. 7e). This suggests that the loss of the excitatory MEC inputs requires a compensatory reduction of inhibitory amplitudes, but that the increased phase variability observed in MEC-lesion data requires no major adjustments in the timing of inhibition.

Interestingly, the other two free parameters in the model – phase differences between the two excitatory inputs (i.e., ψ) up to at least $\pm 90^\circ$ and addition of an inhibitory DC component up to a value of 1.1 – did not produce overly distinct ranges of match with empirical data (Fig. S9b, S10a). In addition, major imbalances between excitatory input amplitudes of DG and MEC (75/125 or 125/75, Fig. S10b) did not

result substantial variation in the state space for allowable empirical data. With complete lesions of each of the excitatory inputs to the model, there is an expected compensatory response in inhibition amplitude, which is approximately equal for loss of MEC and DG inputs (Fig. 7e). However, accounting for the match of empirical data to the model after each of the two lesions required different adjustments in the φ_{inh} dimension. We found that the inhibition phase needed to be broader after loss of DG inputs, while it remained approximately in the control range after loss of MEC inputs. Taken together, the lack of responsiveness of the model to changes in the ψ parameter (i.e., within the physiological range of $\pm 90^\circ$) compared to its dependence on φ_{inh} is interesting because it shows that phase precession is determined to a larger degree by the phase differences between excitatory and inhibitory inputs than by phase differences between two excitatory inputs. Overall, our simulations demonstrate that the range of observed effects in the CA3 circuit can, in principle, be generated by the interaction of major excitatory and inhibitory theta-modulated inputs to a CA3 cell (Fig. 7 and Fig. S11).'

Line 577-436:

'Accordingly, our analyses suggest that DG and MEC circuits make qualitatively different contributions to the organization of the precise temporal profiles of CA3 spiking, consistent with a model in which DG inputs effectively promote spiking at late theta phases early in the spike train of a CA3 neuron (i.e., upon entry into the place field) (Fig. 8a). In contrast, later in the spike train (i.e., in the middle and near the exit from the place field), MEC inputs appear to ensure an appropriate mean theta phase of CA3 spiking by driving CA3 neurons in time windows around a monotonically decreasing mean theta phase over successive theta cycles (Fig. 8b).

Given the major role of inhibition in shaping the spike timing in intact neural circuits, we used a phenomenological model and made sure that it was sufficiently minimalistic (i.e., had few free parameters) to allow comparisons to experimental data. While keeping the parameters to a minimum, we reasoned that it was essential to add the well-established oscillating inhibitory inputs in addition to the two oscillating excitatory inputs that were tested in our analyses of experimental data. The model is therefore conceptually related to previous models of phase precession that have considered oscillatory interference between an inhibitory somatic drive and dendritic excitation^{35, 49, 80} except that two independent excitatory inputs rather than a single input were used here. Model parameters included the theta phase difference between the excitatory and inhibitory inputs and the oscillatory amplitude of inhibition. By exhaustively searching the parameter space of the model, we identified model parameters that gave rise to phase precession measurements that corresponded to the empirical observations. We then eliminated either the DG or the MEC input to the model and repeated the search for a feature space of rhythmic inhibition that corresponded to the empirical data. The resulting phenomenological models can reveal patterns of rhythmic inhibition that are compatible with the observed changes in the temporal firing patterns of CA3 cells. By mapping the experimental findings to the computational model, we found that the results suggest that the DG and MEC input pathways could be exerting two different types of effects on the inhibitory subnetwork. The consequences of loss of DG inputs can be explained by a combined expansion in the inhibition phase and downward shift in the inhibition amplitude, whereas the consequences of loss of MEC inputs can be explained solely by a downward shift in inhibition amplitude (see Fig. 7e). The shifts in amplitude are expected because the loss of one of the excitatory pathways results in diminished excitation that is offset by a lower level of inhibition. Interestingly, the model without DG input is compatible with a less constrained input to fast-spiking interneurons that are

targeted by granule cell projections to CA3^{81,82}. In contrast, MEC inputs are known to not only target pyramidal cells but also somatostatin interneurons that predominantly control dendritic inhibition, and manipulations of dendritic inhibition have been shown to be without effect on the average spike phase throughout the place field⁸², which resembles our observation that the model without MEC inputs does not need major adjustments to the inhibitory phase to explain data. Together, these data therefore suggest that effects from manipulating excitatory inputs do not only arise from diminished direct connectivity to principal cells, but also from how these inputs engage inhibitory interneurons. In particular, our data are consistent with the mossy fiber inputs to CA3 more strongly engaging somatic inhibition, which determines the theta phase of spikes, and with MEC more strongly engaging dendritic inhibition which does not directly set the theta phase.

It is generally assumed that firing at precisely timed theta phases are a prerequisite for generating sequences of neuronal firing patterns^{41,83}, but the combinations of inputs that are necessary for implementing these computations have not been established. For example, one model of phase precession proposes that phase precession can emerge by combining two excitatory inputs that each have a different phase preference with respect to the theta cycle⁵⁰. When the strength of one input increases and of the other decreases along the animal's path, the spiking of a cell that integrates these inputs would show a phase shift over successive theta cycles. We tested versions of our model in which we offset the phases of the two excitatory inputs with respect to each other, and we show that there are no major qualitative differences compared to versions in which the two excitatory inputs are in phase. Rather, we observe that the inhibitory phase continues to constrain the match to control data for any of these models, which suggests that the phase of the inhibition with respect to excitation rather than a phase difference between excitatory inputs is the most critical parameter.'

To support the hypothesis that inhibitory input is differently modulated by the DG and MEC lesions, authors could have attempted to quantify the activity of putative inhibitory cells (rate and timing). Or, cross correlations of spike timings between putative excitatory and inhibitory cells could be used in addition to describe possible effects on the inhibitory network.

Because our recordings focused on principal neurons in the CA3 cell layer, we unfortunately do not have a sufficient number of putative inhibitory interneurons to perform the suggested analyses.

The recurrent excitatory network in CA3 would be an important candidate as a mechanism on phase precession as the authors mention in the manuscript. Although they claim that the recurrent CA3 network by itself is not sufficient based on the disrupted phase precession in the case of DG and MEC lesions, it is self-evident that recurrent network without input to CA3 cannot support phase precession. Therefore, whether the recurrent network plays a crucial role or not needs to be tested by, for example, suppressing specifically the recurrent network. Furthermore, their computational model did not include this recurrent network. In summary, although the recurrent network would be a strong candidate mechanism supporting phase precession as suggested by other researchers, this study does not provide insight into this potentially important mechanism.

We fully agree with the reviewer that the Tsodyks et al. (1996) model together with the Romani and Tsodyks (2015) model provide important and influential accounts for the potential role of CA3 recurrence to hippocampal phase precession. We also agree that our data are not sufficient to probe the role of recurrent excitation directly and that this can currently only be done with computational means

(e.g., Yiu and Leibold 2023). However, our data shows that the mechanisms underlying phase precession (be they dependent on the recurrent network or not) are distinctly affected by the two lesions. Particularly the lack of theta-scale correlations in the DG lesions argues against rigid recurrent theta sequences as in the original Tsodyks et al. (1996) model. Thus, DG input seems necessary either to set up the CA3 network to work like a Tsodyks et al. (1996) model, or DG inputs generate the sequences on their own. Since, as argued, our experimental data cannot distinguish between the two alternatives we refrained (in addition to other reasons outlined in our response) from suggesting a circuit level model. We however, have added a paragraph in the introduction that references Tsodyks et al. (1996) and a paragraph in the discussion relating our findings to the Tsodyks et al. (1996) model.

Line 73-75:

'Although theta phase precession is observed throughout hippocampus and entorhinal cortex, most mechanistic models of phase precession have focused on the CA1 region and on recurrent connections in CA3⁴⁷⁻⁵¹.'

Line 554-562:

'The DG inputs thus confer the CA3 circuit with the propensity to generate sequential activity patterns, such that this computation – even when MEC inputs are diminished – can emerge with remaining DG projections to CA3⁵⁴. Our data therefore suggest the broader DG-CA3 circuit is required to support the computations that generate theta sequences. While this observation is inconsistent with an early phase precession model that uses asymmetric synaptic weights in the recurrent CA3 network to generate phase precession⁴⁷, it has more recently been shown that recurrent networks need to be combined with external inputs or with mechanisms that lead to firing frequency adaptation to robustly generate phase precession or predictive coding^{51, 53, 77}.'

It is not clear what the different effect of the MEC vs DG lesions on phase precession means in terms of memory function. Although the authors discuss the possibility that DG is more important for sequence in the discussion, the fact that MEC is upstream of both CA3 and DG, and the fact that MEC cells show phase precession speaks against the idea that DG is more important for sequence than the MEC. Empirical data backing up the importance of the different effect of DG vs MEC lesions, for example a specific DG lesion impairing CA3 dependent sequential tasks, is needed to reveal the importance of this difference.

This comment speaks to the larger unresolved question that the function of theta phase precession, and more generally, of precise spike timing in the entorhinal-hippocampal circuit has not been firmly established. This is a more challenging question than just addressing whether the entorhinal cortex or the hippocampus is required for sequences because it requires manipulations of spike timing that leaves other aspects of entorhinal and hippocampal function intact, such as spatial firing.

For example, we show here that pairwise spike timing is preserved with MEC lesioned and DG intact, but abolished with DG lesioned and MEC intact (Fig. 6), suggesting that phase precession of MEC cells alone is insufficient for sequential firing in CA3. Therefore, phase precession in MEC could support memory independent of hippocampal function, but not necessarily by generating phase precession in hippocampus.

Furthermore, it is also not straightforward to argue for a direct correspondence between a 'sequential task' and the use of hippocampal and entorhinal firing sequences in these tasks. The time scales of the behavioral sequences and sequential neuronal firing within a theta cycle typically differ by orders of

magnitude and are even unlikely to match up after considering effects of sequence compression (e.g., Dragoi and Buzsaki 2006). We of course understand the importance of linking phase precession to memory, but as mentioned in the comment and in our response to the last major point of reviewer #2, this would require investigation across a much larger circuit that includes MEC and CA1.

The different effect of DG vs MEC lesions could reflect differences in the magnitude of decreased excitability. MEC input synapses on the more distal part of dendrites than the DG input. Therefore, the MEC lesion might have had a weaker effect on the reduction of excitatory input. This possibility should be tested for example by looking into the change of excitatory cell firing. In addition, this possibility could be tested in the computational model by changing the magnitude of DG or MEC input.

This is an interesting point that is somewhat related to the major comment #1 of reviewer #1 on net effects of the lesions on excitability, but this comment also brings up the possibility that the relative strength of the inputs differs even in control conditions. Typically, it is argued that the more distal location and the smaller strength of perforant path synapses onto CA3 are offset by the much larger number of synapses (e.g., see Treves and Rolls 1992). In addition, it is also important to consider to what extent each of the pathways engages inhibitory circuits. The precise balance of these effects, along with the mean activity levels of the inputs, will obviously determine whether there is ultimately some degree of excess excitation or inhibition when cells receive inputs. Furthermore, the lesions may provide an estimate of how effectively the target cell population continues to fire when excitation from a particular input pathway is reduced, and as mentioned by the reviewer, measurements of firing rates of principal cells in the target area (i.e., CA3) are the most obvious readout to determine whether inputs added substantial excitation.

To address this point, we updated Fig. S3 (previously Fig. S2) to now not only show the comparison between the two control groups, but also the comparison of each of the control groups with the corresponding lesion group. Compared to controls, DG lesions did not alter firing rate, while MEC lesions resulted in a minor decrease in firing rate. With regard to the comment on the balance of the excitatory contributions of DG and MEC inputs, our data would therefore support the notion that MEC inputs provide stronger net excitation (i.e., firing rates are reduced when lesioned) than DG inputs (i.e., firing rates are unchanged when lesioned), which is inconsistent with the suggestion that the selective effects of DG lesions on pairwise timing and on late phase spikes emerged from a disproportionate excitatory contribution of this pathway.

The effects of the lesions on excitability are now described in more detail in the main text.

Line 129-141:

‘Furthermore, we also examined whether lesioning a large proportion of DG or MEC inputs to CA3, which are each excitatory, had major effects on firing rates and spatial firing characteristics. Compared to controls, DG lesions did not alter firing rate, sparsity, or spatial information, but decreased selectivity and yielded longer path lengths during trains (Fig. S3b). On balance, there is therefore not a major overall change in excitation relative to inhibition, but if there are any effects, they are generally in the direction of broader fields. The results are therefore not consistent with the notion of a merely reduced excitation, such that only the late (i.e., higher rate) portion of the spatial field is observed in the lesion condition. Performing the same analyses for MEC lesions compared to controls yielded lower selectivity and spatial information, longer path lengths during trains and increased sparsity (Fig. S3c). Again, the less precise spatial firing is inconsistent with the notion that only a portion of the firing field is retained. However, the

minor decrease in firing rate with MEC lesions suggests that loss of excitation may be more predominant with MEC lesions compared to DG lesions.'

In addition, we used the model to test whether substantial differences in the relative strength of the two excitatory pathways would have effects on model behavior and on its match to experimental data. We found that a substantial imbalance (i.e., by 60 %) between the amplitude of each of the two excitatory pathways did not result in notable differences in the parameter space where inhibitory modulation could generate phase precession. This is described in the main text and depicted in a new supplementary figure panel (Fig. S10b).

Line 477-480:

'Interestingly, the other two free parameters in the model – phase differences between the two excitatory inputs (i.e., ψ) up to at least $\pm 90^\circ$ and addition of an inhibitory DC component up to a value of 1.1 – did not produce overly distinct ranges of match with empirical data (Fig. S9b, S10a).'

The computational models are often used to provide an explanation for the empirical data. In contrast, the model provided in this manuscript rather demonstrates that the model does not fit to the observed data unless additional changes in the inhibitory circuit are introduced. While these required changes of inhibitory input (A and ϕ_{inh}) could indeed indicate additional changes that occurred associated with DG and MEC lesions, it should first be considered that the model could be improved. There might be other models which could explain the data without considering additional inhibitory components. An alternative approach could be to implement a hypothetical inhibitory circuit that receives input from the DG or MEC, and then simulate lesioned conditions to show that inclusion of a specific inhibitory circuit can explain the empirical data for a wide range of parameters.

Instead of constructing a full circuit model of the dentate gyrus-CA3 circuitry with large numbers of unconstrained parameters, we decided to use a model that helps us constrain potential mechanism on the level of single-cell subthreshold processing, very much in the spirit of classical papers (e.g., O'Keefe and Recce 1993; Magee 2001; Mehta, Lee, and Wilson 2002; Thurley et al. 2008; Harvey et al. 2009). These approaches, as ours, do not imply that phase precession arises without interaction of different circuit components but rather focus on the net effect of circuit actions on the single-cell level. While these models can of course not fully answer circuit level questions, they have the advantage that their

parameter space is small and can be sufficiently explored, which makes them amenable to normative approaches with the objective to quantitatively fit the phenomenology – which was the main goal in a study that combines data analysis with a phenomenological model.

Such modelling in our paper led to the conclusion that the two types of lesions must have distinct effects on the net inhibitory input. We fully agree with the reviewer that these predictions should then be incorporated by circuit-level models, eventually including multiple inhibitory subpopulations of hilar and MEC interneurons. Such circuit modelling is interesting in its own right and warrants separate dedicated studies to carefully consider the large parameter space. We have started to reconcile different types of circuit level models of phase precession (Tsodyks et al. 1996; Romani and Tsodyks 2015) in a published paper (Yiu and Leibold 2023), but even a full-length manuscript has not been adequate to also explore the full parameter space of the different types of oscillatory inputs as are considered here. Therefore, we feel that a study that comprehensively spans circuit to cellular levels considerably exceeds the scope of the current manuscript.

To better explain the rationale of our modelling approach, we rewrote the paragraph that introduces the rationale for the modeling approach.

Line 410-420:

'A phenomenological computational model of phase precession in CA3 cells revealed distinct effects of DG and MEC inputs on the inhibitory signal

To gain further mechanistic understanding of whether and how the effects on phase precession observed in lesion animals can arise from single-cell integration of the two excitatory theta-modulated inputs to CA3, we devised a minimal phenomenological model based on oscillatory interference. We chose to minimize the number of free parameters (see below) of the model to be able to quantitatively fit simulations of the spiking dynamics to the phase precession statistics observed in the data sets. Although our analyses of experimental data had to be limited to the two excitatory inputs to CA3 that were manipulated in lesion experiments, we reasoned that if a computational model based on oscillatory interference were to emulate the lesions, it must account for inputs beyond the manipulated inputs.'

Minor:

The introduction section does not explain the aim of study very well. For example, it remains unclear why it was important to compare DG vs MEC lesions. Also, the authors spent much text on replay which was not the focus of this manuscript. On the other hand, the introduction of the known and proposed mechanisms of phase precession is rather scarce.

Yes, this is an important comment, and we therefore substantially rewrote the introduction.

Line 73-103:

'Although theta phase precession is observed throughout hippocampus and entorhinal cortex, most mechanistic models of phase precession have focused on the CA1 region and on recurrent connections in CA3⁴⁷⁻⁵¹. Involvement of the DG in phase precession has initially been suggested in a model that considered the strong synaptic facilitation of mossy fiber synapses onto CA3 cells as a potential source for increasing excitation throughout the extent of the place field⁵². Conversely, the DG has also been noted in network models as a brain region that can complement the direct recurrent CA3 to CA3 connections by a longer recurrent loop that includes dentate mossy cells and dentate granule cells.'

Accordingly, the connections in this loop have been proposed to include fixed asymmetrical weights that can give rise to sequential and predictive firing patterns^{30, 53}. Although these computational models raise the possibility that DG may contribute to phase precession with a function that differs from other input pathways, it has not been experimentally tested whether DG inputs are even necessary for phase precession in its direct target cells in CA3, and if necessary, whether the observed effects after selective loss of DG inputs can further constrain computational approaches.

While the effects of DG inputs on theta-related spike timing have not been determined, MEC inputs to CA1 are known to be necessary for CA1 phase precession and for CA1 cell pairs to maintain their spiking order in theta cycles⁵⁴. However, given that the MEC inputs to CA3 are complemented by a second extrinsic input from strongly phase precessing cells in DG⁴⁶, it is possible that MEC inputs are not as strictly required for phase precession in CA3 compared to CA1¹². Here, we therefore compared the contributions of DG inputs and of MEC inputs to provide an understanding of the respective contributions of two external excitatory inputs to CA3 theta phase precession. To distinguish the role of the two inputs, we analyzed and compared CA3 network dynamics during theta oscillations from previously published recordings of CA3 cells during working memory tasks with either intact or diminished MEC or DG inputs^{12, 55}. Based on our finding that DG contributes to prospective coding during SWRs¹², we hypothesized that DG also predominantly controls the emergence of prospective coding during theta oscillations. Our results are consistent with a contribution of DG, but not MEC inputs, to prospective coding and to the organization of temporal relations between CA3 neurons. We devised a phenomenological computational model to synthesize these findings and to make predictions of how the summation of inhibitory and excitatory oscillator inputs might support phase precession in CA3.'

Fig3a and c: sorting the order of cells using their mean phase might make sense?

We completely agree and did just that in the original version of the figure. Please note that – for each cell – we calculate and display the median over the mean onset/offset phases of all the cell's trains to obtain one average onset/offset value per cell. For the requested cell-wise ordering, it is therefore most parsimonious to use each cell's median.

The explanation for using the median for ordering was not sufficiently clear in the original text and we therefore made changes to the methods and to the figure legends.

In the methods, we now state:

Line 1065-1067:

'For statistical analysis of onset and offset phases, the median of the mean phases of each cell's trains was taken, which resulted in one value per cell.'

In the figure legend for Fig. 3 (line 713-716), we now state:

'In each of the four raster plots, each row displays the onset phase (Φ_{on}) or offset phase (Φ_{off}) of the trains of a cell (in grey), and the cell's median onset or offset phase (light blue, control; dark blue, DG-lesion). Cells within each panel are sorted from top to bottom by their median onset/offset phase.'

Fig4 will benefit from example spike trains that are distributed to a wide range of theta phase in DG lesion rats, as opposed to late narrower range in control.

We did previously not point out that the increased variability of spike phase in the first half of the field (quantified in Fig. 4d) is depicted in the examples that are included in Fig. 1b. We therefore now explicitly refer to the examples in the figure legend of Fig. 4 (line 763).

'See Fig. 1b for example spike trains that show this effect.'

Fig7d and f should be presented and explained better. The four columns of panels should be clearly indicated on the figure that they correspond to the onset phase, offset phase, etc. Also, onset phase is written as phy_o here but phy_on in Fig7b? In addition, in the panels with DG lesion, what are the black (not pink) dots?

As mentioned by the reviewer, some of the graphical choices and the labelling in Fig. 7d and f (now Fig. 7d) were not conducive to its readability.

-We now consistently use the symbols Φ_{on} and Φ_{off} for onset and offset phase, and φ_{inh} for the phase of the inhibitory oscillation in the model

-We abandoned the use of pink outlines (which had the appearance of pink dots for small parameter spaces) and consistently use black/white outlines to highlight the model parameter space that explains data. We also increased the line weight of the outlines to make the zone that explains the data more prominent when plotted on top of color-coded plots.

-We updated the color code for Φ_{on} , Φ_{off} , Slope, and R^2 to make the appearance of the patterns in the parameter space more distinct.

-We rearranged the panels such that the parameters are now ordered vertically, which allowed us to place the masks for the correspondence between data and model to the right ('admissible parameter space').

- We reconsidered how to depict the overlap of admissible parameter spaces. Admissible space for each parameter is now shown in white and non-admissible space in blue. This allows us to render the zone of overlapping admissible spaces across parameters (Φ_{on} , Φ_{off} , Slope, and R^2) in white and zones with an increasing number of overlapping non-admissible spaces with increasingly darker shades of blue. This color scheme yields a more intuitive way of reading the summary panel on the bottom.

- The order of the parameters (Φ_{on} , Φ_{off} , Slope, and R^2) were adjusted such that Slope is now on top to provide a more intuitive progression of details.

-The figure legend of Fig. 7 is substantially updated.

Line 814-852:

'Figure 7. A single-cell model with two oscillating excitatory inputs and an inhibitory input reproduced the main empirical results. a, Model construction. The three inputs are modeled after DG, MEC, and local inhibition converging onto the CA3 model neuron. The excitatory DG and MEC inputs oscillate at faster-than-LFP frequencies ($\omega_{DG} = 8.6$ Hz, $\omega_{MEC} = 8.5$ Hz) with DG inputs more prominent early in the field and MEC inputs more prominent later in the field. The inhibitory input oscillates at 8 Hz throughout the place field, corresponding to LFP theta. Small Gaussian noise is added to the inhibitory input to ensure robustness against minor perturbations. The excitatory inputs contribute positively at the fixed phase difference ψ , which is taken to be 0° from published findings⁶⁸ on DG and MEC population activity. The inhibitory input contributes negatively to the total drive at a phase differential φ_{inh} relative to excitation at place field entry. Finally, the total drive is rectified. A reference 8-Hz oscillatory inhibition is displayed at the bottom left, which is used to extract the phase of the simulated spikes. The phase-distance

relationship is then depicted as for experimental data. Not all steps are displayed for brevity (see Methods for full details). **b**, Phase-versus-normalized distance plots of spikes generated by the model. Three randomly selected single pass examples show phase precession. The values of $A/\varphi_{inh} = 3/260^\circ$ (inhibitory oscillation amplitude and phase), $I_{DC} = 0$ (inhibition DC component) and $\psi = 0^\circ$ (excitatory phase differential) are the same across the three plots. The measured slopes from the simulated data are displayed at the top of each panel along with the significance. **c**, Phase-versus-normalized distance plot for spikes across multiple passes. Panels from top to bottom are generated by the control, DG lesion, and MEC lesion models (1 cycle = 360°). The measured slope and significance of phase precession are displayed on top of each panel. n.s., not significant, * $p < 0.05$, *** $p < 0.001$. Lesion experiments were simulated by setting the DG or MEC input to zero, and all other parameters as in **b** except for A lesion = 2. In both lesion cases, phase precession slopes are reduced. (**d** and **e**) DG and MEC lesions alter CA3 phase precession in qualitatively different ways. **d**, Values of slope, explained variance (R^2), onset phase (Φ_{on}), and offset phase (Φ_{off}) are shown for combinations of A and φ_{inh} parameters. The color scale in each panel is according to the color bar to the right. The range of model A - φ_{inh} parameter space that corresponds to empirical phase precession values are shown in blue and white plots to the right with white areas depicting the space that yields 80% of the empirical measurements. The intersection between white areas for multiple phase precession measurements is displayed in the overlap plots at the bottom (dark blue to white, 0 to 4 measurements overlap). The zone of overlap for all four measurements is delineated with outlines (black or white lines) that are projected back onto all other panels. **e**, left, Distribution of parameter values A and φ_{inh} that result in a match with the empirical control and lesion data. To match the empirical phase precession measurements, the DG-lesion but not the MEC-lesion model is forced to take on a shifted set of φ_{inh} values. **e**, right, Distributions of Φ_{on} and Φ_{off} values that are generated by the A - φ_{inh} parameter space that corresponds to the overlap areas of each model type. The DG lesion model generates the broadest Φ_{on} distribution. DG control, light blue; DG lesion, dark blue; MEC control, yellow; MEC lesion, red.'

It would be better if the model can provide a possible mechanism as to how relative timing of place cells can be preserved in the case of MEC lesion.

The model does not include behavioral time, and these analyses are therefore not feasible with a cellular model and would require a circuit level model. As pointed out in our response to the major comments, such circuit modelling is interesting in its own right and warrants separate dedicated studies to carefully consider the large parameter space. Therefore, we feel that inclusion of these simulations to the present manuscript considerably exceeds its scope.

References

- Colgin, L. L., T. Denninger, M. Fyhn, T. Hafting, T. Bonnevie, O. Jensen, M. B. Moser, and E. I. Moser. 2009. 'Frequency of gamma oscillations routes flow of information in the hippocampus', *Nature*, 462: 353-7.
- Dragoi, G., and G. Buzsaki. 2006. 'Temporal encoding of place sequences by hippocampal cell assemblies', *Neuron*, 50: 145-57.
- Harvey, C. D., F. Collman, D. A. Dombeck, and D. W. Tank. 2009. 'Intracellular dynamics of hippocampal place cells during virtual navigation', *Nature*, 461: 941-6.

- Magee, J. C. 2001. 'Dendritic mechanisms of phase precession in hippocampal CA1 pyramidal neurons', *J Neurophysiol*, 86: 528-32.
- Mehta, M. R., A. K. Lee, and M. A. Wilson. 2002. 'Role of experience and oscillations in transforming a rate code into a temporal code', *Nature*, 417: 741-6.
- O'Keefe, J., and M. L. Recce. 1993. 'Phase relationship between hippocampal place units and the EEG theta rhythm', *Hippocampus*, 3: 317-30.
- Romani, S., and M. Tsodyks. 2015. 'Short-term plasticity based network model of place cells dynamics', *Hippocampus*, 25: 94-105.
- Schlesinger, M. I., C. C. Cannova, B. L. Bublil, J. B. Hales, E. A. Mankin, M. P. Brandon, J. K. Leutgeb, C. Leibold, and S. Leutgeb. 2015. 'The medial entorhinal cortex is necessary for temporal organization of hippocampal neuronal activity', *Nat Neurosci*, 18: 1123-32.
- Thurley, K., C. Leibold, A. Gundlfinger, D. Schmitz, and R. Kempfer. 2008. 'Phase precession through synaptic facilitation', *Neural Comput*, 20: 1285-324.
- Treves, A. 2004. 'Computational constraints between retrieving the past and predicting the future, and the CA3-CA1 differentiation', *Hippocampus*, 14: 539-56.
- Treves, A., and E. T. Rolls. 1992. 'Computational constraints suggest the need for two distinct input systems to the hippocampal CA3 network', *Hippocampus*, 2: 189-99.
- Tsodyks, M. V., W. E. Skaggs, T. J. Sejnowski, and B. L. McNaughton. 1996. 'Population dynamics and theta rhythm phase precession of hippocampal place cell firing: A spiking neuron model', *Hippocampus*, 6: 271-80.
- Yiu, Y. H., and C. Leibold. 2023. 'A theory of hippocampal theta correlations accounting for extrinsic and intrinsic sequences', *Elife*, 12.

NCOMMS-23-03987B

We thank the three reviewers for their time to carefully read the rebuttal and our revised manuscript. We appreciate that they considered the manuscript 'significantly clearer and improved in quality' and even minor concerns 'beautifully handled'. We always strive to improve our work and appreciate that the reviewers contributed to substantially improve the quality of the manuscript. With regard to our responses to reviewer #3, we are pleased to hear that we could address at least a subset of the concerns adequately and apologize that there are points on which we continue to disagree. While we find the suggestions extremely helpful, we continue to stress that many of them are so comprehensive that incorporating them in a rigorous way by far exceeds the scope of the manuscript (e.g., building and tuning of spiking neural networks). We realize that this is in many ways an unsatisfactory response, but we would nonetheless ask for understanding that we cannot include extensions of multiple previous papers in a single manuscript. We have provided further detail on our reasoning and also thoroughly rewritten the discussion to further explain limitations and possible future studies. Our comments to each concern are described in detail in the point-by-point responses (**with reviewer comments retained in black and our responses added in blue**), and we are also summarizing the revisions in the manuscript. **A manuscript copy with key revisions highlighted in red is included with the resubmission.**

POINT-BY-POINT RESONSES TO REVIEWER COMMENTS

Reviewer #1 (Remarks to the Author):

The results of this revised manuscript, through much enriched, largely support the same set of conclusions as the initial submission. This new version is an extremely nice piece of work. I only had three concerns that I considered to be substantial before and each have been thoroughly addressed. Incidentally, I had also raised a number of what I called 'minor concerns' and these, too, were beautifully handled. I was enthusiastic about this work already and that enthusiasm has only grown. I have no hesitations in recommending that this manuscript be advanced to publication.

Reviewer #2 (Remarks to the Author):

I want to compliment the authors for carefully addressing all my concerns, with a set of extensive reviews. The manuscript is significantly clearer and improved in quality. I don't have any further comments and, in my opinion, the paper is now suitable for publication.

Reviewer #3 (Remarks to the Author):

The key observation of this work is that MEC and DG differently affect phase precession in CA3. Unfortunately, the revision could not provide any additional support for the proposed mechanism behind this difference. The authors responded to my major comment #2 that the data analysis remains descriptive. In addition, it is difficult to eliminate the possibility that this main observation stems from a different extent of the lesions as pointed out by my major comment 6 as well as by the reviewer 1. Their additional analysis on this aspect indicates that the MEC lesion might have affected the activity of CA3 more strongly, suggesting different extent of the lesions. Together with the lack of correlation between phase precession change and behaviour, scientific advance made by this work might be limited. Therefore, my main concerns were not resolved overall.

The main concern that was previously voiced is that a larger DG lesion effect compared to MEC lesion effect could result in a disruption of the temporal order of CA3 cells for DG, but not MEC lesions. Based on the revised manuscript, which showed that there are minor effects of MEC, but not DG lesions on firing rates in spike trains, the opposite possibility is now brought up. If we understand the concern correctly, the minor effects of MEC lesions on firing rate might affect spike timing over the full extent of the theta cycle, compared to the more limited effect of DG on only late phases of the cycle. There are several observations that are not consistent with this interpretation.

- (1) This line of reasoning would not explain that the DG lesion effects exceed the MEC lesion effects for the key observation that spike timing is disrupted with only DG lesions (e.g., Fig. 6d and h).
- (2) The overall effect of each lesion on phase precession is quantitatively comparable (Figs. 1 and 2)
- (3) The onset phase broadens after DG, not after MEC lesions (Fig. 1b and e). If MEC lesions had a broader effect throughout the entire theta phase, a redistribution of spike phase throughout the cycle would be expected to also occur with MEC lesions. However, the MEC onset phase distribution is precisely matched between control spike trains and spike trains from MEC-lesioned rats (Fig. 1e).
- (4) Similarly, circular variance is increased selectively in the first half of trains with DG lesions, but not in any part of the trains with MEC lesions. While the argument that the MEC lesion effects are more uniform throughout the cycle may hold up here, it is only with regard to seeing no effects in any part of the cycle rather than with seeing stronger effects throughout the cycle.

If we were pointed to evidence that suggests a broader effect of MEC lesions compared to DG lesions, we would of course point out these results in the discussion.

Some of the points were already mention in the previous revision, and the paragraph that discusses the relative effect size of MEC compare to DG lesions has now been expanded to emphasize additional pieces of evidence that are inconsistent with the interpretation that there are merely quantitative differences between the two types of lesions.

Line 538-552

Importantly, neither the pronounced broadening of the onset phase nor the selective effects on spike timing at the onset of spiking were observed with MEC lesions, which nonetheless reduced phase precession in CA3 to a similar extent as the DG lesions. Given that MEC layer II does not only project directly to CA3, but also to DG⁷⁵, it might have been expected that MEC lesions result in larger deficits when direct effects on CA3 and indirect effects via DG on CA3 combine. For example, it could have caused the minor decrease of MEC firing rates with MEC lesions. However, the preserved pairwise spike timing of CA3 after MEC lesions and the moderately preserved phase precession in CA3 after MEC lesions differ from the profound disruption of pairwise spike timing with DG lesions and from the previously reported profound disruption of the temporal order in pairs of CA1 cells and of phase precession in CA1 cells after MEC lesions^{54, 76}. Our results also exclude the interpretation that the MEC lesions have effects over a broader range of theta phases than the DG lesions. The onset phases broadened with DG lesions, but were precisely matched to controls with MEC lesions. Similarly, circular variance increased selectively in the first half of trains with DG lesions, and not in

any part of the trains with MEC lesions.'

Major comment 3: "To support the hypothesis that inhibitory input ..."

It is unfortunate that there are not enough putative interneurons to analyse. However, there are multiple papers which analysed putative interneurons recorded from the electrodes in the CA3 pyramidal cell layer. In this sense, I must point out that it is a short-coming of the current work that the main mechanisms the authors demonstrate from the model is relying on recordings they don't have. In my opinion, such recordings will be a minimum requirement to suggest inhibition as the main mechanisms underlying different phase precession they observe.

Given the challenges of performing stable recordings in lesioned rats, we focused on maximizing the number of principal cells. An important additional limitation here is that we could at most identify one major type of interneuron in the pyramidal cell layer (soma-targeting basket cells) while dendrite-targeting cells do not have their cell body in the pyramidal cell layer and cannot be simultaneously recorded. Furthermore, even if these recordings were performed and included, it would not be possible to reliably distinguish different interneuron subtypes. We therefore continue to posit that rigorously performing these recordings would considerably exceed the scope of the study. In addition, it is not uncommon that publications that include experimental data and models make predictions that are not immediately tested.

The rigor of the studies that is needed for the proposed experimental tests of the model is now more explicitly stated.

Line 624 to 630:

'Although these predictions from the model are yet to be confirmed by recording from identified dendrite-targeting and soma-targeting interneurons, these data suggest that effects from manipulating excitatory inputs do not only arise from diminished direct connectivity to principal cells, but also from how these inputs engage inhibitory interneurons. In particular, our data are consistent with the mossy fiber inputs to CA3 more strongly engaging somatic inhibition, which determines the theta phase of spikes, and with MEC more strongly engaging dendritic inhibition which does not directly set the theta phase.'

Major comment 4: "The recurrent excitatory network in CA3..."

It might be ok not to include CA3 recurrent network if empirical data of the authors suggest the importance of inhibition. However, that is not the case here. In this case, a more convincing modelling approach would be to test multiple known models and come to a conclusion that inhibition can but recurrent excitation, for example, cannot explain their data.

It is our reading of this comment that the proposed approach would need to include a comparison between multiple implementations of spiking network models, which is certainly a topic that one or more additional manuscripts could be covering. We are well aware that this is an important new direction based on our data, and this is the direction that we are planning on pursuing in future research studies.

To better emphasize the scope of the simulations that are needed, we explicitly mention which inputs and interneuron types would need to be considered in the future studies with spiking neural networks.

Line 565-574:

'While this observation is inconsistent with an early phase precession model that uses asymmetric synaptic weights in the recurrent CA3 network to generate phase precession⁴⁷, it has more recently been shown that recurrent networks need to be combined with external inputs or with mechanisms that lead to firing frequency adaptation to robustly generate phase precession or predictive coding^{51, 53, 77}. While our data or any data that we are aware of do not directly test a role of recurrent collaterals, our findings support the notion that external inputs to CA3 are needed in addition to or in lieu of recurrent circuits for the generation of phase precession and for precise theta-scale spike timing. To test the suggested role of DG, spiking network models will need to be developed that consider separate DG and MEC inputs to CA3 in conjunction with a separate role of somatic and dendritic inhibition.'

Major comment 5: "It is not clear what ..."

Unfortunately, the revision could not improve this aspect and there still is no link between lesion types causing different phase precession deficits and behaviour.

We now further emphasized the importance of linking phase precession to memory in our discussion (line 630-634), but we would also like to point out that this needs to be addressed in the broader context of linking sequential firing (e.g., theta sequences) to memory processes. This is currently a topic that many laboratories are investigating, and we hope that our data will be the foundation for including the role of DG and MEC in these studies.

'The different functions of input pathways imply that manipulations and models that link phase precession to theta sequences and theta sequences to behavior will need to consider multiple input pathways and perhaps even a much larger circuit that includes other brain regions with phase precession, such as MEC and CA1.'

Major comment 6: "The different effect of DG vs MEC lesions"

Their additional analysis on this was useful and additional simulation of different excitation strength strengthened the model. However, I think the larger effect of MEC lesion they found could have caused a larger effect of MEC lesion on phase precession (affecting entire phase).

Please see our responses to the additional comments on previous comment #2

Major comment 7: "The computational models are...".

Please see my comment on the comment #4 above.

Please see our responses to the additional comments on previous comment #4